# Visual homogeneity computations in the brain enable solving property-based visual tasks

Georgin Jacob*, RT Pramod[†], SP Arun*

Centre for Neuroscience & Department of Electrical Communication Engineering, Indian Institute of Science, Bangalore, India

## eLife Assessment

This study uses carefully designed experiments to generate a **useful** behavioural and neuroimaging dataset on visual cognition. The results provide **solid** evidence for the involvement of higher-order visual cortex in processing visual oddballs and asymmetry. However, the evidence provided for the very strong claims of homogeneity as a novel concept in vision science, separable from existing concepts such as target saliency, is **incomplete**. The authors and the reviewers do not agree on several points, which are explained in the reviews and author response.

*For correspondence:
georginjacob@gmail.com (GJ);
sparun@iisc.ac.in (SPA)

Present address: [†]Massachusetts Institute of Technology, Cambridge, MA, United States

**Abstract** Most visual tasks involve looking for specific object features. But we also often perform property-based tasks where we look for specific property in an image, such as finding an odd item, deciding if two items are same, or if an object has symmetry. How do we solve such tasks? These tasks do not fit into standard models of decision making because their underlying feature space and decision process is unclear. Using well-known principles governing multiple object representations, we show that displays with repeating elements can be distinguished from heterogeneous displays using a property we define as visual homogeneity. In behavior, visual homogeneity predicted response times on visual search, same-different and symmetry tasks. Brain imaging during visual search and symmetry tasks revealed that visual homogeneity was localized to a region in the object-selective cortex. Thus, property-based visual tasks are solved in a localized region in the brain by computing visual homogeneity.

## Introduction

Many visual tasks involve looking for specific objects or features, such as a friend in a crowd or selecting vegetables in the market. In such tasks, which have been studied extensively, we form a template in our brain that helps guide eye movements and locate the target (*Peelen and Kastner, 2014*). This template becomes a decision variable that can be used to solve the task: the degree of match to the template indicates the presence or absence of the desired object (*Figure 1A*). However, we also easily perform tasks that do not involve any specific feature template but rather involve finding a property in the image. Examples of such property-based tasks are same-different task, finding the oddball item and judging if an object has symmetry (*Figure 1B*). These tasks cannot be solved by looking for any particular feature, and therefore present a major challenge for standard models of decision making since the underlying feature space and decision variable are unknown. Even machine vision algorithms, which are so successful at solving feature-based tasks (*Serre, 2019*), fail at detecting properties like same-different and at other similar visual reasoning challenges (*Fleuret et al., 2011*; *Kim*

**Figure 1.** Feature-based and property-based visual tasks. (**A**) *Feature-based visual tasks.* Most visual tasks involve making decisions based on looking at specific features. Face recognition is shown here as an example. According to standard theories of decision making, such tasks are solved in the brain by setting up a decision variable in a multidimensional feature space (*arrow*), and making decisions based on whether the value of the decision variable is larger or smaller than a decision boundary (*dashed line*). (**B**) *Property-based visual tasks.* By contrast, some tasks involve detecting properties in the image, such as a same-different task (illustrated using faces; *top row*), detecting an oddball item (illustrated using *middle row*) or judging if an object is symmetric (*bottom row*). These tasks cannot be solved by looking for any specific feature. As a result, such tasks do not fit into standard models of decision making since the underlying feature space and decision variable are unknown.

*et al., 2018*; *Ricci et al., 2021*; *Puebla and Bowers, 2022*). How do we solve such property-based visual tasks? What are the underlying features and what is the decision variable?

To start with, these tasks appear completely different, at least in the way we describe them verbally. Even theoretical studies have considered visual search (*Verghese, 2001*; *Wolfe and Horowitz, 2017*), same-different judgments (*Nickerson, 1969*; *Petrov, 2009*) and symmetry detection (*Wagemans, 1997*; *Bertamini and Makin, 2014*) as conceptually distinct tasks. However, we note that at a deeper level, these tasks are similar because they all involve discriminating between items with repeating features from those without repeating features. We reasoned that if images with repeating features are somehow represented differently in the brain, this difference could be used to solve all such tasks without requiring separate computations for each task.

## Key predictions

Our key predictions are summarized in *Figure 2*. Consider an oddball search task where participants have to indicate if a display contains an oddball target (*Figure 2A*) or not (*Figure 2B*). According to the well-known principle of divisive normalization in high-level visual cortex (*Zoccolan et al., 2005*; *Agrawal et al., 2020*; *Katti and Arun, 2022*), the neural response to multiple objects is the average of the single object responses. Accordingly, the response to an array of identical items will be the same as the response to the single item. Similarly, the response to an array containing a target among distractors, being a mix of images, would lie along the line joining the target and distractor in underlying representational space. These possibilities are shown for all possible arrays made from three objects in *Figure 2C*. It can be seen that the homogeneous (target-absent) arrays naturally stand apart since they contain repeating items, whereas the heterogeneous (target-present) arrays come closer to each other since they contain a mixture of items. Since displays with repeating items are further away from the center of this space, this distance can be used to discriminate them from heterogeneous displays (*Figure 2C*, *inset*). While this decision process may not capture all types of visual search, we reasoned that it could potentially guide the initial stages of target selection.

We reasoned similarly for symmetry detection: here, participants have to decide if an object is asymmetric (*Figure 2D*) or symmetric (*Figure 2E*). According to multiple object normalization, objects with two different parts would lie along the line joining objects containing the two repeated parts

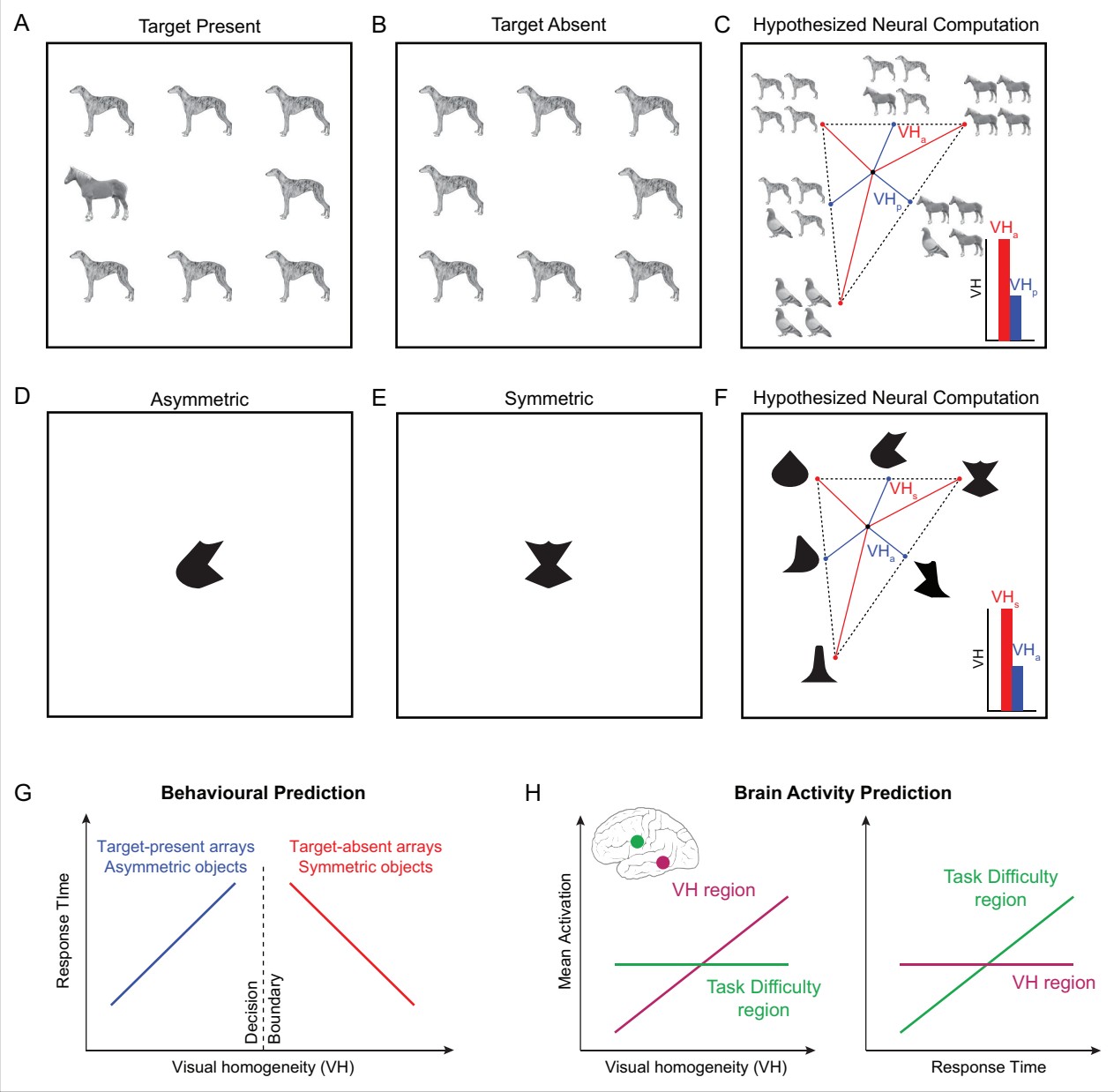

**Figure 2.** Solving oddball search and symmetry tasks using visual homogeneity. (**A**) Example target-present search display, containing a single oddball target (horse) among identical distractors (dog). Participants in such tasks have to indicate whether the display contains an oddball or not, without knowing the features of the target or distractor. This means they have to perform this task by detecting some property of each display rather than some feature contained in it. (**B**) Example target-absent search display containing no oddball target. (**C**) Hypothesized neural computation for target present/absent judgements. According to multiple object normalization, the response to multiple items is an average of the responses to the individual items. Thus, the response to a target-absent array will be identical to the individual items, whereas the response to a target-present array will lie along the line joining the corresponding target-absent arrays. This causes the target-absent arrays to stay apart (*red lines*), and the target-present arrays to come closer due to mixing (*blue lines*). If we calculate the distance (VH, for visual homogeneity) for each display, then target-absent arrays will have a larger distance to the center (VH$_a$) compared to target-present arrays (VH$_p$), and this distance can be used to distinguish between them. *Inset:* Schematic distance from center for target-absent arrays (red) and target-present arrays (blue). Note that this approach might only reflect the initial target selection process involved in oddball visual search and may not capture all forms of visual search. Nonetheless it is a quantitative and falsifiable model. (**D**) Example asymmetric object in a symmetry detection task. Here too, participants have to indicate if the display contains a symmetric object or not, without knowing the features of the object itself. This means they have to perform this task by detecting some property in the display. (**E**) Example symmetric object in a symmetry detection task. (**F**) Hypothesized neural computations for symmetry detection. Following multiple object normalization, the response to an object containing repeated parts is equal the response to the individual part, whereas the response to an object containing two different parts will lie along the line joining the objects with the two parts repeating. This causes symmetric objects to stand apart (red

*Figure 2 continued on next page*

*Figure 2 continued*

lines) and asymmetric objects to come closer due to mixing (*blue lines*). Thus, the visual homogeneity for symmetric objects (VH$_s$) will be larger than for asymmetric objects (VH$_a$). *Inset:* Schematic distance from center for symmetric objects (red) and asymmetric objects (blue). (**G**) *Behavioral predictions for VH.* If visual homogeneity (VH) is a decision variable in visual search and symmetry detection tasks, then response times (RT) must be largest for displays with VH close to the decision boundary. This predicts opposite correlations between response time and VH for the present/absent or symmetry/asymmetry judgements. It also predicts zero overall correlation between VH and RT. (**H**) *Neural predictions for VH. Left:* Correlation between brain activations and VH for two hypothetical brain regions. In the VH-encoding region, brain activations should be positively correlated with VH. In any region that encodes task difficulty as indexed by response time, brain activity should show no correlation since VH itself is uncorrelated with RT (see Panel **G**). *Right:* Correlation between brain activations and RT. Since VH is uncorrelated with RT overall, the region VH should show little or no correlation, whereas the regions encoding task difficulty would show a positive correlation.

(*Figure 2F*). Indeed, both symmetric and asymmetric objects show part summation in their neural responses (*Pramod and Arun, 2018*). Consequently, symmetric objects will be further away from the centre of this space compared to asymmetric objects, and this can be the basis for distinguishing them (*Figure 2F*, *inset*).

We define this distance from the center for each image as its *visual homogeneity* (VH). We made two key experimental predictions to test in behavior and brain imaging. First, if VH is being used to solve visual search and symmetry detection tasks, then responses should be slowest for displays with VH close to the decision boundary and faster for displays with VH far away (*Figure 2G*). This predicts opposite correlations between response time and VH: for target-present arrays and asymmetric objects, the response time should be positively correlated with VH. By contrast, for target-absent arrays and symmetric objects, response time should be negatively correlated with VH. Importantly, because response times of the two choices are positively and negatively correlated with VH, the net correlation between response time and VH will be close to zero.

Second, if VH is encoded by a dedicated brain region, then brain activity in that region will be positively correlated with VH (*Figure 2H*). Such a positive correlation cannot be explained easily by cognitive processes linked to response time such as attention or task difficulty, since response times will have zero correlation with the mean activity of this region.

## Overview of this study

The above predictions are based on computing distances in an underlying neural representation that is the basis for a variety of visual tasks. We drew upon the principle that object representations in high-level visual cortex match strongly with perceived dissimilarity measured in visual search (*Sripati and Olson, 2010*; *Zhivago and Arun, 2014*; *Agrawal et al., 2020*). We therefore asked whether distance-to-center or visual homogeneity computations, performed on object representations estimated empirically from dissimilarity measurements, could explain the behavior and brain imaging predictions laid out in the preceding section (*Figure 2*).

We performed two sets of experiments to test our key predictions. In the first set, we measured perceptual dissimilarities on a set of grayscale natural images using visual search (Experiment 1) and then asked whether visual homogeneity computations on this estimated representation could explain response times as well as brain activations during a oddball target detection task (Experiment 2). In the second set of experiments, we measured perceptual dissimilarities between a set of silhouette images (Experiment 3) and asked whether visual homogeneity computations on this estimated representation could explain response times and brain activations during a symmetry detection task on these images (Experiment 4).

## Results

In Experiments 1–2, we investigated whether visual homogeneity computations could explain decisions about targets being present or absent in an array. Since visual homogeneity requires measuring distance in perceptual space, we set out to first characterize the underlying representation of a set of natural objects using measurements of perceptual dissimilarity.

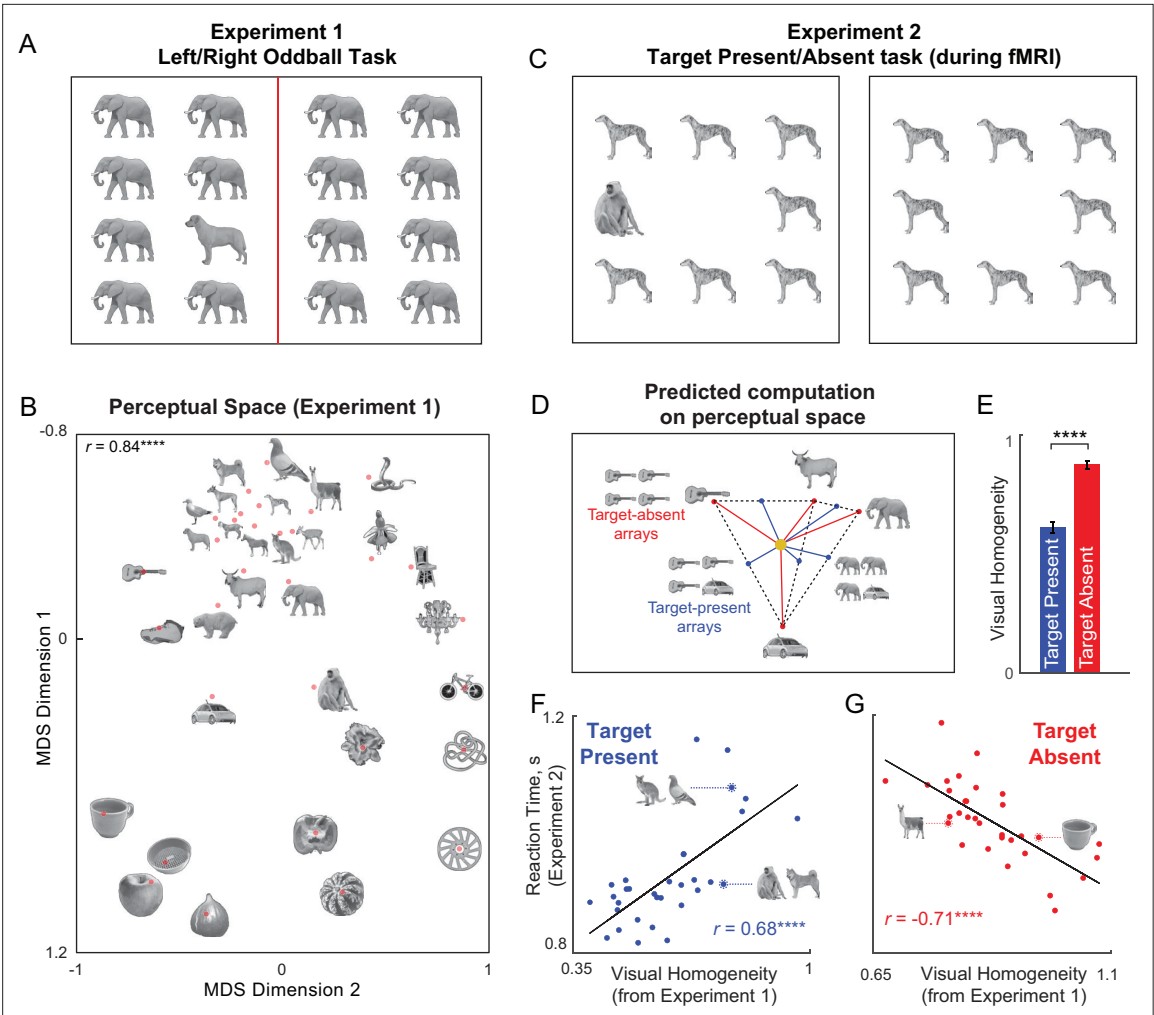

**Figure 3.** Visual homogeneity predicts target present/absent responses. (**A**) Example search array in an oddball search task (Experiment 1). Participants viewed an array containing identical items except for an oddball present either on the left or right side, and had to indicate using a key press which side the oddball appeared. The reciprocal of average search time was taken as the perceptual distance between the target and distractor items. We measured all possible pairwise distances for 32 grayscale natural objects in this manner. (**B**) Perceptual space reconstructed using multidimensional scaling performed on the pairwise perceptual dissimilarities. In the resulting plot, nearby objects represent hard searches, and far away objects represent easy searches. Some images are shown at a small size due to space constraints; in the actual experiment, all objects were equated to have the same longer dimension. The correlation on the top right indicates the match between the distances in the 2D plot with the observed pairwise distances (**** is p<0.00005). (**C**) Example display from Experiment 2. Participants performed this task inside the scanner. On each trial, they had to indicate whether an oddball target is present or absent using a key press. (**D**) Predicted response to target-present and target-absent arrays, using the principle that the neural response to multiple items is the average of the individual item responses. This predicts that target-present arrays become similar due to mixing of responses, whereas target-absent arrays stand apart. Consequently, these two types of displays can be distinguished using their distance to a central point in this space. We define this distance as visual homogeneity, and it is obtained by finding the optimum center that maximizes the difference in correlations with response times (see Methods). (**E**) Mean visual homogeneity relative to the optimum center for target-present and target-absent displays. Error bars represent s.e.m across all displays. Asterisks represent statistical significance (**** is p<0.00005, unpaired rank-sum test comparing visual homogeneity for 32 target-absent and 32 target-present arrays). (**F**) Response time for target-present searches in Experiment 2 plotted against visual homogeneity calculated from Experiment 1. Asterisks represent statistical significance of the correlation (**** is p<0.00005). Note that a single model is fit to find the optimum center in representational space that predicts the response times for both target-present and target-absent searches. (**G**) Response time for target-absent searches in Experiment 2 plotted against visual homogeneity calculated from Experiment 1. Asterisks represent statistical significance of the correlation (**** is p<0.00005).

## Measuring perceptual space for natural objects

In Experiment 1, 16 human participants viewed arrays made from a set of 32 grayscale natural objects, with an oddball on the left or right (*Figure 3A*), and had to indicate the side on which the oddball appeared using a key press. Participants were highly accurate and consistent in their

responses during this task (accuracy, mean ± sd: 98.8 ± 0.9%; correlation between mean response times of even- and odd-numbered participants: $r$=0.91, p<0.0001 across all $^{32}C_2$=496 object pairs). The reciprocal of response time is a measure of perceptual distance (or dissimilarity) between the two images (*Arun, 2012*). To visualize the underlying object representation, we performed a multidimensional scaling analysis, which embeds objects in a multidimensional space such that their pairwise dissimilarities match the experimentally observed dissimilarities (see Materials and methods). The resulting two-dimensional embedding of all objects is shown in *Figure 3B*. In the resulting plot, nearby objects correspond to hard searches, and far away objects correspond to easy searches. Such representations reconstructed from behavioural data closely match population neural responses in high-level visual areas (*Op de Beeck et al., 2001*; *Sripati and Olson, 2010*). To capture the object representation accurately, we took the multidimensional embedding of all objects and treated the values along each dimension as the responses of an individual artificial neuron. We selected the number of dimensions in the multidimensional embedding so that the correlation between the observed and embedding dissimilarities matches the noise ceiling in the data. Subsequently, we averaged these single object responses to obtain responses to larger visual search arrays, as detailed below.

## Visual homogeneity predicts target present/absent judgments (Experiments 1-2)

Having characterized the underlying perceptual representation for single objects, we set out to investigate whether target present/absent responses during visual search can be explained using this representation. In Experiment 2, 16 human participants viewed an array of items on each trial, and indicated using a key press whether there was an oddball target present or not (*Figure 3C*). This task was performed inside an MRI scanner to simultaneously observe both brain activity and behaviour. Participants were highly accurate and consistent in their responses (accuracy, mean ± sd: 95 ± 3%; correlation between average response times of even- and odd-numbered participants: $r$=0.86, p<0.0001 across 32 target-present searches, $r$=0.63, p<0.001 across 32 target-absent searches).

Next we set out to predict the responses to target-present and target-absent search displays containing these objects. We first took the object coordinates returned by multidimensional scaling in Experiment 1 as the neural responses of multiple neurons. We then used the well-known principle of object representations in high-level visual areas: the response to multiple objects is the average of the single object responses (*Zoccolan et al., 2005*; *Agrawal et al., 2020*). Thus, we took the response vector for a target-present array to be the average of the response vectors of the target and distractor (*Figure 3D*). Likewise, we took the response vector for a target-absent array to be equal to the response vector of the single item. We then asked if there is any point in this multidimensional representation such that distances from this point to the target-present and target-absent response vectors can accurately predict the target-present and target-absent response times with a positive and negative correlation, respectively (see Materials and methods). Note that this model has only five free parameters, which are the coordinates of this unknown point or center in multidimensional space, and this model can simultaneously predict both target-present and target-absent judgments. We used nonlinear optimization to find the coordinates of the center to best match the data (see Materials and methods).

We denoted the distance of each display to the optimized center as the visual homogeneity. As expected, the visual homogeneity of target-present arrays was significantly smaller than target-absent arrays (*Figure 3E*). The resulting model predictions are shown in *Figure 3F–G*. The response times for target-present searches were positively correlated with visual homogeneity ($r$=0.68, p<0.0001; *Figure 3F*). By contrast, the response times for target-absent searches were negatively correlated with visual homogeneity ($r$=–0.71, p<0.0001; *Figure 3G*). This is exactly as predicted if visual homogeneity is the underlying decision variable (*Figure 2G*). We note that the range of visual homogeneity values for target-present and target-absent searches do overlap, suggesting that visual homogeneity contributes but does not fully determine task performance. Rather, we suggest that visual homogeneity provides a useful and initial first guess at the presence or absence of a target, which can be refined further through detailed scrutiny.

## Confirming the generality of visual homogeneity

We performed several additional analysis to confirm the generality of our results, and to reject alternate explanations.

First, it could be argued that our results are circular because they involve taking oddball search times from Experiment 1 and using them to explain visual search response times in Experiment 2. While this might appear so, we are merely using the search dissimilarities from Experiment 1 only as a proxy for the underlying neural representation, based on previous reports that neural dissimilarities closely match oddball search dissimilarities (*Sripati and Olson, 2010*; *Zhivago and Arun, 2014*). Nonetheless, to thoroughly refute this possibility, we reasoned that we would get similar predictions of the target present/absent responses in Experiment 2 using any other brain-like object representation. To confirm this, we replaced the object representations derived from Experiment 1 with object representations derived from deep neural networks pretrained for object categorization, and asked if distance-to-center computations could predict the target present/absent responses in Experiment 2. This was indeed the case (Appendix 1).

Second, we wondered whether the nonlinear optimization process of finding the best-fitting center could be yielding disparate optimal centres each time. To investigate this, we repeated the optimization procedure with many randomly initialized starting points, and obtained the same best-fitting center each time (see Materials and methods).

Third, to confirm that the above model fits are not due to overfitting, we performed a leave-one-out cross validation analysis. We left out all target-present and target-absent searches involving a particular image, and then predicted these searches by calculating visual homogeneity estimated from all other images. This too yielded similar positive and negative correlations ($r$=0.63, p<0.0001 for target-present, $r$=–0.63, p<0.001 for target-absent).

Fourth, if heterogeneous displays indeed elicit similar neural responses due to mixing, then their average distance to other objects must be related to their visual homogeneity. We confirmed that this was indeed the case, suggesting that the average distance of an object from all other objects in visual search can predict its visual homogeneity (Appendix 2).

Fifth, the above results are based on taking the neural response to oddball arrays to be the average of the target and distractor responses. To confirm that averaging was indeed the optimal choice, we repeated the above analysis by assuming a range of relative summation weights between the target and distractor. The best correlation was obtained for almost equal weights in the lateral occipital (LO) region, consistent with averaging and its role in the underlying perceptual representation (Appendix 2).

Finally, we performed several additional experiments on a larger set of natural objects as well as on silhouette shapes. In all cases, present/absent responses were explained using visual homogeneity (Appendix 3).

## Conclusions

These findings are non-trivial for several reasons. First, we have shown that a specific decision variable, computed over an underlying object representation, can be used to make target present/absent judgements, without necessarily knowing the precise features of the target or distractor. Second, we have identified a specific image property that explains why target-absent response times vary so systematically. If target-distractor dissimilarity were the sole determining factor in visual search, it would predict no systematic variation in target-absent searches since the target-distractor dissimilarity is zero. Our results elucidate this puzzle by showing that visual homogeneity varies systematically across images, and that this explains systematic variation in target-absent response times. To the best of our knowledge, our model provides for the first time a unified explanation to explain how target present/absent judgements might be made in a generic visual search task.

In sum, we conclude that visual homogeneity can explain oddball target present/absent judgements during visual search.

## Visual homogeneity predicts same/different responses

We have proposed that visual homogeneity can be used to solve any task that requires discriminating between homogeneous and heterogeneous displays. In Experiments 1–2, we have shown that visual homogeneity predicts target present/absent responses in visual search. We performed an additional

experiment to assess whether visual homogeneity can be used to solve an entirely different task, namely a same-different task. In this task, participants have to indicate whether two items are the same or different. We note that instructions to participants for the same/different task ('you have to indicate if the two items are same or different') are quite different from the visual search task ('you have to indicate whether an oddball target is present or absent'). Yet both tasks involve discriminating between homogeneous and heterogeneous displays. We therefore predicted that 'same' responses would be correlated with target-absent judgements and 'different' responses would be correlated with target-present judgements. Remarkably, this was indeed the case (Appendix 4), demonstrating that same/different responses can also be predicted using visual homogeneity.

## Visual homogeneity is independent of experimental context

In the above analyses, visual homogeneity was calculated for each display as its distance from an optimum center in perceptual space. This raises the possibility that visual homogeneity could be modified depending on experimental context since it could depend on the set of objects relative to which the visual homogeneity is computed. We performed a number of experiments to evaluate this possibility: we found that target-absent response times, which index visual homogeneity, are unaffected by a variety of experimental context manipulations (Appendix 5).

We therefore propose that visual homogeneity is an image-computable property that remains stable across tasks. Moreover, it can be empirically estimated as the reciprocal of the target-absent response times in a visual search task.

## A localized brain region encodes visual homogeneity (Experiment 2)

So far, we have found that target present/absent response times had opposite correlations with visual homogeneity (*Figure 3F–G*), suggesting that visual homogeneity is a possible decision variable for this task. Therefore, we reasoned that visual homogeneity may be localized to specific brain regions, such as in the visual or prefrontal cortices. Since the task in Experiment 2 was performed by participants inside an MRI scanner, we set out to investigate this issue by analyzing their brain activations.

We estimated brain activations in each voxel for individual target-present and target-absent search arrays (see Materials and methods). To identify the brain regions whose activations correlated with visual homogeneity, we performed a whole-brain searchlight analysis. For each voxel, we calculated the mean activity in a 3 x 3 × 3 volume centered on that voxel (averaged across voxels and participants) for each present/absent search display, and calculated its correlation with visual homogeneity predictions derived from behavior (see Materials and methods). The resulting map is shown in *Figure 4A*. Visual homogeneity was encoded in a highly localized region just anterior of the lateral occipital (LO) region, with additional weak activations in the parietal and frontal regions. To compare these trends across key visual regions, we calculated the correlation between mean activation and visual homogeneity for each region. This revealed visual homogeneity to be encoded strongly in this region VH, and only weakly in other visual regions (*Figure 4D*).

To ensure that the high match between visual homogeneity and neural activations in the VH region is not due to an artefact of voxel selection, we performed subject-level analysis (Appendix 6). We repeated the searchlight analysis for each subject and defined VH region for each subject. We find this VH region consistently anterior to the LO region in each subject. Next, we divided participants into two groups, and repeated the brain-wide searchlight analysis. Importantly, the match between mean activation and visual homogeneity remained significant even when the VH region was defined using one group of participants and the correlation was calculated using the mean activations of the other group (Appendix 6).

To confirm that neural activations in VH region are not driven by other cognitive processes linked to response time, such as attention, we performed a whole-brain searchlight analysis using response times across both target-present and target-absent searches. Proceeding as before, we calculated the correlation between mean activations to the target-present, target-absent and all displays with the respective response times. The resulting maps show that mean activations in the VH region are uncorrelated with response times overall (Appendix 6). By contrast, activations in EVC and LO are negatively correlated with response times, suggesting that faster responses are driven by higher activation of these areas. Finally, mean activation of parietal and prefrontal regions were strongly correlated with response times, consistent with their role in attentional modulation (Appendix 6).

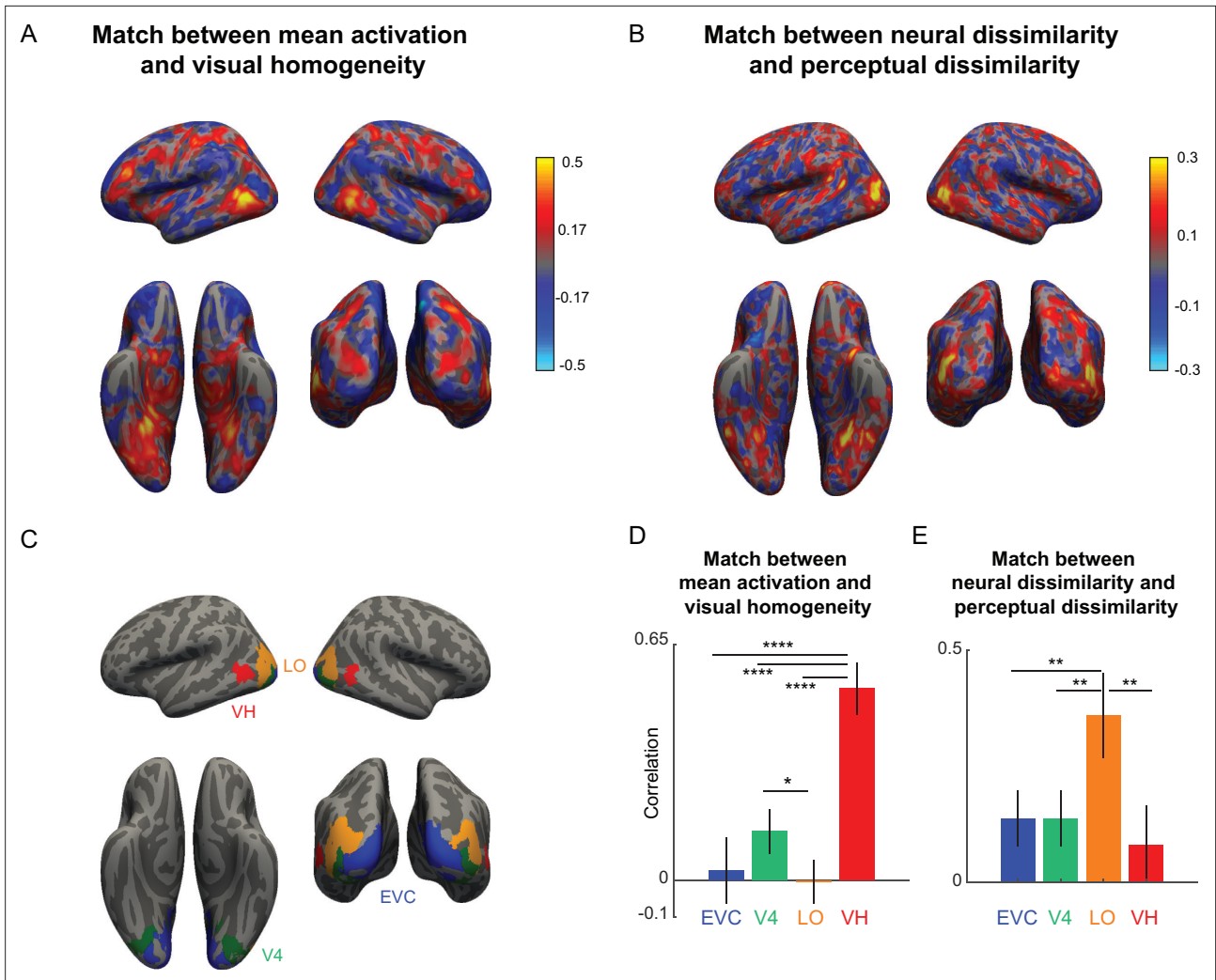

**Figure 4.** A localized brain region encodes visual homogeneity. (**A**) Searchlight map showing the correlation between mean activation in each 3 x 3 × 3 voxel neighborhood and visual homogeneity. (**B**) Searchlight map showing the correlation between neural dissimilarity in each 3 x 3 × 3 voxel neighborhood and perceptual dissimilarity measured in visual search. (**C**) Key visual regions identified using standard anatomical masks: early visual cortex (EVC), area V4, lateral occipital (LO) region. The visual homogeneity (VH) region was identified using the searchlight map in Panel A. (**D**) Correlation between the mean activation and visual homogeneity in key visual regions EVC, V4, LO, and VH. Error bars represent standard deviation of the correlation obtained using a bootstrap process, by repeatedly sampling participants with replacement for 10,000 times. Asterisks represent statistical significance, estimated by calculating the fraction of bootstrap samples in which the observed trend was violated (* is p<0.05, ** is p<0.01, **** is p<0.0001). (**E**) Correlation between neural dissimilarity in key visual regions with perceptual dissimilarity. Error bars represent the standard deviation of correlation obtained using a bootstrap process, by repeatedly sampling participants with replacement 10,000 times. Asterisks represent statistical significance, estimated by calculating the fraction of bootstrap samples in which the observed trend was violated (** is p<0.001).

## Object representations in LO match with visual search dissimilarities

To investigate the neural space on which visual homogeneity is being computed, we performed a dissimilarity analysis. Since target-absent displays contain multiple instances of a single item, we took the neural response to target-absent displays as a proxy for the response to single items. For each pair of objects, we took the neural activations in a 3 x 3 × 3 neighborhood centered around a given voxel and calculated the Euclidean distance between the two 27-dimensional response vectors (averaged across participants). In this manner, we calculated the neural dissimilarity for all $^{32}C_2=496$ pairs of objects used in the experiment, and calculated the correlation between the neural dissimilarity in each local neighborhood and the perceptual dissimilarities for the same objects measured using oddball search in Experiment 1. The resulting map is shown in *Figure 4B*. It can be seen that perceptual dissimilarities from visual search are best correlated in the lateral occipital region, consistent with

previous studies (*Figure 4E*). To compare these trends across key visual regions, we performed this analysis for early visual cortex (EVC), area V4, LO and for the newly identified region VH (average MNI coordinates (x, y, z): (−48,−59, −6) with 111 voxels in the left hemisphere; (49, -56, -7) with 60 voxels in the right hemisphere). Perceptual dissimilarities matched best with neural dissimilarities in LO compared to the other visual regions (*Figure 4E*). We conclude that neural representations in LO match with perceptual space. This is concordant with many previous studies (*Haushofer et al., 2008*; *Kriegeskorte et al., 2008*; *Agrawal et al., 2020*; *Storrs et al., 2021*; *Ayzenberg et al., 2022*).

## Equal weights for target and distractor in target-present array responses

In the preceding sections, visual homogeneity was calculated using behavioural experiments assuming a neural representation that gives equal weights to the target and distractor. We tested this assumption experimentally by asking whether neural responses to target-present displays can be predicted using the response to the target and distractor items separately. The resulting maps revealed that target-present arrays were accurately predicted as a linear sum of the constituent items, with roughly equal weights for the target and distractor (Appendix 6).

## Visual homogeneity predicts symmetry perception (Experiments 3-4)

The preceding sections show that visual homogeneity predicts target present/absent responses as well same/different responses. We have proposed that visual homogeneity can be used to solve any task that involves discriminating homogeneous and heterogeneous displays. In Experiments 3 and 4, we extend the generality of these findings to an entirely different task, namely symmetry perception. Here, asymmetric objects are akin to heterogeneous displays whereas symmetric objects are like homogeneous displays.

In Experiment 3, we measured perceptual dissimilarities for a set of 64 objects (32 symmetric, 32 asymmetric objects) made from a common set of parts. On each trial, participants viewed a search array with identical items except for one oddball, and had to indicate the side (left/right) on which the oddball appeared using a key press. An example search array is shown in *Figure 5A*. Participants performed searches involving all possible $^{64}C_2$=2016 pairs of objects. Participants made highly accurate and consistent responses on this task (accuracy, mean ± sd: 98.5 ± 1.33%; correlation between average response times from even- and odd-numbered subjects: $r$=0.88, p<0.0001 across 2016 searches). As before, we took the perceptual dissimilarity between each pair of objects to be the reciprocal of the average response time for displays with either item as target and the other as distractor. To visualize the underlying object representation, we performed a multidimensional scaling analysis, which embeds objects in a multidimensional space such that their pairwise dissimilarities match the experimentally observed dissimilarities. The resulting plot for two dimensions is shown in *Figure 5B*, where nearby objects correspond to similar searches. It can be seen that symmetric objects are generally more spread apart than asymmetric objects, suggesting that visual homogeneity could discriminate between symmetric and asymmetric objects.

In Experiment 4, we tested this prediction experimentally using a symmetry detection task that was performed by participants inside an MRI scanner. On each trial, participants viewed a briefly presented object, and had to indicate whether the object was symmetric or asymmetric using a key press (*Figure 5C*). Participants made accurate and consistent responses in this task (accuracy, mean ± sd: 97.7 ± 1.7%; correlation between response times of odd- and even-numbered participants: $r$=0.47, p<0.0001).

To investigate whether visual homogeneity can be used to predict symmetry judgements, we took the embedding of all objects from Experiment 3, and asked whether there was a center in this multidimensional space such that the distance of each object to this center would be oppositely correlated with response times for symmetric and asymmetric objects (see Materials and methods). The resulting model predictions are shown in *Figure 5E–G*. As predicted, visual homogeneity was significantly larger for symmetric compared to asymmetric objects (visual homogeneity, mean ± sd: 0.60±0.24 s$^{-1}$ for asymmetric objects; 0.76±0.29 s$^{-1}$ for symmetric objects; p<0.05, rank-sum test; *Figure 5E*). For asymmetric objects, symmetry detection response times were positively correlated with visual homogeneity ($r$=0.56, p<0.001; *Figure 5F*). By contrast, for symmetric objects, response times were negatively correlated with visual homogeneity ($r$=–0.39, p<0.05; *Figure 5G*). These patterns are

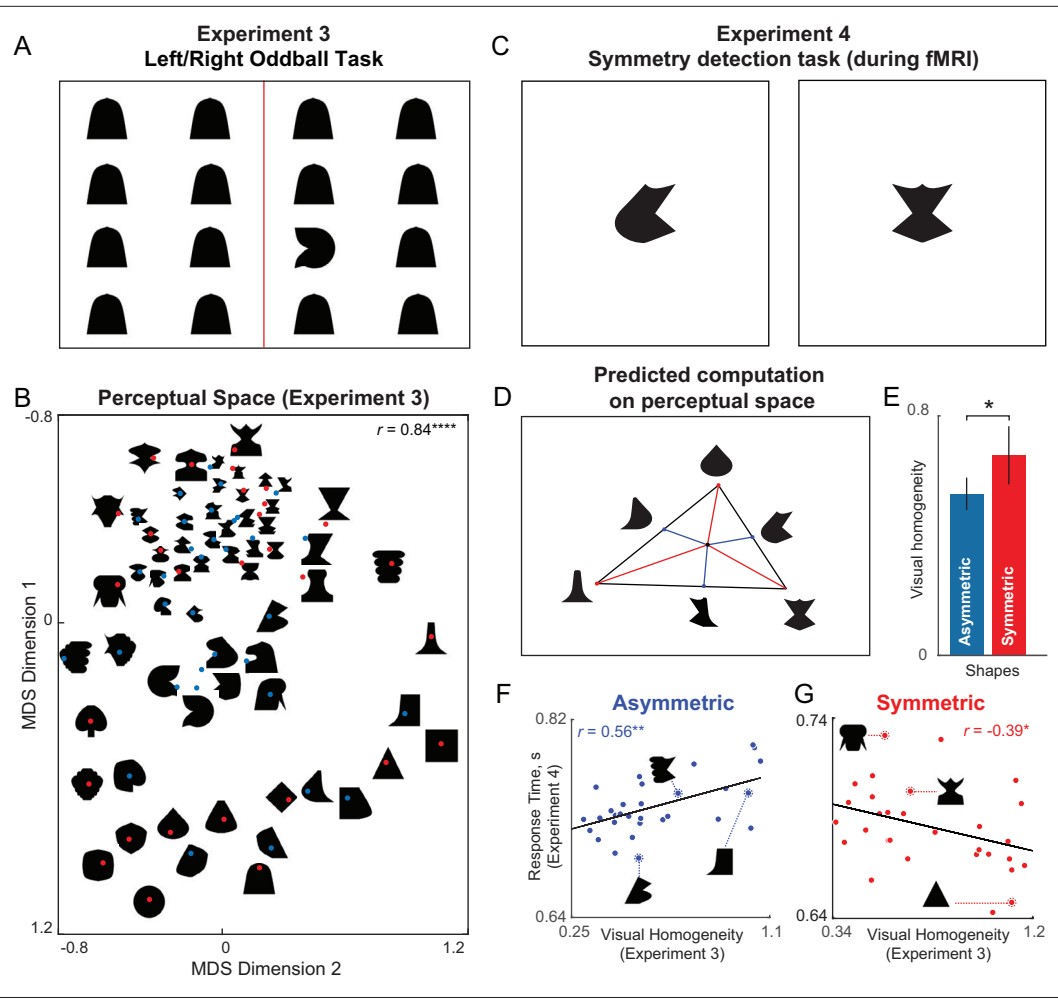

**Figure 5.** Visual homogeneity predicts symmetry perception. (**A**) Example search array in Experiment 3. Participants viewed an array containing identical items except for an oddball present either on the left or right side, and had to indicate using a key press which side the oddball appeared. The reciprocal of average search time was taken as the perceptual distance between the target and distractor items. We measured all possible pairwise distances for 64 objects (32 symmetric, 32 asymmetric) in this manner. (**B**) Perceptual space reconstructed using multidimensional scaling performed on the pairwise perceptual dissimilarities. In the resulting plot, nearby objects represent hard searches, and far away objects represent easy searches. Some images are shown at a small size due to space constraints; in the actual experiment, all objects were equated to have the same longer dimension. The correlation on the *top right* indicates the match between the distances in the 2D plot with the observed pairwise distances (**** is p<0.00005). (**C**) Two example displays from Experiment 4. Participants had to indicate whether the object is symmetric or asymmetric using a key press. (**D**) Using the perceptual representation of symmetric and asymmetric objects from Experiment 3, we reasoned that they can be distinguished using their distance to a center in perceptual space. The coordinates of this center are optimized to maximize the match to the observed symmetry detection times. (**E**) Visual homogeneity relative to the optimum center for asymmetric and symmetric objects. Error bar represents s.e.m. across images. Asterisks represent statistical significance (* is p<0.05, unpaired rank-sum test comparing visual homogeneity for 32 symmetric and 32 asymmetric objects). (**F**) Response time for asymmetric objects in Experiment 4 plotted against visual homogeneity calculated from Experiment 3. Asterisks represent statistical significance of the correlation (** is p<0.001). (**G**) Response time for symmetric objects in Experiment 4 plotted against visual homogeneity calculated from Experiment 3. Asterisks represent statistical significance of the correlation (* is p<0.05).

exactly as expected if visual homogeneity was the underlying decision variable for symmetry detection. However, we note that the range of visual homogeneity values for asymmetric and symmetric objects do overlap, suggesting that visual homogeneity contributes but does not fully determine task performance. We therefore propose that visual homogeneity provides a useful and initial first guess at symmetry in an image, which can be refined further through detailed scrutiny.

To confirm that the above model fits are not due to overfitting, we performed a leave-one-out cross validation analysis, where we left out one object at a time, and then calculated its visual homogeneity. This too yielded similar correlations ($r$=0.44 for asymmetric, $r$=–0.39 for symmetric objects, $p$<0.05 in both cases).

In sum, we conclude that visual homogeneity can predict symmetry perception. Taken together, these experiments demonstrate that the same computation (distance from a center) explains two disparate property-based visual tasks: symmetry perception and visual search.

## Visual homogeneity is encoded by the VH region during symmetry detection

If visual homogeneity is a decision variable for symmetry detection, it could be localized to specific regions in the brain. To investigate this issue, we analyzed the brain activations of participants in Experiment 4.

To investigate the neural substrates of visual homogeneity, we performed a searchlight analysis. For each voxel, we calculated the correlation between mean activations in a 3x3 × 3 voxel neighborhood and visual homogeneity. This revealed a localized region in the visual cortex as well as some parietal regions where this correlation attained a maximum (*Figure 6A*). This VH region (average MNI coordinates (x, y, z): (−57,–56, –8) with 93 voxels in the left hemisphere; (58, -50, -8) with 73 voxels in the right hemisphere) overlaps with VH region defined during visual search in Experiment 3 (for a detailed comparison, see Appendix 8).

We note that it is not straightforward to interpret the overlap between the VH regions identified in Experiments 2 and 4. The lack of overlap could be due to stimulus differences (natural images in Experiment 2 vs silhouettes in Experiment 4), visual field differences (items in the periphery in Experiment 2 vs items at the fovea in Experiment 4) and even due to different participants in the two experiments. There is evidence supporting all these possibilities: stimulus differences (*Yue et al., 2014*), visual field differences (*Kravitz et al., 2013*) as well as individual differences can all change the locus of neural activations in object-selective cortex (*Weiner and Grill-Spector, 2012*; *Glezer and Riesenhuber, 2013*). We speculate that testing the same participants on search and symmetry tasks using similar stimuli and display properties would reveal even larger overlap in the VH regions that drive behavior.

To confirm that neural activations in VH region are not driven by other cognitive processes linked to response time, such as attention, we performed a whole-brain searchlight analysis using response times across both symmetric and asymmetric objects. This revealed that mean activations in the VH region were poorly correlated with response times overall, whereas other parietal and prefrontal regions strongly correlated with response times, consistent with their role in attentional modulation and executive functions (Appendix 7).

To investigate the perceptual representation that is being used for visual homogeneity computations, we performed a neural dissimilarity analysis. For each pair of objects, we took the neural activations in a 3 x 3 × 3 neighborhood centered around a given voxel and calculated the Euclidean distance between the two 27-dimensional response vectors. In this manner, we calculated the neural dissimilarity for all $^{64}C_2$=2016 pairs of objects used in the experiment, and calculated the correlation between the neural dissimilarity in each local neighborhood and the perceptual dissimilarities for the same objects measured using oddball search in Experiment 3. The resulting map is shown in *Figure 6B*. The match between neural and perceptual dissimilarity was maximum in the lateral occipital region (*Figure 6B*).

To compare these trends for key visual regions, we repeated this analysis for anatomically defined regions of interest in the visual cortex: early visual cortex (EVC), area V4, the lateral occipital (LO) region, and the VH region defined based on the searchlight map in *Figure 5A*. These regions are depicted in *Figure 6C*. We then asked how mean activations in each of these regions is correlated with visual homogeneity. This revealed that the match with visual homogeneity is best in the VH region

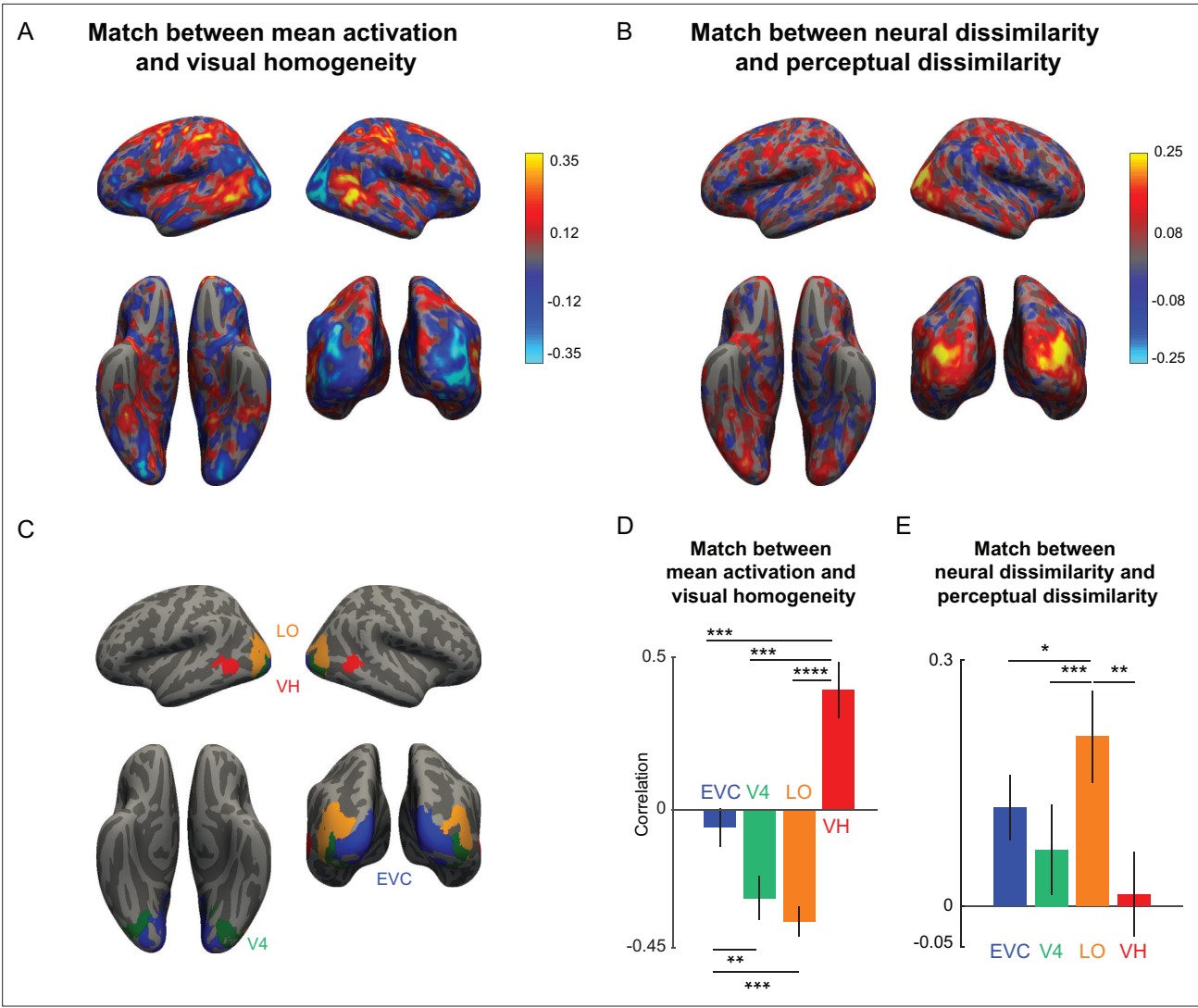

**Figure 6.** Brain region encoding visual homogeneity during symmetry detection. (**A**) Searchlight map showing the correlation between mean activation in each 3x3 × 3 voxel neighborhood and visual homogeneity. (**B**) Searchlight map showing the correlation between neural dissimilarity in each 3x3 × 3 voxel neighborhood and perceptual dissimilarity measured in visual search. (**C**) Key visual regions identified using standard anatomical masks: early visual cortex (EVC), area V4, Lateral occipital (LO) region. The visual homogeneity (VH) region was identified using searchlight map in Panel A. (**D**) Correlation between the mean activation and visual homogeneity in key visual regions EVC, V4, LO, and VH. Error bars represent standard deviation of the correlation obtained using a boostrap process, by repeatedly sampling participants with replacement for 10,000 times. Asterisks represent statistical significance, estimated by calculating the fraction of bootstrap samples in which the observed trend was violated (* is p<0.05, ** is p<0.01, **** is p<0.0001). (**E**) Correlation between neural dissimilarity in key visual regions with perceptual dissimilarity. Error bars represent the standard deviation of correlation obtained using a bootstrap process, by repeatedly sampling participants with replacement 10,000 times. Asterisks represent statistical significance, estimated by calculating the fraction of bootstrap samples in which the observed trend was violated (** is p<0.001).

compared to the other regions (*Figure 6D*). To further confirm that visual homogeneity is encoded in a localized region in the symmetry task, we repeated the searchlight analysis on two independent subgroups of participants. This revealed highly similar regions in both groups (Appendix 7).

Finally, we compared neural dissimilarities and perceptual dissimilarities in each region as before. This revealed that perceptual dissimilarities (measured from Experiment 3, during visual search) matched best with the LO region (*Figure 6E*), suggesting that object representations in LO are the basis for visual homogeneity computations in the VH region.

In sum, our results suggest that visual homogeneity is encoded by the VH region, using object representations present in the adjoining LO region.

### Target-absent responses predict symmetry detection

So far, we have shown that visual homogeneity predicts target present/absent responses in visual search as well as symmetry detection responses. These results suggest a direct empirical link between these two tasks. Specifically, since target-absent response time is inversely correlated with visual homogeneity, we can take its reciprocal as an estimate of visual homogeneity. This in turn predicts opposite correlations between symmetry detection times and reciprocal of target-absent response time: in other words, we should see a positive correlation for asymmetric objects, and a negative correlation for symmetric objects. We confirmed these predictions using additional experiments (Appendix 9). These results reconfirm that a common decision variable, visual homogeneity, drives both target present/absent and symmetry judgements.

### Visual homogeneity explains animate categorization

Since visual homogeneity is always calculated relative to a location in perceptual space, we reasoned that shifting this center towards a particular object category would make it a decision variable for object categorization. To test this prediction, we reanalyzed data from a previous study in which participants had to categorize images as belonging to three hierarchical categories: animals, dogs or Labradors (*Mohan and Arun, 2012*). By adjusting the center of the perceptual space measured using visual search, we were able to predict categorization responses for all three categories (Appendix 10). We further reasoned that, if the optimum center for animal/dog/Labrador categorization is close to the default center in perceptual space that predicts target present/absent judgements, then even the default visual homogeneity, as indexed by the reciprocal of target-absent search time, should predict categorization responses. Interestingly, this was indeed the case (Appendix 10). We conclude that, at least for the categories tested, visual homogeneity computations can serve as a decision variable for object categorization.

## Discussion

Here, we investigated three disparate visual tasks: detecting whether an oddball is present in a search array, deciding if two items are same or different, and judging whether an object is symmetric/asymmetric. Although these tasks are superficially different in the way we describe them, our key insight is that they all involve discriminating between homogeneous and heterogeneous displays. We defined a new image property computable from the underlying perceptual representation, namely visual homogeneity, that can be used to solve these tasks. Visual homogeneity predicted response times in all three tasks. Finally, visual homogeneity estimated from behavior was best correlated with mean activations in a region anterior to the lateral occipital cortex. This finding strongly suggests a specific function to this part of the high-level visual cortex. Below we discuss these findings in relation to the existing literature.

### Visual homogeneity unifies visual search, same-different and symmetry tasks

Our main finding, that a single decision variable (visual homogeneity) can be used to solve three disparate visual tasks (visual search, same/different and symmetry detection) is novel to the best of our knowledge. This finding is interesting and important because it establishes a close correspondence between all three tasks, and explains some unresolved puzzles in each of these tasks, as detailed below.

First, with regard to visual search, theoretical accounts of search are based on signal detection theory (*Verghese, 2001*; *Wolfe and Horowitz, 2017*), but define the signal only for specific target-distractor pairs. By contrast, the task of detecting whether an oddball item is present requires a more general decision rule that has not been identified. Our results suggest that visual homogeneity is the underlying decision variable at least for oddball visual search tasks. Of course, visual search is a far more complex process driven by many other factors such as distractor statistics, attentional guidance and familiarity (*Wolfe and Horowitz, 2017*). Our findings suggest that visual homogeneity could also be an additional driving factor for making decisions during visual search. Our findings also offer additional insights into visual search. Target-absent search times have always been noted to vary systematically, but lack a clear explanation in the literature. The slope of target-absent search times as

a function of set size are typically twice the slope of target present searches (*Wolfe, 1998*). However this observation is based on averaging across many target-present searches. Since there is only one unique item in a target-absent search array, any systematic variation in target-absent search must be due to an intrinsic image property. Our results resolve this puzzle by showing that this systematic variation is driven by visual homogeneity. Finally, our findings also help explain why we sometimes know a target is present without knowing its exact location – this is because the underlying decision variable, visual homogeneity, arises in high-level visual areas with relatively coarse spatial information, and its computation does not entail knowledge of the oddball's location. However, we note that visual homogeneity computations described in this study do not completely explain all the variance observed in oddball search times in our study – rather they offer a quantitative model that could explain the initial phase of target selection. We speculate that this initial phase could be shared by all forms of visual search (e.g. searching among non-identical distractors, memory-guided search, conjunction search), and these would be interesting possibilities for future work.

Second, with regards to same-different tasks, most theoretical accounts use signal detection theory but usually with reference to specific stimulus pairs (*Nickerson, 1969*; *Petrov, 2009*). It has long been observed that 'different' responses become faster with increasing target-distractor dissimilarity but this trend logically predicts that 'same' responses, which have zero difference, should be the slowest (*Nickerson, 1967*; *Nickerson, 1969*). But in fact, 'same' responses are faster than 'different' responses. This puzzle has been resolved by assuming a separate decision rule for 'same' judgements, making the overall decision process more complex (*Petrov, 2009*; *Goulet, 2020*). Our findings resolve this puzzle by identifying a novel variable, visual homogeneity, which can be used to implement a simple decision rule for making same/different responses. Our findings also explain why some images elicit faster 'same' responses than others: this is due to image-to-image differences in visual homogeneity.

Third, with regard to symmetry detection, most theoretical accounts assume that symmetry is explicitly detected using symmetry detectors along particular axes (*Wagemans, 1997*; *Bertamini and Makin, 2014*). By contrast, our findings indicate an indirect mechanism for symmetry detection that does not invoke any special symmetry computations. We show that visual homogeneity computations can easily discriminate between symmetric and asymmetric objects. This is because symmetric objects have high visual homogeneity since they have repeated parts, whereas asymmetric objects have low visual homogeneity since they have disparate parts (*Pramod and Arun, 2018*). In a recent study, symmetry detection was explained by the average distance of objects relative to other objects (*Pramod and Arun, 2018*). This finding is consistent with ours since visual homogeneity is correlated with the average distance to other objects (Appendix 1). However, there is an important distinction between these two quantities. Visual homogeneity is an intrinsic image property, whereas the average distance of an object to other objects depends on the set of other objects on which the average is being computed. Indeed, we have confirmed through additional experiments that visual homogeneity is independent of experimental context (Appendix 4). We speculate that visual homogeneity can explain many other aspects of symmetry perception, such as the relative strength of symmetries.

## Visual homogeneity in other visual tasks

Our finding that visual homogeneity explains property-based visual tasks has several important implications for visual tasks in general. First, we note that visual homogeneity can be easily extended to explain other property-based tasks such as delayed match-to-sample tasks or n-back tasks, by taking the response to the test stimulus as being averaged with the sample-related information in working memory. In such tasks, visual homogeneity will be larger for sequences with repeated compared to non-repeated stimuli, and can easily be used to solve the task. Testing these possibilities will require comparing systematic variations in response times in these tasks across images, and measurements of perceptual space for calculating visual homogeneity.

Second, we note that visual homogeneity can also be extended to explain object categorization, if one assumes that the center in perceptual space for calculating visual homogeneity can be temporarily shifted to the center of an object category. In such tasks, visual homogeneity relative to the category center will be small for objects belonging to a category and large for objects outside the category, and can be used as a decision variable to solve categorization tasks. This idea is consistent with prevalent accounts of object categorization (*Stewart and Morin, 2007*; *Ashby and Maddox, 2011*; *Mohan and Arun, 2012*). Indeed, categorization response times can be explained using perceptual distances to

category and non-category items (*Mohan and Arun, 2012*). By reanalyzing data from this study, we have found that, at least for the animate categories tested, visual homogeneity can explain categorization responses (Appendix 9). However, this remains to be tested in a more general fashion across multiple object categories.

## Neural encoding of visual homogeneity

We have found that visual homogeneity is encoded in a specific region of the brain, which we denote as region VH, in both visual search and symmetry detection tasks (*Figures 4D and 5D*). This finding is consistent with observations of norm-based encoding in IT neurons (*Leopold et al., 2006*) and in face recognition (*Valentine, 1991*; *Rhodes and Jeffery, 2006*; *Carlin and Kriegeskorte, 2017*). However, our findings are significant because they reveal a dedicated region in high-level visual cortex for solving property-based visual tasks.

We have found that the VH region is located just anterior to the lateral occipital (LO) region, where neural dissimilarities match closely with perceptual dissimilarities (*Figures 4E and 5E*). Based on this proximity, we speculate that visual homogeneity computations are based on object representations in LO. However, confirming this prediction will require fine-grained recordings of neural activity in VH and LO. An interesting possibility for future studies would be to causally perturb brain activity separately in VH or LO using magnetic or electrical stimulation, if at all possible. A simple prediction would be that perturbing LO would distort the underlying representation, whereas perturbing VH would distort the underlying decision process. We caution however that the results might not be so easily interpretable if visual homogeneity computations in VH are based on object representations in LO.

Recent observations from neural recordings in monkeys suggest that perceptual dissimilarities and visual homogeneity need not be encoded in separate regions. For instance, the overall magnitude of the population neural response of monkey inferior temporal (IT) cortex neurons was found to correlate with memorability (*Jaegle et al., 2019*). These results are consistent with encoding of visual homogeneity in these regions. However, we note that neural responses in IT cortex also predict perceptual dissimilarities (*Op de Beeck et al., 2001*; *Sripati and Olson, 2010*; *Zhivago and Arun, 2014*; *Agrawal et al., 2020*). Taken together, these findings suggest that visual homogeneity computations and the underlying perceptual representation could be interleaved within a single neural population, unlike in humans where we found separate regions. Indeed, in our study, the mean activations of the LO region were also correlated with visual homogeneity for symmetry detection (*Figure 6A*), but not for target present/absent search (*Figure 4A*). We speculate that perhaps visual homogeneity might be intermingled into the object representation in monkeys but separated into a dedicated region in humans. These are interesting possibilities for future work.

Although many previous studies have reported brain activations in the vicinity of the VH region, we are unaware of any study that has ascribed a specific function to this region. The localized activations in our study match closely with the location of the recently reported ventral stream attention module in both humans and monkeys (*Sani et al., 2021*). Previous studies have observed important differences in brain activations in this region, which can be explained using visual homogeneity, as detailed below.

First, previous studies have observed larger brain activations for animate compared to inanimate objects in high-level visual areas which have typically included the newly defined VH region reported here (*Bracci and Op de Beeck, 2015*; *Proklova et al., 2016*; *Thorat et al., 2019*). In our study, visual homogeneity, as indexed by the reciprocal of target-absent search time, is smaller for animate objects compared to inanimate objects (Appendix 9). Likewise, brain activations were weaker for animate objects compared to inanimate objects in region VH (average VH activations, mean ± sd across participants: 0.50±0.61 for animate target-absent displays, 0.64±0.59 for inanimate target-absent displays, p<0.05, sign-rank test across participants). These discrepancies could be due to differences in stimuli or task demands. Our results do however suggest that visual homogeneity may be an additional organizing factor in human ventral temporal cortex. Reconciling these observations will require controlling animate/inanimate stimuli not only for shape but also for visual homogeneity.

Second, previous studies have reported larger brain activations for symmetric objects compared to asymmetric objects in the vicinity of this region (*Sasaki et al., 2005*; *Van Meel et al., 2019*). This can be explained by our finding that symmetric objects have larger visual homogeneity (*Figure 5E*), leading to activation of the VH region (*Figure 6A*). But the increased activations in previous studies

were located in the V4 and LO regions, whereas we have found greater activations more anteriorly in the VH region. This difference could be due to the stimulus-related differences: both previous studies used dot patterns, which could appear more object-like when symmetric, leading to more widespread differences in brain activation due to other visual processes like figure-ground segmentation (*Van Meel et al., 2019*). By contrast, both symmetric and asymmetric objects in our study are equally object-like. Resolving these discrepancies will require measuring visual homogeneity as well as behavioural performance during symmetry detection for dot patterns.

Finally, our results are consistent with the greater activity observed for objects with shared features observed in ventral temporal cortex during naming tasks (*Taylor et al., 2012*; *Tyler et al., 2013*). Our study extends these observations by demonstrating an empirical measure for shared feature (target-absent times in visual search) and encoding of this empirical measure into a localized region in object selective cortex across many tasks. We speculate that visual homogeneity may at least partially explain semantic concepts such as those described in these studies.

### Relation to image memorability and saliency

We have defined a novel image property, visual homogeneity, which refers to the distance of a visual image to a central point in the underlying perceptual representation. It can be reliably estimated for each image as the inverse of the target-absent response time in a visual search task (*Figure 3*) and seems to be an intrinsic image property that is unaffected by the immediate experimental context (Appendix 4).

At the outset, the way we have defined visual homogeneity suggests that it could be related to other empirically measured quantities such as image memorability, or saliency. It has long been noted that faces that are rated as being distinctive or unusual are also easier to remember (*Murdock, 1960*; *Valentine and Bruce, 1986a*; *Valentine and Bruce, 1986b*; *Valentine, 1991*). Recent studies have elucidated this observation by showing that there are specific image properties that predict image memorability (*Bainbridge et al., 2017*; *Lukavský and Děchtěrenko, 2017*; *Rust and Mehrpour, 2020*). However, image memorability, as elegantly summarized in a recent review (*Rust and Mehrpour, 2020*), could be driven by a number of both intrinsic and extrinsic factors. Likewise, saliency is empirically measured as the relative proportion of fixations towards an image, and could be driven by top-down as well as by bottom-up factors (*Eimer, 2014*; *Peelen and Kastner, 2014*). Since visual homogeneity, image memorability and saliency are all empirically measured in different tasks, it would be interesting to compare how they are related on the same set of images.

### Conclusions

Taken together, our results show that many property-based visual tasks can be solved using visual homogeneity as a decision variable, which is localized to a specific region anterior to the lateral occipital cortex. While this does not explain all possible variations of these tasks, our study represents an important first step in terms of demonstrating a quantitative, falsifiable model and a localized neural substrate. We propose further that visual homogeneity computations might contribute to a variety of other visual tasks as well, and these would be interesting possibilities for future work.

## Materials and methods

All participants had a normal or corrected-to-normal vision and gave informed consent to an experimental protocol approved by the Institutional Human Ethics Committee of the Indian Institute of Science (IHEC # 6–15092017). Participants provided written informed consent before each experiment and were monetarily compensated.

### Experiment 1. Oddball detection for perceptual space (natural objects)

#### Participants

A total of 16 participants (8 males, 22±2.8 years) participated in this experiment.

#### Stimuli

The stimulus set comprised a set of 32 grayscale natural objects (16 animate, 16 inanimate) presented against a black background.

## Procedure

Participants performed an oddball detection task with a block of practice trials involving unrelated stimuli followed by the main task. Each trial began with a red fixation cross (diameter 0.5°) for 500ms, followed by a 4x4 search array measuring 30° x 30° for 5 s or until a response was made. The search array always contained one oddball target and 15 identical distractors, with the target appearing equally often on the left or right. A vertical red line divided the screen equally into two halves to facilitate responses. Participants were asked to respond as quickly and as accurately as possible using a key press to indicate the side of the screen containing the target ('Z' for left, M' for right). Incorrect trials were repeated later after a random number of other trials. Each participant completed 992 correct trials ($^{32}C_2$ object pairs x 2 repetitions with either image as target). The experiment was created using PsychoPy (*Peirce et al., 2019*) and ported to the online platform Pavlovia for collecting data.

Since stimulus size could vary with the monitor used by the online participants, we equated the stimulus size across participants using the ScreenScale function (https://doi.org/10.17605/OSF.IO/8FHQK). Each participant adjusted the size of a rectangle on the screen such that its size matched the physical dimensions of a credit card. All the stimuli presented were then scaled with the estimated scaling function to obtain the desired size in degrees of visual angle, assuming an average distance to screen of 60 cm.

## Data analysis

Response times faster than 0.3 s or slower than 3 s were removed from the data. This step removed only 1.25% of the data and improved the overall response time consistency, but did not qualitatively alter the results.

## Characterizing perceptual space using multidimensional scaling

To characterize the perceptual space on which present/absent decisions are made, we took the inverse of the average response times (across trials and participants) for each image pair. This inverse of response time (i.e. 1/RT) represents the dissimilarity between the target and distractor (*Arun, 2012*), indexes the underlying salience signal in visual search (*Sunder and Arun, 2016*) and combines linearly across a variety of factors (*Pramod and Arun, 2014*; *Pramod and Arun, 2016*; *Jacob and Arun, 2020*). Since there were 32 natural objects in the experiment and all possible ($^{32}C_2$=496) pairwise searches in the experiment, we obtained 496 pairwise dissimilarities overall. To calculate target-present and target-absent array responses, we embedded these objects into a multidimensional space using multidimensional scaling analysis (*mdscale* function; MATLAB 2019). This analysis finds the n-dimensional coordinates for each object such that pairwise distances between objects best matches with the experimentally observed pairwise distances. We then treated the activations of objects along each dimension as the responses of a single artificial neuron, so that the response to target-present arrays could be computed as the average of the target and distractor responses.

## Experiment 2. Target present-absent search during fMRI

### Participants

A total of 16 subjects (11 males; age, mean ± sd: 25±2.9 years) participated in this experiment. Participants with history of neurological or psychiatric disorders, or with metal implants or claustrophobia were excluded through screening questionnaires.

### Procedure

Inside the scanner, participants performed a single run of a one-back task for functional localizers (block design, object vs scrambled objects), eight runs of the present-absent search task (event-related design), and an anatomical scan. The experiment was deployed using custom MATLAB scripts written using Psychophysics Toolbox (*Brainard, 1997*).

### Functional localizer runs

Participants had to view a series of images against a black background and press a response button whenever an item was repeated. On each trial, 16 images were presented (0.8 s on, 0.2 s off), containing one repeat of an image that could occur at random. Trials were combined into blocks of

16 trials each containing either only objects or only scrambled objects. A single run of the functional localizers contained 12 such blocks (6 object blocks and 6 scrambled-object blocks). Stimuli in each block were chosen randomly from a larger pool of 80 naturalistic objects with the corresponding phase-scrambled objects (created by taking the 2D Fourier transform of each image, randomly shuffling the Fourier phase, and performing the Fourier inverse transform). This is a widely used method for functional localization of object-selective cortex. In practice, however, we observed no qualitative differences in our results upon using voxels activated during these functional localizer runs to further narrow down the voxels selected using anatomical masks. As a result, we did not use the functional localizer data, and all the analyses presented here are based on anatomical masks only.

### Visual search task

In the present-absent search task, participants reported the presence or absence of an oddball target by pressing one of two buttons using their right hand. The response buttons were fixed for a given participant and counterbalanced across participants. Each search array had eight items, measuring 1.5° along the longer dimension, arranged in a 3x3 grid, with no central element to avoid fixation biases (as shown in *Figure 3C*). The entire search array measured 6.5°, with an average inter-item spacing of 2.5°. Item positions were jittered randomly on each trial according to a uniform distribution with range ±0.2°. Each trial lasted 4 s (1 s ON time and 3 s OFF time), and participants had to respond within 4 s. Each run had 64 unique searches (32 present, 32 absent) presented in random order, using the natural objects from Experiment 1. Target-present searches were chosen randomly from all possible searches such that all 32 images appeared equally often. Target-absent searches included all 32 objects. The location of the target in the target-present searches was chosen such that all eight locations were sampled equally often. In this manner, participants performed 8 such runs of 64 trials each.

### Data acquisition

Participants viewed images projected on a screen through a mirror placed above their eyes. Functional MRI (fMRI) data were acquired using a 32-channel head coil on a 3T Skyra (Siemens, Mumbai, India) at the HealthCare Global Hospital, Bengaluru. Functional scans were performed using a T2*-weighted gradient-echo- planar imaging sequence with the following parameters: repetition time (TR)=2 s, echo time (TE)=28ms, flip angle = 79°, voxel size = $3 \times 3 \times 3$ mm$^3$, field of view = $192 \times 192$ mm$^2$, and 33 axial-oblique slices for whole-brain coverage. Anatomical scans were performed using T1-weighted images with the following parameters: TR = 2.30 s, TE = 1.99ms, flip angle = 9°, voxel size = $1 \times 1 \times 1$ mm$^3$, field of view = $256 \times 256 \times 176$ mm$^3$.

### Data preprocessing

The raw fMRI data were preprocessed using Statistical Parametric Mapping (SPM) software (Version12; Welcome Center for Human Neuroimaging; https://www.fil.ion.ucl.ac.uk/spm/software /spm12/), running on MATLAB 2019b. Raw images were realigned, slice-time corrected, co-registered to the anatomical image, segmented, and normalized to the Montreal Neurological Institute (MNI) 305 anatomical template. Repeating the key analyses with voxel activations estimated from individual subjects yielded qualitatively similar results. Smoothing was performed only on the functional localizer blocks using a Gaussian kernel with a full-width half-maximum of 5 mm. Default SPM parameters were used, and voxel size after normalization was kept at $3 \times 3 \times 3$ mm$^3$. The data were further processed using GLMdenoise (*Kay et al., 2013*). GLMdenoise improves the signal-to-noise ratio in the data by regressing out the noise estimated from task-unrelated voxels. The denoised time-series data were modeled using generalized linear modeling in SPM after removing low-frequency drift using a high-pass filter with a cutoff of 128 s. In the main experiment, the activity of each voxel was modeled using 79 regressors (64 stimuli +1 fixation +6 motion regressors +8 runs). In the localizer block, each voxel was modeled using 10 regressors (2 stimuli +1 fixation +6 motion regressors +1 run).

### ROI definitions

The regions of interest (ROI) of Early Visual Cortex (EVC) and Lateral Occipital (LO) regions were defined using anatomical masks from the SPM anatomy toolbox (*Eickhoff et al., 2005*). All brain maps were visualized on the inflated brain using Freesurfer (https://surfer.nmr.mgh.harvard.edu/fswiki/).

### Behavioral data analysis

Response times faster than 0.3 s or slower than 3 s were removed from the data. This step removed only 0.75% of the data and improved the overall response time consistency, but did not qualitatively alter the results.

## Model fitting for visual homogeneity

We took the multi-dimensional embedding returned by the perceptual space experiment (Experiment 1) in 5 dimensions as the responses of 5 artificial neurons to the entire set of objects. For each target-present array, we calculated the neural response as the average of the responses elicited by these 5 neurons to the target and distractor items. Likewise, for target-absent search arrays, the neural response was simply the response elicited by these 5 neurons to the distractor item in the search array. To estimate the visual homogeneity of the target-present and target-absent search arrays, we calculated the distance of each of these arrays from a single point in the multidimensional representation. We then calculated the correlation between the visual homogeneity calculated relative to this point and the response times for the target-present and target-absent search arrays. The 5 coordinates of this center point was adjusted using constrained nonlinear optimization to maximize the difference between correlations with the target-present and target-absent response times, respectively. This optimum center remained stable across many random starting points, and our results were qualitatively similar upon varying the number of embedding dimensions.

Additionally, we performed a leave-one-out cross-validation analysis to validate the number of dimensions or neurons used for the multidimensional scaling analysis in the visual homogeneity model fits. For each choice of number of dimensions, we estimated the optimal centre for visual homogeneity calculations while leaving out all searches involving a single image. We then calculated the visual homogeneity for all the target-present and target-absent searches involving the left-out image. Compiling these predictions by leaving out all images by turn results in a leave-one-out predicted visual homogeneity, which we correlated with the target-present and target-absent response times. We found that the absolute sum of the correlations between visual homogeneity and present/absent reaction times increased monotonically from 1 to 5 neurons, remained at a steady level from 5 to 9 neurons and decreased beyond 9 neurons. Furthermore, the visual homogeneity using the optimal center is highly correlated for 5–9 neurons. We therefore selected 5 neurons or dimensions for reporting visual homogeneity computations.

## Searchlight maps for mean activation (Figure 4A, Figure 6A)

To characterize the brain regions that encode visual homogeneity, we performed a whole-brain searchlight analysis. For each voxel, we took the voxels in a 3x3 × 3 neighborhood and calculated the mean activations across these voxels across all participants. To avoid overall activation level differences between target-present and target-absent searches, we z-scored the mean activations separately across target-present and target-absent searches. Similarly, we calculated the visual homogeneity model predictions from behaviour, and z-scored the visual homogeneity values for target-present and target-absent searches separately. We then calculated the correlation between the normalized mean activations and the normalized visual homogeneity for each voxel, and displayed this as a colormap on the inflated MNI brain template in *Figures 3A and 5A*.

Note that the z-scoring of mean activations and visual homogeneity removes any artefactual correlation between mean activation and visual homogeneity arising simply due to overall level differences in mean activation or visual homogeneity itself, but does not alter the overall positive correlation between the visual homogeneity and mean activation across individual search displays.

## Searchlight maps for neural and behavioural dissimilarity (Figure 4B, Figure 6B)

To characterize the brain regions that encode perceptual dissimilarity, we performed a whole-brain searchlight analysis. For each voxel, we took the voxel activations in a 3x3 × 3 neighborhood to target-absent displays as a proxy for the neural response to the single image. For each image pair, we calculated the pair-wise Euclidean distance between the 27-dimensional voxel activations evoked by the two images, and averaged this distance across participants to get a single average distance. For 32 target-absent displays in the experiment, taking all possible pairwise distances results in $^{32}C_2$=496 pairwise distances. Similarly, we obtained the same 496 pairwise perceptual dissimilarities between these items from the oddball detection task (Experiment 1). We then calculated the correlation between the mean neural dissimilarities at each voxel with perceptual dissimilarities, and displayed this as a colormap on the flattened MNI brain template in *Figures 3B and 5B*.

## Experiment 3. Oddball detection for perceptual space (Symmetric/ Asymmetric objects)

### Participants

A total of 15 participants (11 males, 22.8±4.3 years) participated in this experiment.

### Paradigm

Participants performed an oddball visual search task. Participants completed 4032 correct trials ($^{64}C_2$ shape pairs x 2 repetitions) as two sessions in 2 days. We used a total of 64 baton shapes (32 symmetric and 32 asymmetric), and all shapes were presented against a black background. We created 32 unique parts with the vertical line as part of the contour. We created 32 symmetric by joining the part and its mirror-filled version, and 32 asymmetric objects were created by randomly pairing the left part and mirror flipped version of another left part. All parts were occurring equally likely. All other task details are the same as Experiment 1.

## Experiment 4. Symmetry judgment task (fMRI and behavior)

### Participants

A total of 15 subjects participated in this study. Participants had normal or corrected to normal vision. Participants had no history of neurological or psychiatric impairment. We excluded participants with metal implants or claustrophobia from the study.

### Paradigm

Inside the scanner, participants performed two runs of one-back identity task (functional localizer), eight runs of symmetry judgment task (event-related design), and one anatomical scan. We excluded the data from one participant due to poor accuracy and long response times.

### Symmetry task

On each trial, participants had to report whether a briefly presented object was symmetric or not using a keypress. Objects measured 4° and were presented against a black background. Response keys were counterbalanced across participants. Each trial lasted 4 s, with the object displayed for 200ms followed by a blank period of 3800ms. Participants could respond at any time following appearance of the object, up to a time out of 4 s after which the next trial began. Each run had 64 unique conditions (32 symmetric and 32 asymmetric).

### 1-back task for functional localizers

Stimuli were presented as blocks, and participants reported repeated stimuli with a keypress. Each run had blocks of either silhouettes (asymmetric/symmetric), dot patterns (asymmetric/symmetric), combination of baton and dot patterns (asymmetric/symmetric) and natural objects (intact/scrambled).

## Data analysis

### Noise removal in RT

Very fast (<100ms) reaction times were removed. We also discarded all reaction times to an object if participant's accuracy was less than 80%. This process removed 3.6% of RT data.

### Model fitting for visual homogeneity

We proceeded as before by embedding the oddball detection response times into multidimensional space with three dimensions. For each image, the visual homogeneity was defined as its distance from an optimum center. We then calculated the correlation between the visual homogeneity calculated relative to this optimum center and the response times for the target-present and target-absent search arrays separately. This optimum center was estimated using a constrained nonlinear optimization to maximize the difference between the correlations for asymmetric object response times and symmetric object response times. Other details were the same as in Experiment-2.

## Acknowledgements

We thank Divya Gulati for help with data collection of Experiments S4 & S5. This research was supported by the DBT/Wellcome Trust India Alliance Senior Fellowship awarded to SPA (Grant# IA/S/17/1/503081). GJ was supported by a Senior Research Fellowship from MHRD, Government of India.

## Additional information

### Competing interests

SP Arun: Reviewing editor, eLife. The other authors declare that no competing interests exist.

### Funding

| Funder | Grant reference number | Author |
|---|---|---|
| Wellcome Trust DBT India Alliance | IA/S/17/1/503081 | SP Arun |
| MHRD | Senior Research Fellowship | Georgin Jacob |

The funders had no role in study design, data collection and interpretation, or the decision to submit the work for publication. For the purpose of Open Access, the authors have applied a CC BY public copyright license to any Author Accepted Manuscript version arising from this submission.

### Author contributions

Georgin Jacob, Conceptualization, Resources, Data curation, Software, Formal analysis, Validation, Investigation, Visualization, Methodology, Writing – original draft, Project administration, Writing – review and editing; RT Pramod, Conceptualization, Resources, Data curation, Software, Formal analysis, Validation, Investigation, Visualization, Methodology, Writing – review and editing; SP Arun, Conceptualization, Resources, Data curation, Software, Formal analysis, Supervision, Funding acquisition, Validation, Investigation, Visualization, Methodology, Writing – original draft, Project administration, Writing – review and editing

### Author ORCIDs

Georgin Jacob ● https://orcid.org/0000-0001-8262-0155
RT Pramod ● https://orcid.org/0000-0002-5933-7893
SP Arun ● https://orcid.org/0000-0001-9602-5066

### Ethics

Human subjects: All participants gave informed consent to an experimental protocol approved by the Institutional Human Ethics Committee of the Indian Institute of Science (IHEC # 6-15092017).

Reviewer #1 (Public review): https://doi.org/10.7554/eLife.93033.4.sa1
Reviewer #2 (Public review): https://doi.org/10.7554/eLife.93033.4.sa2
Reviewer #3 (Public review): https://doi.org/10.7554/eLife.93033.4.sa3
Author response https://doi.org/10.7554/eLife.93033.4.sa4

## Additional files

### Supplementary files
MDAR checklist

### Data availability
All data and code required to reproduce the results are publicly available at https://osf.io/cvzxt/.

The following dataset was generated:

| Author(s) | Year | Dataset title | Dataset URL | Database and Identifier |
|---|---|---|---|---|
| Jacob G, Pramod RT, Arun SP | 2024 | Jacob2025 Visual homogeneity | https://doi.org/10.17605/OSF.IO/CVZXT | Open Science Framework, 10.17605/OSF.IO/CVZXT |

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

# Appendix 1

## Visual homogeneity in deep networks explains search

### Introduction

In the main text, we reconstructed the underlying object representation using oddball search in Experiment 1, and used this representation to compute visual homogeneity, and predict target present/absent searches in Expeirment 2. At first glance, this result could appear circular since we are using search data from Experiment 1 to predict search data in Experiment 2. While this is highly unlikely since we are using search dissimilarities in Experiment 1 only to reconstruct the underlying object representations, we nonetheless reasoned that this concern can also be addressed by computing visual homogeneity on any brain-like object representation.

To this end, we repeated the visual homogeneity predictions of Experiment 2 using object representations derived from deep networks whose representations are known to be brain-like.

### Methods

We used a ResNet-50 architecture trained for object classification on the ImageNet dataset. First, we identified the layer that best matches the human representation by comparing previously observed perceptual distances (*Pramod and Arun, 2020*) with distances predicted by the ResNet-50 model. We then obtained the ResNet-50 representation of stimuli used in Experiment 1 specifically from layer3.5conv2. This higher dimensional stimulus representation was reduced to lower dimension using PCA. Using this reduced dimensional representation, we fitted the visual homogeneity model as before.

### Results

To identify the layer in the ResNet-50 model that best aligns with human perceptual representation, we compared observed perceptual distances from 32 experiments in a previous study with distances predicted by each layer of the ResNet-50 model. The representations from the convolutional layer (layer3.5conv2, layer 134) demonstrated the closest match to human perceptual distances (*Appendix 1—figure 1A*). Specifically, this selected layer of the ResNet-50 model predicted the observed perceptual distances for Experiment-1 ($r=0.66$, $p<0.0001$ across all $^{32}C_2=496$ object pairs).

The visual homogeneity, computed from lower-dimensional ResNet-50 model embeddings, of target-present arrays was significantly smaller than that of target-absent arrays (*Appendix 1—figure 1B*). The model's predictions are depicted in *Appendix 1—figure 1C and D*. Response times for target-present searches were positively correlated with visual homogeneity ($r=0.42$, $p<0.05$; *Appendix 1—figure 1C*), while response times for target-absent searches were negatively correlated with visual homogeneity ($r=0.56$, $p<0.0001$ across all $^{32}C_2=496$ object pairs). Additionally, The visual homogeneity computed using ResNet-50 is highly correlated with visual homogeneity computed using Experiments 1&2 ($r=0.76$, $p<0.0001$).

Hence, we demonstrate that our model can predict target-present/absent search responses based on any technique to estimate the representational space, and it is not dependent on the oddball visual search experiment.

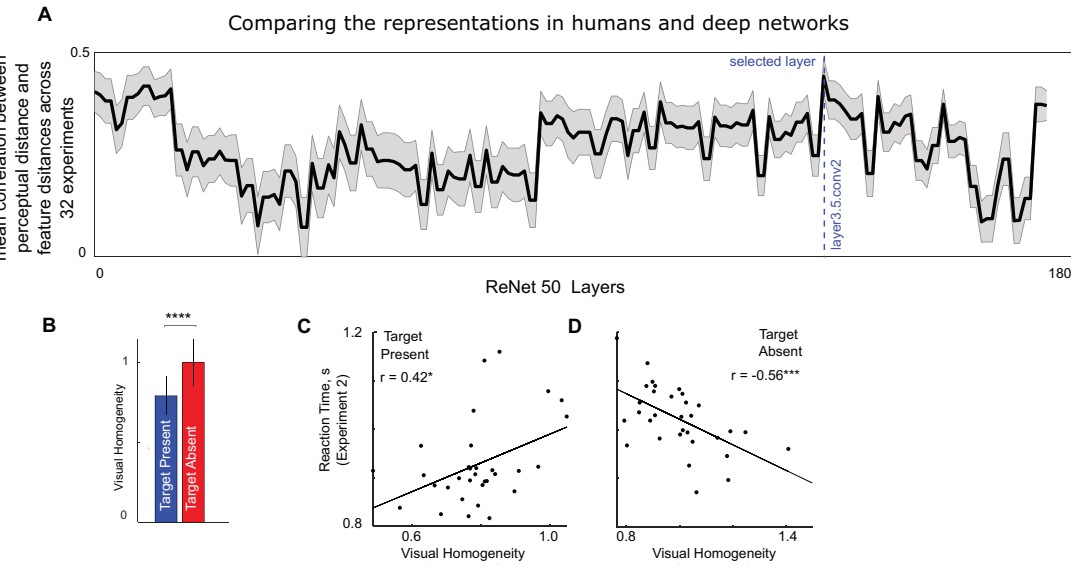

**Appendix 1—figure 1.** Visual homogeneity in deep networks predicts oddball search. (**A**) Correlation between perceptual dissimilarities and deep network dissimilarities across 32 oddball search experiments shown for each layer of ResNet-50 (median ±sem calculated across experiments). The layer with highest median correlation (*r*=0.46) is marked with a *blue dashed line*. This layer is taken for further analyses. (**B**) Predicted visual homogeneity (calculated from ResNet-50 layer 134) for target-present and target-absent searches. Error bars represent s.e.m across all displays. Asterisks represent statistical significance (**** is p<0.00005, unpaired rank-sum test comparing visual homogeneity for 32 target-absent and 32 target-present arrays). (**C**) Observed response time for target-present searches in Experiment 2 plotted against visual homogeneity calculated from ResNet-50 layer 134. Asterisks represent statistical significance of the correlation (**** is p<0.00005). Note that a single model is fit to find the optimum center that predicts the response times for both target-present and target-absent searches. (**D**) Same as (**C**) but for target-absent searches.

## Appendix 2

## Additional analysis for Experiment 1

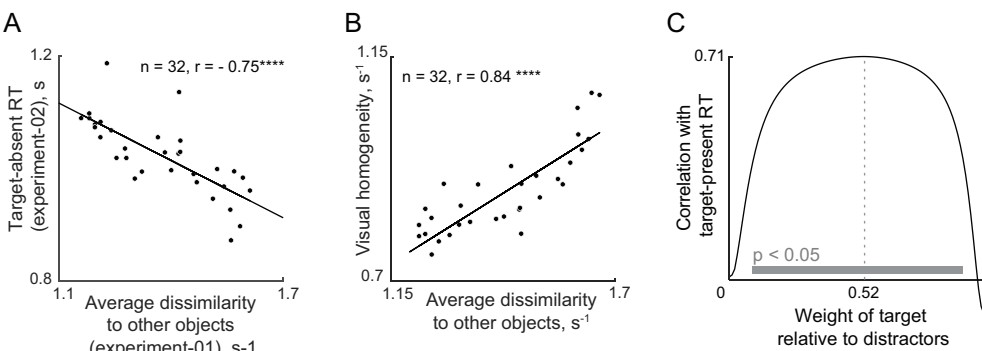

**Appendix 2—figure 1.** Additional analysis for Experiment 1. (**A**) Reaction times of target-absent searches (Experiment 2) plotted against the average dissimilarity to all other objects (Experiment 1). (**B**) Visual homogeneity for each object plotted against the average distance of each object to all other objects, suggesting that visual homogeneity is closely related to the average distance of an object to all others. (**C**) Correlation between predicted and observed target-present RT as a function of the weight of target relative to distractors in the search array. The analysis in the main text assumes that the responses to target-present arrays are an average of the target and distractor responses. To validate this assumption, we repeated the analysis by assuming taking target-present array response to be $r_{array} = w * r_{target} + (1 - w) * r_{distractor}$, where $r_{array}$ is the response to the target-present array, $r_{target}$ and $r_{distractor}$ are the responses to the target and distractor, and $w$ represents the weight of the target relative to the distractor. If w=0, it means that the target does not contribute to the overall response, and w=1 implies that the target dominates the overall response. In this plot, for each value of w, we optimized the coordinates of the center to best match the data, and plotted the correlation between predicted and observed target-present RT. It can be seen that roughly equal weighting (w=0.52) of the target and distractor yields the best fit to the data. The gray bar represent the range of weights for which the correlation is statistically significant (p<0.05).

## Appendix 3

### Generalization to other objects

#### Introduction

In the main text, using Experiments 1 and 2, we found that visual homogeneity predicts target present/absent responses, but these results were based on testing 32 natural objects. Here, we performed four additional experiments to confirm the generality of these findings. In Experiments S1 and S2, we tested 48 natural objects. In Experiments S3 and S4, we tested 49 silhouette shapes.

#### Methods

All participants had a normal or corrected-to-normal vision and gave informed consent to an experimental protocol approved by the Institutional Human Ethics Committee of the Indian Institute of Science (IHEC # 6–15092017). Participants provided written informed consent before each experiment and were monetarily compensated.

### Experiment S1: Odd ball detection for perceptual space (natural objects)

This data is Experiment 1 of a previously published study (*Mohan and Arun, 2012*), and the salient details are reproduced here. The analysis of visual homogeneity is novel and unique to this study.

#### Participants

A total of 12 participants aged 20–30 years participated in this experiment.

#### Stimuli

The stimulus set comprised a set of 48 grayscale natural objects (24 animate, 24 inanimate) presented against a black background. Images were equated for brightness. The longer dimension was presented at a visual angle of $4.8^0$.

#### Procedure

Each participant performed a visual search task, in which a 4x4 search array was presented with one oddball. Participants had to indicate whether the oddball was on the right or left of the screen using a key press ('Z' for left, 'M' for right). Distractors were varied randomly in size to prevent low-level cues from influencing search.

#### Data analysis

Response times faster than 0.3 seconds or slower than 4 seconds were removed from the data. This step removed only 0.76% of the data and improved the overall response time consistency but did not qualitatively alter the results.

### Experiment S2: Target-present absent search with natural objects

#### Participants

A total of 16 participants (9 males, 23.31±5.24 years) participated in this experiment.

#### Stimuli

The stimuli were identical to those used in Experiment S1.

#### Procedure

Participants performed a present-absent visual search task. Each trial began by presenting a red-coloured fixation cross of size 0.5 dva for 500ms, followed by the presentations of a circular search array measuring 13°x13° for 10 s or until a response, whichever was sooner. The search array contained eight elements, of which one could be different from the others. Participants identified and reported the presence of the target with a keypress ('A' for target absence and 'P' for target presence). Participants were instructed to respond as quickly and as accurately as possible. Incorrect trials were randomly repeated later.

Each participant completed 384 correct trials (absent trials: 48 shapes x 4 repetitions = 192, present trials: 24 image pairs x 2 (AB/BA) x 4 repetitions = 192). We used a total of 48 natural images from animate and inanimate categories. All images were presented against a black background. The main experiment block explained above was preceded by an identical practice experiment block of 20 trials involving unrelated stimuli. This experiment was hosted at the Pavlovia platform (https://pavlovia.org/) using custom programs written in PyschoPy (*Peirce et al., 2019*).

## Data analysis
Image Pairs with an overall accuracy of less than 50% were removed. One target-present search was removed based on this criterion from further analysis.

## Experiment S3: Odd ball detection for perceptual space (Silhouettes)
This data is from Experiment 1 of a previously published study (*Pramod and Arun, 2016*), and the salient details are reproduced here. The analysis of visual homogeneity is novel and unique to this study.

## Participants
A total of 8 participants (5 females, aged 20–30 years) participated in this experiment.

## Stimuli
Each stimulus was created by joining two of the seven possible parts. The full set consisted of 49 objects formed by all possible combinations of seven parts at two locations.

## Procedure
Participants were seated approximately 60 cm from the computer monitor that was under the control of a custom program written using Psychtoolbox in Matlab. Each trial began by presenting a fixation cross for 500ms, followed by a 4x4 search array containing one oddball item among multiple identical distractors with a red vertical line down the middle. Items were jittered. Participants were asked to report the side on which the oddball target appeared as quickly and as accurately as possible using a keypress (Z for left, M for right). All images were presented against a black background.

## Data analysis
Response times faster than 0.3 s or slower than 4.5 s were removed from the data. This step removed 1.2% of the data and improved the overall response time consistency but did not qualitatively alter the results.

## Experiment S4: Target-present absent search with silhouettes
## Participants
A total of 11 participants (6 females, aged 20–30 years) participated in this experiment.

## Stimuli
Stimuli were identical to Experiment S3.

## Procedure
Participants were seated approximately 60 cm from the computer monitor that was under the control of a custom program written using Psychtoolbox in Matlab. Each trial began by presenting a fixation cross for 500ms, followed by a 4x4 search array containing one oddball item among multiple identical distractors. Items were jittered. Participants were asked to report the presence of an oddball target as quickly and as accurately as possible using keypress (Y for the presence of the target, N for the absence of the target). All images were presented against a black background.

## Data analysis
Response times faster than 0.3 s or slower than 5 s were removed from the data. This step removed 7% of the data and improved the overall response time consistency but did not qualitatively alter the results. We predicted the response times of target-present and target-absent search display

the same approach used for experiment 2 (see main text). We used the MDS embeddings from experiment S3, and the model has 7 free parameters. To avoid overfitting, we used leave-one-out cross-validation.

## Results

In Experiment S1, we characterized the perceptual space of 48 natural objects using a left/right oddball detection task as before. Participants were highly consistent in their responses (correlation between mean response times of even- and odd-numbered participants: $r=0.90$, p<0.0001 across all $^{48}C_2=1128$ object pairs). In Experiment S2, we checked if visual homogeneity predicted target present/absent responses in visual search. Participants were highly accurate and consistent in their responses (accuracy, mean ± sd: 96 ± 1.8%; correlation between mean response times of even- and odd-numbered participants: $r=0.94$, p<0.0001 across 48 target-present searches, $r=0.93$, p<0.0001 across 48 target-absent searches). We estimated the visual homogeneity using the RT data from Experiment S1&2 by fitting model with seven free parameters. The model predictions are shown in **Appendix 3—figure 1A**. As expected, the response times of present searches were positively correlated with the visual homogeneity ($r=0.44$, p<0.05),and the response times of absent searches were negatively correlated with visual homogeneity ($r=-0.72$, p<0.0001).

In Experiment S3, we characterized the perceptual space of 49 silhouette objects using a left/right oddball detection task as before. Participants were highly accurate and consistent in their responses (accuracy, mean ± sd: 99 ± 0.5%; correlation between mean response times of even- and odd-numbered participants: $r=0.80$, p<0.0001 across all $^{49}C_2=1176$ object pairs). In Experiment S4, we checked if visual homogenity predicted target present/absent responses in visual search. Participants were highly accurate and consistent in their responses (accuracy, mean ± sd: 92.74 ± 4.27%; correlation between mean response times of even- and odd-numbered participants: $r=0.80$, p<0.0001 across 48 target-present searches, $r=0.67$, p<0.0001 across 49 target-absent searches). We estimated the visual homogeneity using the RT data from the Experiment S3 and 4 by fitting a model with seven free parameters. The model predictions are shown in **Appendix 3—figure 1B**. As expected, the response times of present searches were positively correlated with the visual homogeneity ($r=0.70$, p<0.0001), and the response times of the absent searches were negatively correlated with visual homogeneity ($r=-0.81$, p<0.0001).

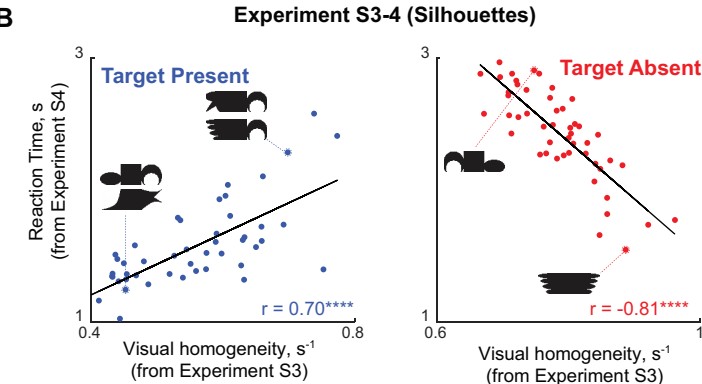

**Appendix 3—figure 1.** Generalization to other objects. (**A**) Response time for target present/absent responses in Experiment S2 (involving a larger set of natural objects) plotted against visual homogeneity calculated from Experiment S1. (**B**) Response time for target present/absent responses in Experiment S4 (involving silhouettes) plotted against visual homogeneity calculated from Experiment S3.

# Appendix 4

## Visual homogeneity predicts same/different responses

### Introduction

In the main text, we have shown that visual homogeneity computations can be used to solve any task that requires discriminating between homogeneous and heterogeneous displays. Having shown in Experiments 1–2 that visual homogeneity predicts target present/absent responses, we wondered whether these results would generalize to other tasks. Here, we investigated the same/different task, in which participants have to view two items and indicate whether they are same or different using a key press.

Although the exact instructions of the target present/absent visual search task and the same/different task are quite different, we predicted that 'same' responses would be related to the target-absent response, while the 'different' responses would be related to the target-present response. This correspondence is interesting and non-trivial since it has never been made previously to the best of our knowledge.

## Experiment S5. Same/different task

In Experiment S5, we recruited participants to perform a same/different task involving the same natural objects as in Experiment 1. The 'same'trials and 'different' trials were exactly matched to the target-absent and target-present conditions in the two experiments, to enable direct comparisons.

### Methods

### Participants

A total of 16 participants (8 males aged 23.5±3.5 years) participated in this experiment.

### Stimuli

The stimulus set comprised a set of 32 grayscale natural objects (16 animate, 16 inanimate) presented against a black background. Same as in the main experiments 1 and 2.

### Procedure

Participants performed a same-different task with a block of practice trials involving unrelated stimuli followed by the main task. Each trial began with a fixation cross (width 0.6°) for 500ms, followed by two images of size 3.5 dva presented for 500ms on a black background. The images were presented on either side of the fixation cross along the x-axis 3 dva away from the centre. A random position jitter of 0.5 dva was added in both directions. Participants were asked to identify if two images were the same or different and respond with a keypress ('Z' for same, 'M' for different) as quickly and accurately as possible. Participants could give responses up to 3.5 s. Each participant completed 256 same trials (32 objects x 8 repeats) and 256 different trials (32 randomly selected image pairs x 8 repeats). The experiment was created using PsychoPy (*Peirce et al., 2019*) and ported to the online Pavlovia platform for collecting data. Each participant adjusted a credit card image presented on the screen with keypresses to match their credit card size, ensuring identical image presentation across participants.

### Results

In Experiment S5, participants were highly accurate and consistent in their responses (accuracy, mean ± sd: 91.4 ± 3.4%; correlation between mean reaction times of even- and odd-numbered participants: $r=0.50$, $p<0.005$ across 32 same trials, $r=0.71$, $p<0.0005$ across 32 diff trials). The absent search response times from Experiment 2 predicted the same response times of Experiment S5 (correlation between response times: $r=0.67$, $p<0.0005$; *Appendix 4—figure 1E*). The present search response time from Experiment 2 predicted the different response times of Experiment S5 (correlation between response times: $r=0.75$, $p<0.0005$; *Appendix 4—figure 1F*). The response times of one task predict the response times of another on the same set of image pairs is expected when identical computations can solve both tasks.

We conclude that visual homogeneity computations can explain same-different judgements.

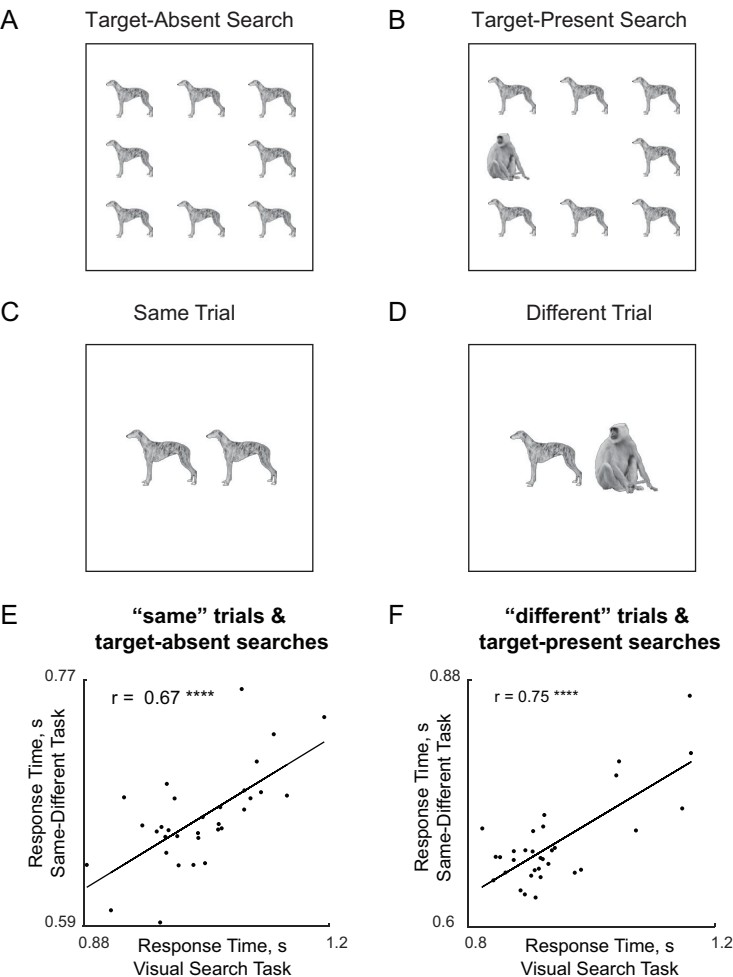

**Appendix 4—figure 1.** Target absent/present responses predict same/different responses. (**A**) Example target-absent trial from the visual search task (Experiment 2). (**B**) Example target-present trial from the visual search task (Experiment 2). (**C**) Example 'same' trial from the same-different task (Experiment S5), matched exactly to the target-absent trial in panel A. (**D**) Example 'different' trial from the same-different task (Experiment S5), matched exactly to the target-present trial in panel B. (**E**) Response time on 'same' trials in Experiment S5 plotted against response time for the corresponding target-absent trials from Experiment 1. (**F**) Response time on 'different' trials in Experiment S5 plotted against response time for the corresponding target-present trials from Experiment 1.

# Appendix 5

## Visual homogeneity is unaffected by context

In Appendix 1, we have shown that visual homogeneity for each image is proportional to its distance to all other images, and that it is negatively correlated with target-absent response times. This raises the question of whether target-absent response times are affected by the other objects used in the experiment. We addressed this question using two experiments.

## Experiment S6. Visual search across diverse category context

We first investigated the possibility that target present/absent response times could vary depending on the experimental context of other searches. To this end, participants had to perform target present/absent task in three separate blocks: searches involving animate objects, searches involving silhouette objects and a mixed block containing both animate and silhouette objects.

### Methods

### Participants

A total of 18 participants (12 males, aged 23.5±3.2 years) participated in this experiment.

### Stimuli

The stimulus set comprised a set of 16 grayscale natural objects (animates) and 16 silhouettes presented against a black background.

### Procedure

This experiment constituted three blocks: Animal block, Silhouettes block, and mixed block. All participants performed trials from all the blocks, and the block order was counterbalanced across participants. Each participant performed 512 correct trials (128 from animate block, 128 from silhouette block and 256 from mixed block). In the animate block, there were 32 uniques conditions (16 present searches and 16 absent searches) repeated 4 times created from 16 animate images. A similar design was repeated for the silhouette block on 16 silhouette images. The mixed block was created by combining the animate blocks and silhouette blocks. The trial presentation order was random within a block. Participants were seated approximately 60 cm from the computer monitor that was under the control of a custom program written using Psychtoolbox in Matlab. Each trial started by presenting a fixation cross for 500ms, followed by a 4x4 search array containing one oddball item among multiple identical distractors. Items were randomly jittered. Participants were asked to report the presence of an oddball target as quickly and as accurately as possible using keypress (Y for the presence of the target, N for the absence of the target). All images were presented against a black background.

### Results

Participants were highly accurate and consistent in their responses (accuracy, mean ±bsd: 96.9 ± 1.8% for Animate block, 97.9 ± 1.3% for mixed block, 97.7±1.8 for silhouette block; correlation between mean reaction times of even- and odd-numbered participants: $r=0.93$, p<0.0005 for Animate block, $r=0.94$, p<0.0005 for Mixed block, $r=0.95$, p<0.0005 for Silhouette block). The response time to searches in the animate block shows a striking correlation with the response times to the corresponding searches in the mixed block (correlation between reaction times: $r<0.96$, p<0.0001 for present searches, $r<0.95$, p<0.0001 for absent searches). Similarly, the response time to searches in the silhouette block shows a striking correlation with the response times to the corresponding searches in the mixed block (correlation between reaction times: $r<0.96$, p<0.0001 for present searches, $r<0.91$, p<0.0001 for absent searches). Even when the animal and silhouette images have completely different image statistics, there was no significant difference between reaction times in the interleaved and separate blocks (*Appendix 5—figure 1A and B*). Hence, we conclude that context has no effect on visual homogeneity.

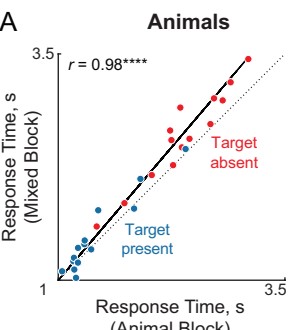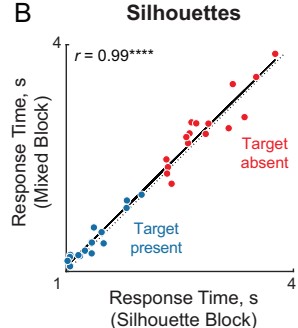

**Appendix 5—figure 1.** Target-absent responses are unaffected by mixing disparate searches. (**A**) Response times in the Mixed Block plotted against the corresponding response times in the Animal Block for present searches (*blue*), and absent searches (*red*). (**B**) Response time in Mixed Block plotted against the corresponding response times in the Silhouette-only Block, with conventions as in panel A.

## Experiment S7. Objects with uncorrelated dissimilarity

Although the results of Experiment S6 show that target-absent searches are unaffected by whether they are surrounded by similar or mixed object categories, it is still possible that the average distance of each object to other objects was unaffected by mixing with other category objects but could be influenced by the experimental context of other objects in the same category.

To this end, we constructed two sets of 29 shapes (20 unique to each set, 9 common to both sets) (*Appendix 5—figure 2A*). We chose the two sets so that the predicted visual homogeneity (i.e. average distance to other objects in that set) relative to each set is uncorrelated (*Appendix 5—figure 2B*). We then performed a present/absent visual search block in which the 9 common objects are viewed in the context of the other objects in Set 1 or Set 2 in separate blocks.

If visual homogeneity depends on the objects being seen in a given block, we predicted that the target absent response times of the common objects will be uncorrelated. If on the other hand, the visual homogeneity of an object is unaffected by experimental context, we predicted that the absent search response times will be correlated. We tested this prediction using a target present/absent search experiment.

### Methods

Participants. A total of 12 participants (7 males, aged 23.5±1.16 years) participated in this experiment.

Stimuli. This stimulus set comprised of 49 silhouette shapes. Sets 1 and 2 were constructed by selecting 29 of 49 shapes where 9 shapes were common across sets in such a manner that the average dissimilarity of the 9 common shapes relative to the other objects in the two sets were uncorrelated (*Appendix 5—figure 1A-B*).

Procedure. This experiment comprised two blocks. All participants completed both blocks, and the block order was counterbalanced across participants. Each participant performed 464 correct trials (232 from block 1 and 232 from block 2). In each block there were 58 unique conditions (29 absent searches, 29 present searches) and each condition was repeated 4 times. The trial presentation order was random within the block. Other details were same as experiment S6.

Data Analysis. Response times faster than 0.3 s or slower than 5 s were removed. We used '*isoutlier*' Matlab function to remove the outliers in the data. These steps removed 6.6% of the data and improved the overall response time consistency but did not quantitatively alter the results.

### Results

Participants were highly accurate and consistent in their responses (accuracy, mean ± sd: 94.8 ± 2.6% for block 1, and 95.2 ± 3.1% for block 2; correlation between mean reaction times of even- and odd-numbered participants: $r=0.94$, $p<0.0005$ for block 1 and $r=0.95$, $p<0.0005$ for block 2).

For the common objects, we computed the average dissimilarity to other objects within the set and the average dissimilarity of the common objects were not correlated across sets ($r=0.21$, $p=0.6$; *Appendix 5—figure 2B*), which is by design. Interestingly, the target-absent responses were highly correlated across sets ($r=0.83$, $p=0.05$; *Appendix 5—figure 2C*).

We conclude that visual homogeneity is unaffected by the experimental context regardless similar or dissimilar objects.

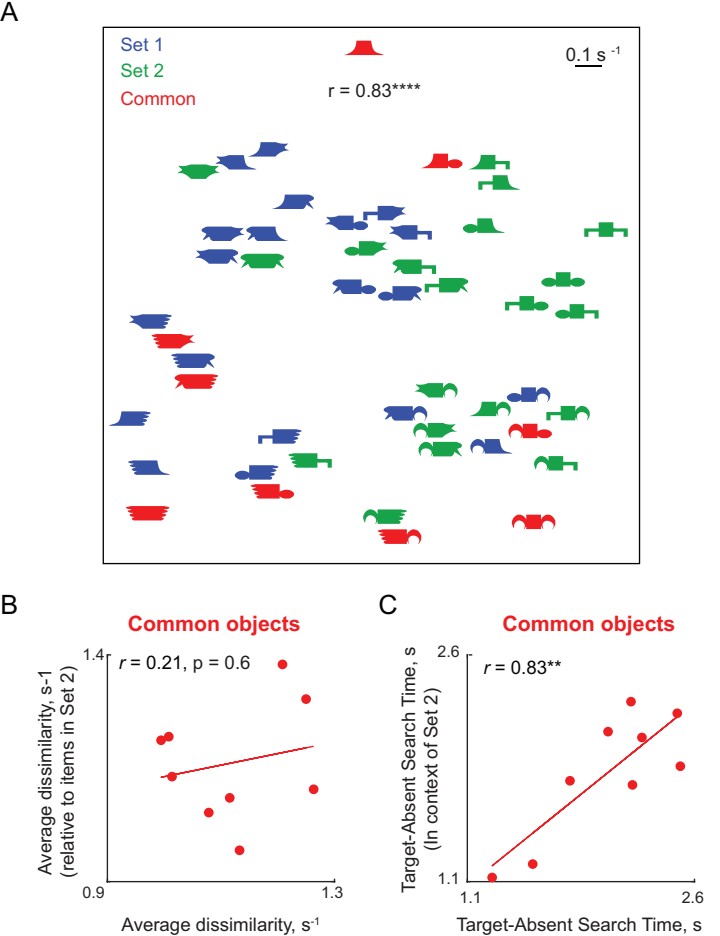

**Appendix 5—figure 2.** Target-absent responses are unaffected by disparate object context. (**A**) 2D embedding of the 49 silhouettes based in the pairwise dissimilarities (1/RT) measured using odd-ball visual search experiment (Experiment S3). Shapes are coloured according to the set to which they are grouped: *red* for shapes common to set 1 and set 2, *blue* for shapes only in set 1, and *green* for shapes only in set 2. (**B**) Average dissimilarity of the common items relative to items in Set 2 plotted against the average dissimilarity relative to items in Set 1. If visual homogeneity depends on the average distance to other objects in the immediate experimental context, then target-absent responses for the common objects should be uncorrelated when presented in a block containing Set 1 items compared to a block containing Set 2 items. (**C**) Absent search response times for the common items in Block 2 (containing Set 2 items) against the corresponding search times in Block 1 (containing Set 1 items). The strong and significant correlation indicates that target-absent search times are independent of the immediate experimental context.

# Appendix 6

## Additional analyses for experiment 2 (fMRI)

**Activation difference Target-present - target-absent**

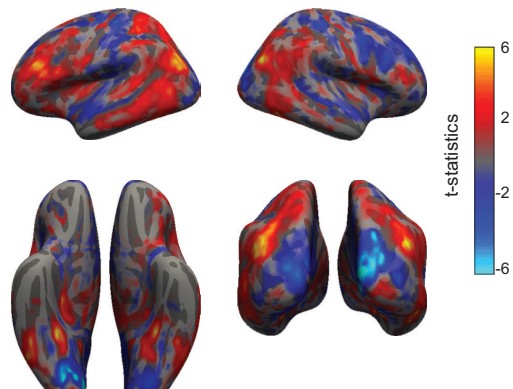

**Appendix 6—figure 1.** Brain activations for target-present and target-absent searches. Whole brain colormap of activation difference between target-present and target-absent searches. The color at each voxel represents the t-statistic computed between the participant-wise mean activations for target-present minus target-absent searches (averaged across searches of each type, and across a 3 x 3 × 3 voxel neighborhood centered around that voxel).

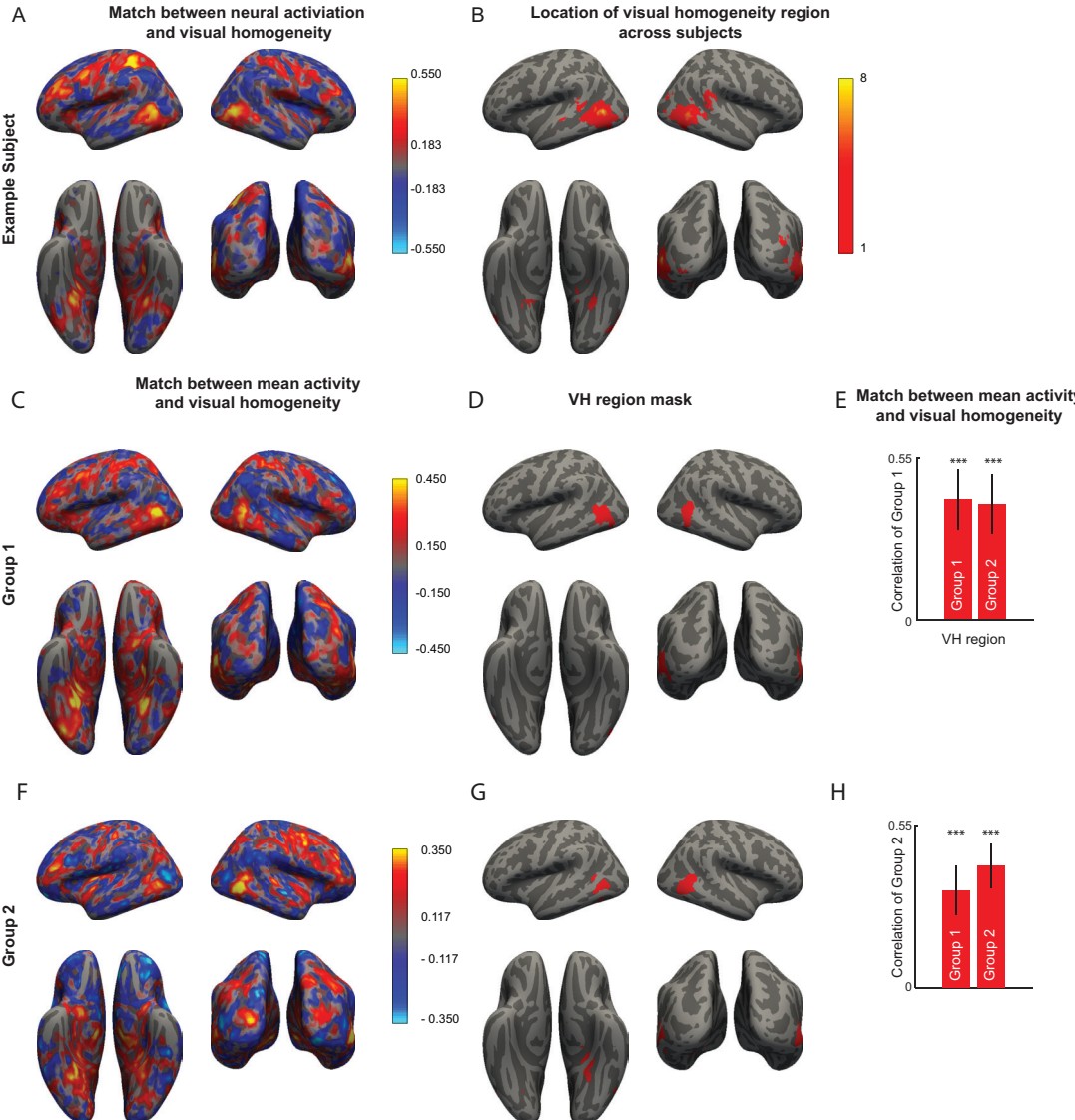

**Appendix 6—figure 2.** Robustness of VH region in target present/absent search. (**A**) Searchlight map showing the correlation between visual homogeneity and mean activation of an example subject. (**B**) Colormap representing the number of subjects for which a particular voxel belonged to the localized VH region. (**C**) Colormap of the correlation between visual homogeneity and mean activation across eight subjects in Group 1. (**D**) VH region obtained by thresholding the searchlight map in panel C. (**E**) Correlation between visual homogeneity and mean activation for participants in Group 1 for the VH region identified from Group 1, and for the VH region identified from Group 2. Asterisks indicate statistical significance of each correlation, obtained by sampling participants with replacement 10,000 times, and calculating the fraction of times the correlation was below zero (**** is p<0.0005). The significant correlation in Group 2 for the region identified using Group 1 participants suggest that the VH region is consistently localized across subjects. (**F**) Searchlight map similar to panel C, but for participants in Group 2. (**G**) VH region obtained by thresholding the searchlight map in panel F. (**H**) Correlation between visual homogeneity and mean activation for participants in Group 2, for the VH regions identified from Group 1 and from Group 2. Asterisks indicate statistical significance of each correlation, obtained by sampling participants with replacement 1000 times, and calculating the fraction of times the correlation was below zero (***is p<0.005). The significant correlation in Group 1 for the region identified using Group 2 participants suggest that the VH region is consistently localized across subjects.

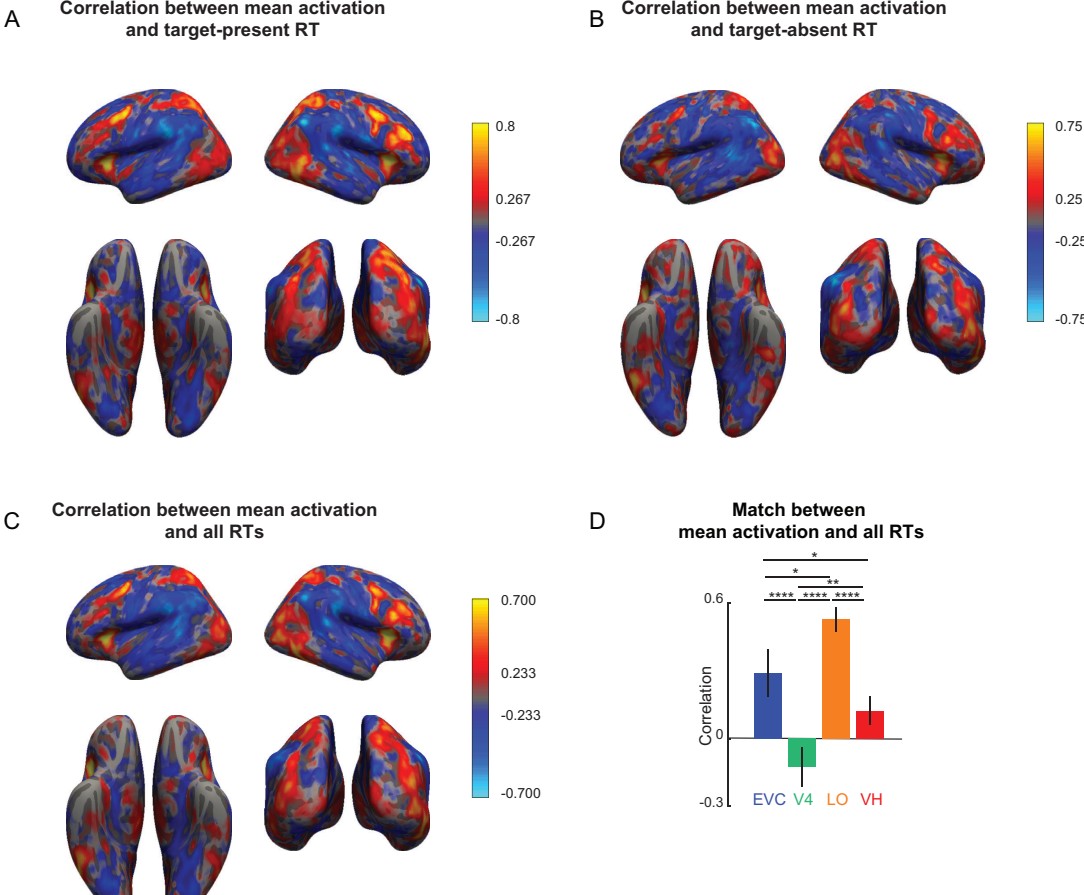

**Appendix 6—figure 3.** Searchlight maps with response times. (**A**) Colormap of correlation between mean activation and response times for target-present search arrays. (**B**) Colormap of correlation between mean activation and response times for target-absent search arrays. (**C**) Correlation between mean activation and response times for both target-present and target-absent search arrays. To prevent image-wise correlations from being confounded by overall activation level differences, we z-scored the mean activations for each voxel within a particular search type (present/absent) and then combined the mean activations. Likewise, for similar reasons, we z-scored the response times for each particular search type. (**D**) Correlation between mean activation and all response times for key visual regions.Error bars represent the standard deviation of correlation obtained using a bootstrap process by repeatedly sampling participants with replacement 10,000 times. Asterisks represent statistical significance, estimated by calculating the fraction of bootstrap samples in which the observed trend was violated (* is p<0.05, ** is p<0.01, **** is p<0.0001).

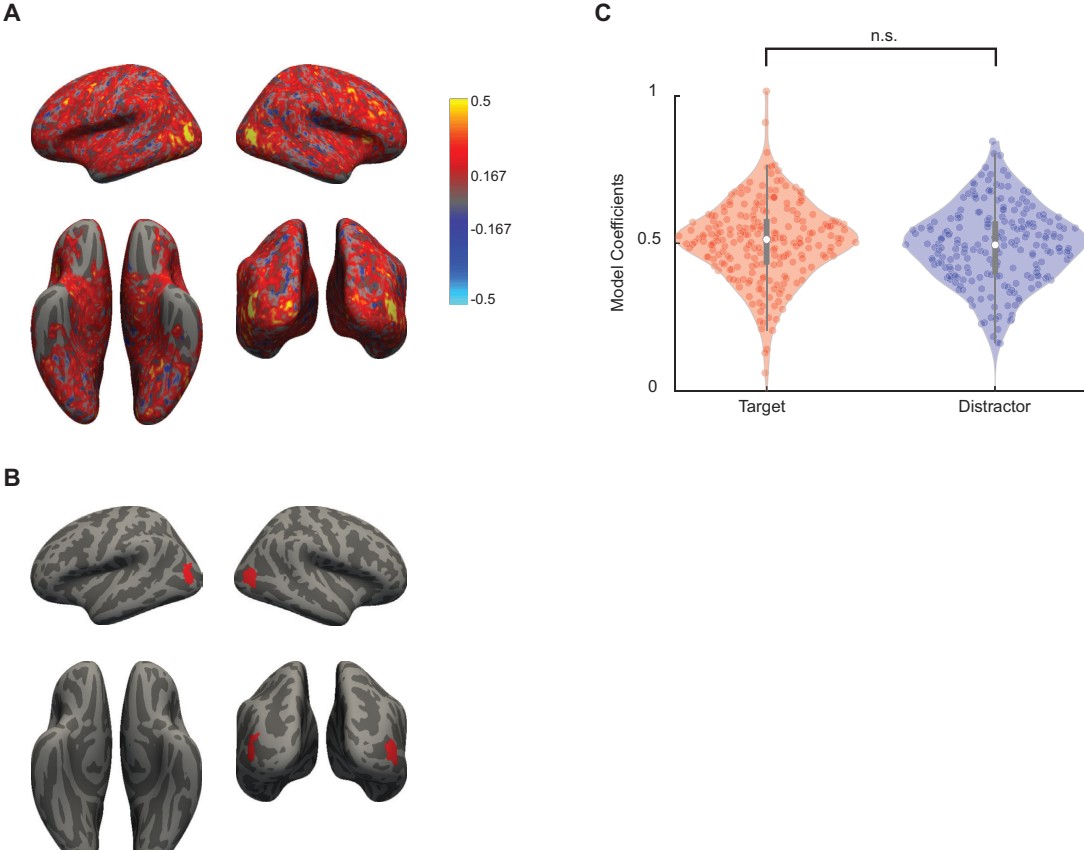

**Appendix 6—figure 4.** Relative weights of target and distractor in target-present arrays. (**A**) Colormap of correlation between observed and predicted voxel activity of the linear voxel model, in which target-present search array response is modelled as a linear combination of target and distractor activity (taken from responses to target-absent arrays). (**B**) Region showing good model prediction, obtained by thresholding the colormap in (**A**). (**C**) Target and distractor model coefficients in this region. Each point corresponds to model coefficients derived from a single voxel. Model coefficients for target and distractor are equal in weight (p=0.33, sign-rank test across weights of 222 voxels in this region).

## Appendix 7

## Additional analysis for experiment 4 (fMRI)

**Activation difference
Asymmetric - Symmetric objects**

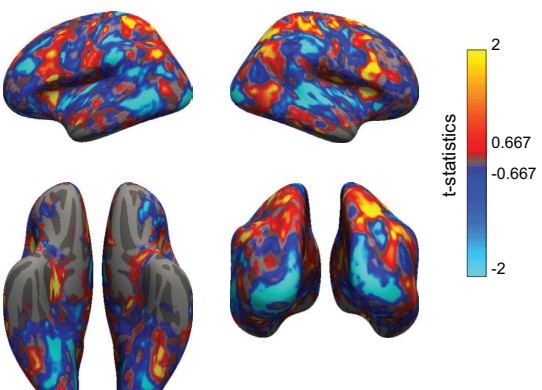

**Appendix 7—figure 1.** Brain activations for asymmetric and symmetric objects. Whole brain colormap of activation difference between asymmetric and symmetric objects during the symmetry task. The color at each voxel represents the t-statistic computed between the participant-wise mean activations for asymmetric minus symmetric objects (averaged across objects of each type, and across a 3 x 3 × 3 voxel neighborhood centered around that voxel).

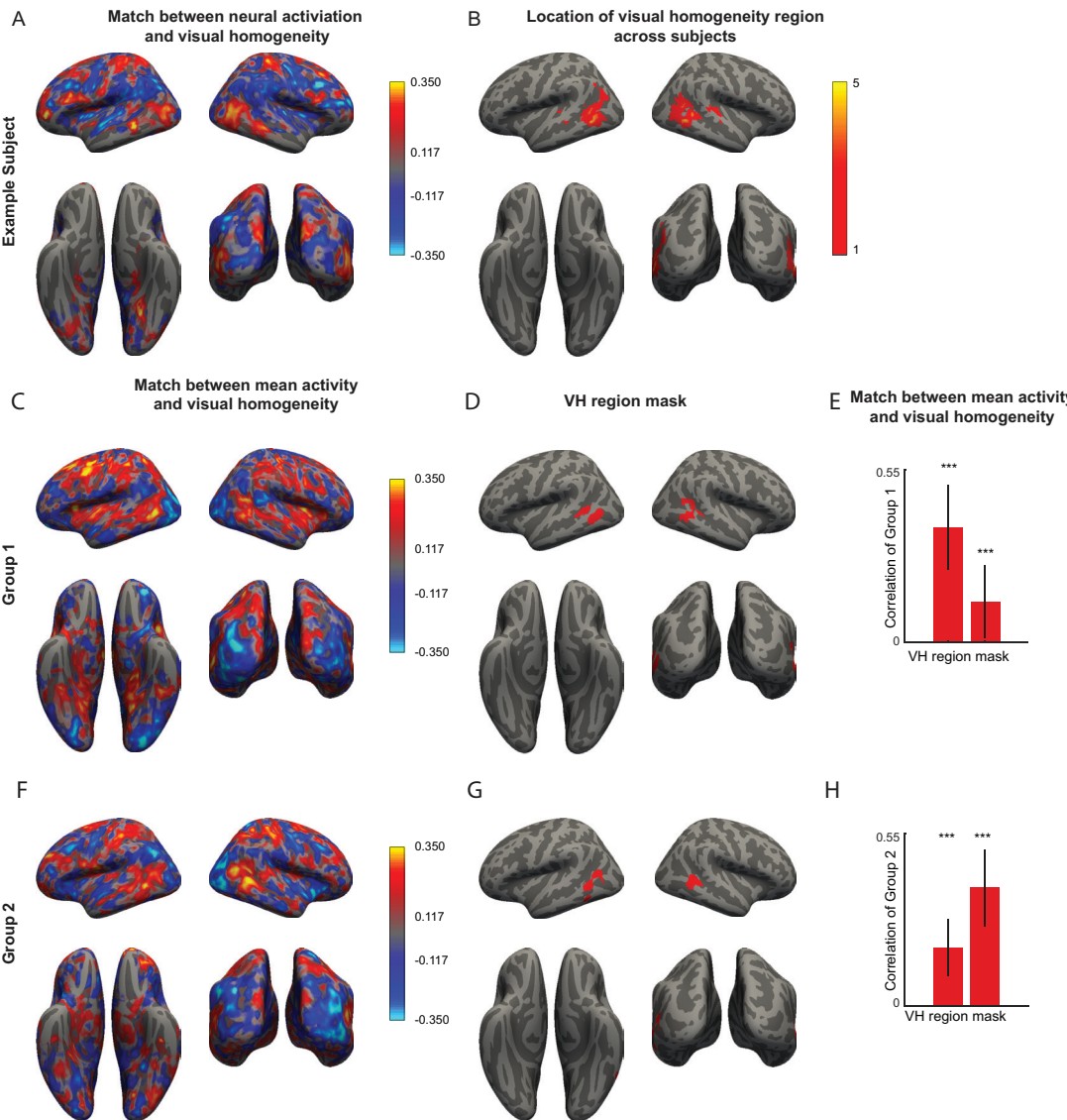

**Appendix 7—figure 2.** Robustness of VH region in symmetry detection. (**A**) Searchlight map showing the correlation between visual homogeneity and mean activation of an example subject. (**B**) Colormap representing the number of subjects for which a particular voxel belonged to the localized VH region. (**C**) Colormap of the correlation between visual homogeneity and mean activation across eight subjects in Group 1. (**A**) VH region obtained by thresholding the searchlight map in panel C. (**B**) Correlation between visual homogeneity and mean activation for participants in Group 1 for the VH region identified from Group 1, and for the VH region identified from Group 2. Asterisks indicate statistical significance of each correlation, obtained by sampling participants with replacement 1000 times, and calculating the fraction of times the correlation was below zero (*** is p<0.005). The significant correlation in Group 2 for the region identified using Group 1 participants suggest that the VH region is consistently localized across subjects. (**C**) Searchlight map similar to panel C, but for 7 participants in Group 2. (**D**) VH region obtained by thresholding the searchlight map in panel F. (**E**) Correlation between visual homogeneity and mean activation for participants in Group 2, for the VH regions identified from Group 1 and from Group 2. Asterisks indicate statistical significance of each correlation, obtained by sampling participants with replacement 1000 times, and calculating the fraction of times the correlation was below zero (*** is p<0.005). The significant correlation in Group 1 for the region identified using Group 2 participants suggest that the VH region is consistently localized across subjects.

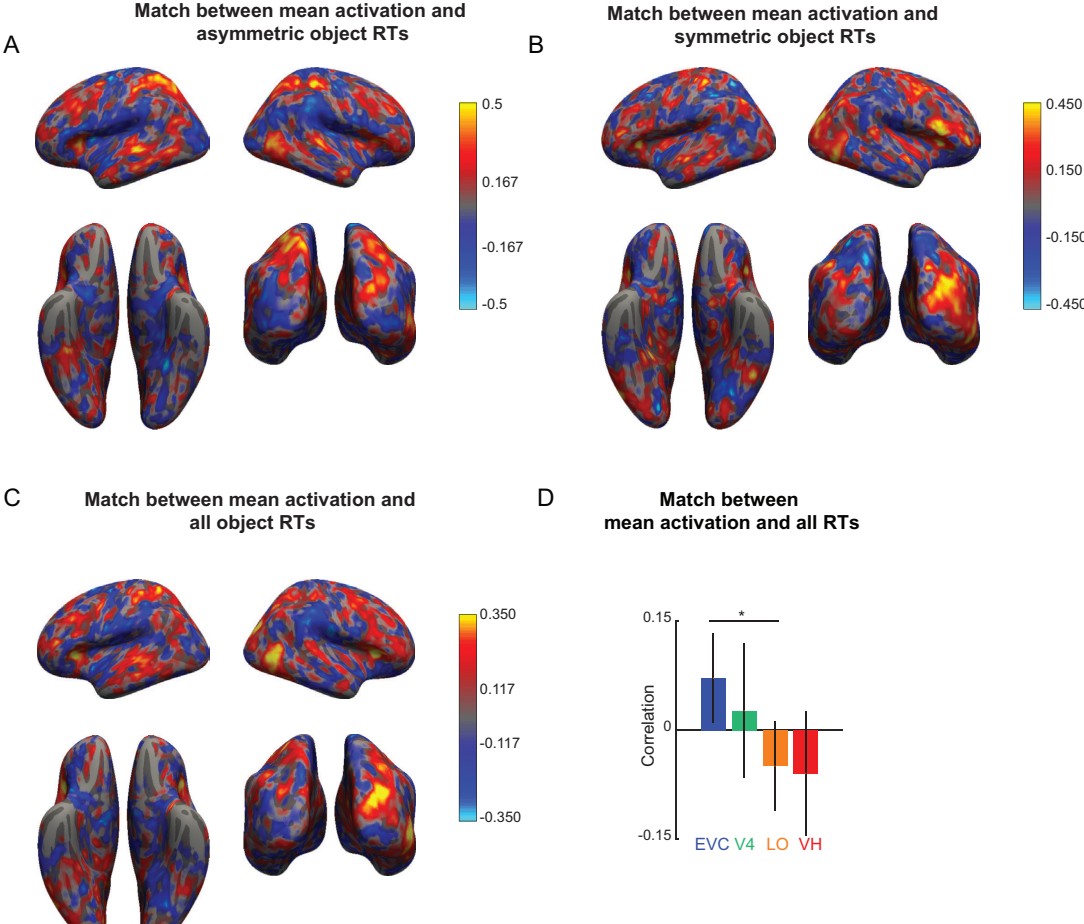

**Appendix 7—figure 3.** Searchlight maps for response time during symmetry task. (**A**) Colormap of correlation between mean activation and response times for asymmetric objects. (**B**) Colormap of correlation between mean activation and response times for symmetric objects. (**C**) Correlation between mean activation and response times across both asymmetric and symmetric objects. To prevent image-wise correlations from being confounded by overall activation level differences, we z-scored the mean activations for each voxel within a particular object type (asymmetric/symmetric) and then combined the mean activations. Likewise, for similar reasons, we z-scored the response times for each particular object type before combining. (**D**) Correlation between mean activation and all response times for key visual regions. Asterisks indicate statistical significance calculated in the same way as *Figure 4D*.

## Appendix 8

### Encoding of visual homogeneity in brain across experiments

In Experiments 1–4, we showed that visual homogeneity can be used to solve a variety of visual tasks that require discriminating between homogeneous and heterogeneous displays. We found that the region anterior to Lateral Occipital Cortex is encoding visual homogeneity signal during both present-absent search task and symmetry judgement task. We compared the spatial location of the visual homogeneity region by directly comparing the region masks derived from Experiments 2 and 4. It can be seen that the VH region localized from the present/absent visual search task in Experiment 2 overlaps reasonably well with the VH region localized from the symmetry detection task in Experiment 4 (*Appendix 8—figure 1*).

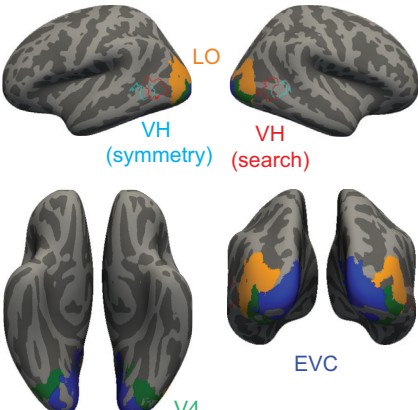

**Appendix 8—figure 1.** Comparing the VH region from Experiments 2 and 4. Key visual regions identified using standard anatomical masks: early visual cortex (EVC), area V4, lateral occipital (LO) region. The VH region from the present/absent search task (Experiment 2, *Figure 4C*) is overlaid with the VH region identified from the symmetry task (Experiment 4, *Figure 6C*).

# Appendix 9

## Relating target-absent responses and symmetry

In Experiments 1–4, we showed that visual homogeneity can be used to solve a variety of visual tasks that require discriminating between homogeneous and heterogeneous displays. These findings suggest a direct empirical link between two disparate tasks, namely visual search for a target and symmetry detection.

In particular, since target-absent response times are inversely related to visual homogeneity, we can take its reciprocal as a direct estimate of visual homogeneity. Since response times for asymmetric and symmetric objects during a symmetry detection task have opposite correlations to visual homogeneity, we predict that they will have opposite correlations with the reciprocal of target-absent response time. In Experiment S7, we tested this prediction by measuring target-absent response times for symmetric and asymmetric objects used in Experiment 4.

## Methods

### Experiment S8. Target present-absent search with symmetric/asymmetric objects

### Participants

A total of 17 participants (10 females, aged 22±5.3 years) participated in this experiment.

### Paradigm

Participants performed a target-present/absent search task. Participants completed 512 correct trials (64 present conditions x 4 repetitions +64 absent conditions x 4 repetitions). Participants reported the presence or absence of the target using the 'P' or 'A' key press. We used the same 64 baton shapes used in Experiment 3. Each array had eight items arranged along a circular grid of radius 5 dva from the centre of the screen. Each item measured 2 dva. There was a random position jitter of ±0.6° in both X and Y directions. All other task details are the same as Experiment 1.

## Results

An example search array from Experiment S8 is shown in *Appendix 9—figure 1A*. Likewise, an example screen from the symmetry detection task is shown in *Appendix 9—figure 1B*.

Participants were highly accurate and consistent in their responses (accuracy, mean ± sd: 97.19 ± 2%; correlation between mean reaction times of even- and odd-numbered participants: $r=0.86$, $p<0.0001$). As predicted, response times to asymmetric objects during the symmetry detection task in Experiment 4 were positively correlated with the inverse of the absent-search times, but this correlation was not statistically significant (*Appendix 9—figure 1C*). Likewise, response times for symmetric objects in Experiment 4 were negatively and significantly correlated with the inverse of absent-search times (*Appendix 9—figure 1D*). Thus, the reciprocal of absent search times can be taken as a decision variable for the symmetry detection task.

Thus, there is a direct and non-trivial empirical link between target-absent response times in a visual search task and response times in a symmetry detection task, because both tasks rely on the same decision variable, namely visual homogeneity.

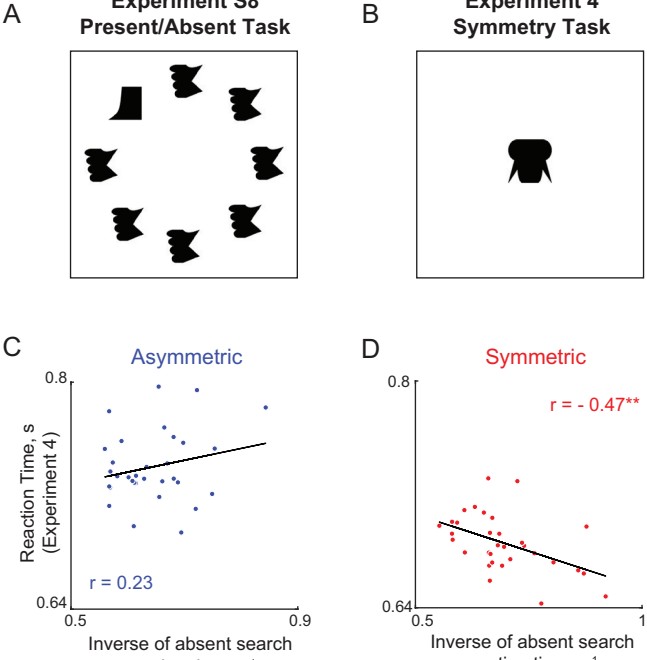

**Appendix 9—figure 1.** Target-absent search times predict symmetry detection. (**A**) Example search array from Experiment S7. (**B**) Example display containing a symmetric object from Experiment 4. (**C**) Response times for asymmetric objects in Experiment 4 plotted against their target-absent response time inverse in Experiment S7. (**D**) Response times for symmetric objects in Experiment 4 plotted against their target-absent response time inverse in Experiment S7.

## Appendix 10

### Visual homogeneity explains animate categorization

#### Introduction

Since visual homogeneity for any image is computed as its distance to some fixed point in perceptual space, we reasoned that shifting this fixed point to the center of a particular object category would make it automatically a decision variable for this category.

To test this prediction, we analyzed data from Experiment 1 of a previous study in which participants had to categorize images as belonging to three hierarchical categories: animals, dogs or Labradors, and also performed an oddball detection task using these same images (*Mohan and Arun, 2012*). The oddball detection task data is already described above as Experiment S1. The categorization task is summarized below.

#### Methods

##### Participants

A total of 12 participants, aged 20–30 years were recruited for the experiment.

##### Stimuli

The stimuli consisted of 48 grayscale natural objects (24 animate, 24 inanimate) presented against a black background.

##### Procedure

Participants performed three separate blocks of categorization: an animal task, a dog task and a Labrador task, with block order counterbalanced across participants. Each block began with a preview of all objects to avoid confusion regarding the category, and each trial consisted of a fixation cross displayed for 750ms, followed by a test object for 50ms followed by a noise mask for 250ms. Participants had to indicate whether the object belonged to the category (animal/dog/Labrador) using a keypress ('Y' if yes, 'N' if no).

##### Model fitting for visual homogeneity

We started by embedding all 48 objects into an eight-dimensional space using multidimensional scaling. For each image, visual homogeneity is calculated as its distance from a fixed point in this multidimensional space. To calculate visual homogeneity for a particular category task, we adjusted the fixed center for visual homogeneity calculations so as to maximize the difference between the correlation of visual homogeneity with objects outside vs within that category. In this manner, we calculated the correlation between visual homogeneity for all objects in the category and the corresponding categorization times, and likewise for the non-category objects.

#### Results

*Appendix 10—figure 1A* shows an example screen from the animal categorization task. For this task, we estimated an optimal center for visual homogeneity computations by maximizing the difference in correlation for inanimate and animate objects. The resulting model fits are shown in *Appendix 10—figure 1B–C*. It can be seen that categorization times for are positively correlated with visual homogeneity for animate objects (*Appendix 10—figure 1B*) but negatively correlated for inanimate objects (*Appendix 10—figure 1C*), suggesting that visual homogeneity can be used to solve the animate categorization task. We obtained similar results for the dog task (*Appendix 10—figure 1E–G*) and for the Labrador task (*Appendix 10—figure 1I–K*).

We conclude that visual homogeneity can serve as a decision variable for categorization tasks at least for the categories tested here, and provided the fixed center for the visual homogeneity computations can be shifted to the center of each category.

### Can target-absent response times explain categorization?

Since the inverse of target-absent response times is a measure of visual homogeneity, we wondered whether they would directly predict the visual homogeneity predictions of each task. To this end, we took the absent-search response times from Experiment S2 and asked whether they are correlated

with the visual homogeneity optimized for each categorization task. This revealed a significant positive correlation for all three tasks (*Appendix 10—figure 2*).

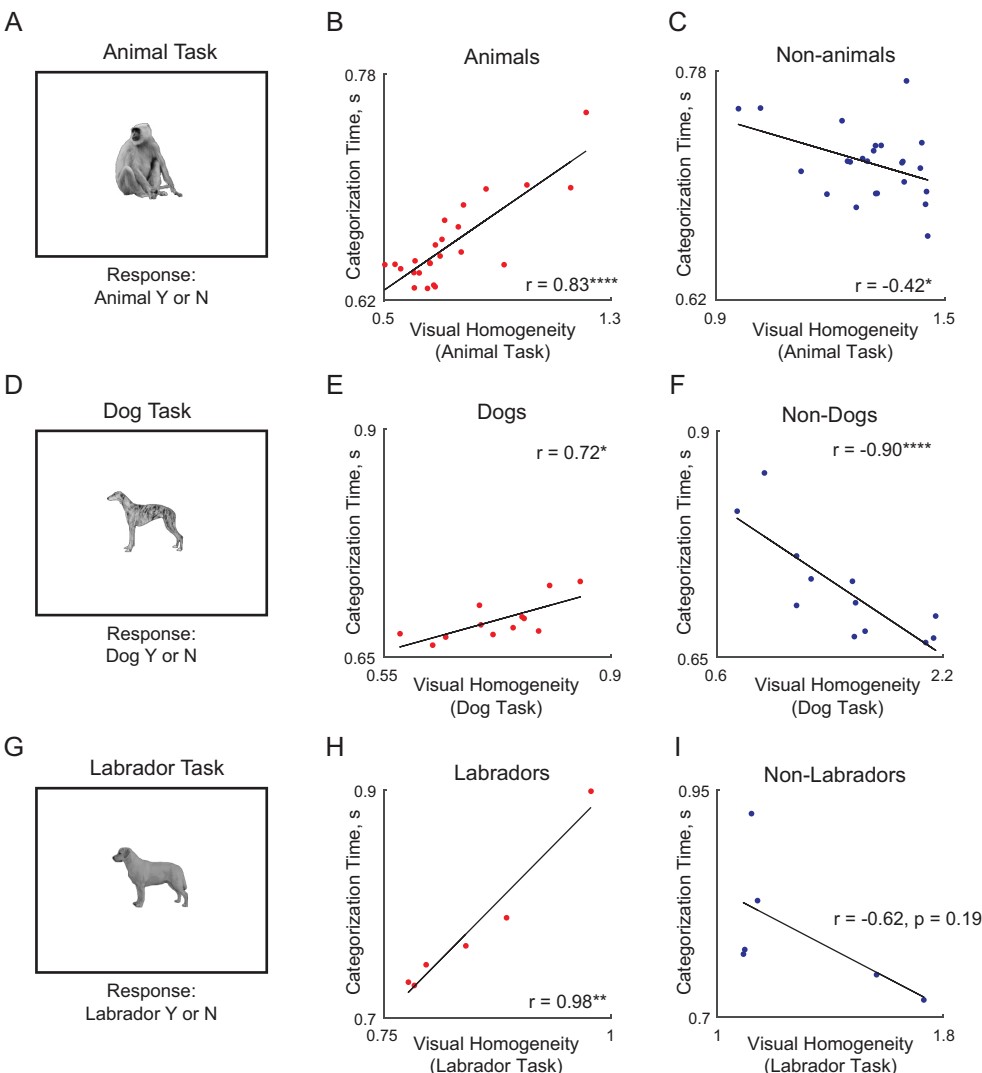

**Appendix 10—figure 1.** Visual homogeneity predicts categorization times.

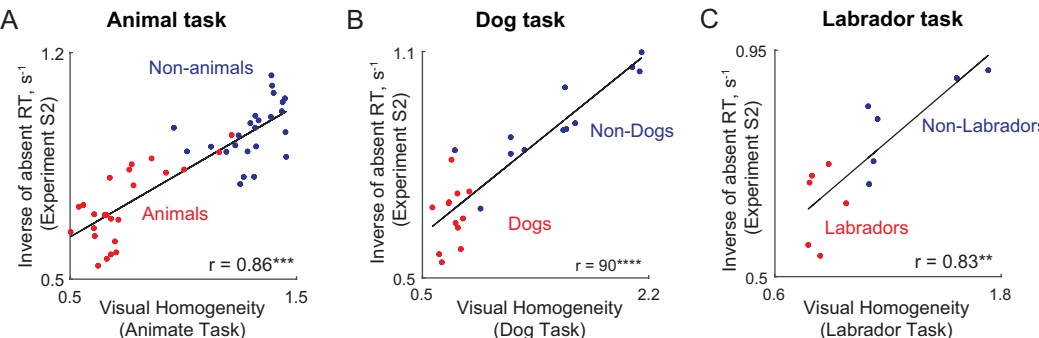

**Appendix 10—figure 2.** Target-absent search predicts visual homogeneity for each category. (**A**) Example trial of the animal categorization task. Stimuli was presented for 50ms followed by a noise mask. Subjects responded if the presented image is an animal or not with Y/N key responses. (**B**) Average categorization times for animals plotted against visual homogeneity relative to an optimum center for this task, calculated from oddball detection task *Appendix 10—figure 2 continued on next page*

*Appendix 10—figure 2 continued*

data (Experiment S1). (**C**) Same as panel B but for inanimate objects. (**D**) Example trial of the dog categorization task. (**E**) Average categorization times for dogs plotted against visual homogeneity relative to an optimum center for this task, calculated from oddball detection task data (Experiment S1). (**F**) Same as panel E but for non-dogs. (**G**) Example trial of the Labrador categorization task. (**H**) Average categorization times for dogs plotted against visual homogeneity relative to an optimum center for this task, calculated from oddball detection task data (Experiment S1). (**I**) Same as panel H but for non-Labradors. (**A**) Inverse of target-absent search times from Experiment S2 plotted against the optimized visual homogeneity from the animate task. (**B**) Inverse of target-absent search times from Experiment S2 but now plotted against the optimized visual homogeneity from the dog task. (**C**) Inverse of target-absent search times from Experiment S2 but now plotted against the optimized visual homogeneity from the Labrador task.

