## [Editor Report · eLife Assessment]

This study uses carefully designed experiments to generate a **useful** behavioural and neuroimaging dataset on visual cognition. The results provide **solid** evidence for the involvement of higher-order visual cortex in processing visual oddballs and asymmetry. However, the evidence provided for the very strong claims of homogeneity as a novel concept in vision science, separable from existing concepts such as target saliency, is **incomplete**. The authors and the reviewers do not agree on several points, which are explained in the reviews and author response.

---

## [Referee Report · Reviewer #1 (Public review)]

Summary:

The authors define a new metric for visual displays, derived from psychophysical response times, called visual homogeneity (VH). They attempt to show that VH is explanatory of response times across multiple visual tasks. They use fMRI to find visual cortex regions with VH-correlated activity. On this basis, they declare a new visual region in human brain, area VH, whose purpose is to represent VH for the purpose of visual search and symmetry tasks.

Link to original review: https://elifesciences.org/reviewed-preprints/93033v2/reviews#peer-review-0

Comments on latest version:

Authors rebuttal: We agree that visual homogeneity is similar to existing concepts such as target saliency, memorability etc. We have proposed it as a separate concept because visual homogeneity has an independent empirical measure (the reciprocal of target-absent search time in oddball search, or the reciprocal of same response time in a same-different task, etc) that may or may not be the same as other empirical measures such as saliency and memorability. Investigating these possibilities is beyond the scope of our study but would be interesting for future work. We have now clarified this in the revised manuscript (Discussion, p. 42).

Reviewer response to rebuttal: Neither the original ms nor the comments on that ms pretended that "visual homogeneity" was entirely separate from target saliency etc. So this is a response to a criticism that was never made. What the authors do claim, and what the comments question, is that they have successfully subsumed long-recognized psychophysical concepts like target saliency etc. under a new, uber-concept, "visual homogeneity" that explains psychophysical experimental results in a more unified and satisfying way. This subsumption of several well-established psychophysical concepts under a new, unified category is what reviewers objected to.

Authors rebuttal: However, we'd like to emphasize that the question of whether visual homogeneity is novel or related to existing concepts misses entirely the key contribution of our study.

Reviewer response to rebuttal: Sorry, but the claim of a new uber-concept in psychophysics, "visual homogeneity", is a major claim of the paper. The fact that it is not the only claim made does not absolve the authors from having to prove it satisfactorily.

"Authors rebuttal: "In addition, the large regions of VH correlations identified in Experiments 1 and 2 vs. Experiments 3 and 4 are barely overlapping. This undermines the claim that VH is a universal quantity, represented in a newly discovered area of visual cortex, that underlies a wide variety of visual tasks and functions."

• We respectfully disagree with your assertion. First of all, there is partial overlap between the VH regions, for which there are several other obvious explanations that must be considered first before dismissing VH outright as a flawed construct. We acknowledge these alternatives in the Results (p. 27), and the relevant text is reproduced below.

"We note that it is not straightforward to interpret the overlap between the VH regions identified in Experiments 2 & 4. The lack of overlap could be due to stimulus differences (natural images in Experiment 2 vs silhouettes in Experiment 4), visual field differences (items in the periphery in Experiment 2 vs items at the fovea in Experiment 4) and even due to different participants in the two experiments. There is evidence supporting all these possibilities: stimulus differences (Yue et al., 2014), visual field differences (Kravitz et al., 2013) as well as individual differences can all change the locus of neural activations in object-selective cortex (Weiner and Grill-Spector, 2012a; Glezer and Riesenhuber, 2013). We speculate that testing the same participants on search and symmetry tasks using similar stimuli and display properties would reveal even larger overlap in the VH regions that drive behavior."

Reviewer response to rebuttal: The authors are saying that their results merely look unconvincing (weak overlap between VH regions defined in different experiments) because there were confounding differences between their experiments, in subject population, stimuli, etc. That is possible, but in that case it is up to the authors to show that their definition of a new "area VH" is convincing when the confounding differences are resolved, e.g. by using the same stimuli in the different experiments they attempt to agglomerate here. That would require new experiments, and none are offered in this revision.

Authors rebuttal: • Thank you for carefully thinking through our logic. We agree that a distance-to-centre calculation is entirely unnecessary as an explanation for target-present visual search. The similarity between target and distractor, so there is nothing new to explain here. However, this is a narrow and selective interpretation of our findings because you are focusing only on our results on target-present searches, which are only half of all our data. The other half is the target-absent responses which previously have had no clear explanation. You are also missing the fact that we are explaining same-different and symmetry tasks as well using the same visual homogeneity computation. We urge you to think more deeply about the problem of how to decide whether an oddball is present or not in the first place. How do we actually solve this task?

Reviewer response to rebuttal: It is the role of the authors to think deeply about their paper and on that basis present a clear and compelling case that readers can understand quickly and agree with. That is not done here.

Authors rebuttal: There must be some underlying representation and decision process. Our study shows that a distance-to-centre computation can actually serve as a decision variable to solve disparate property-based visual tasks. These tasks pose a major challenge to standard models of decision-making because the underlying representation and decision variable have been unclear. Our study resolves this challenge by proposing a novel computation that can be used by the brain to solve all these disparate tasks, and bring these tasks into the ambit of standard theories of decision-making.

Reviewer response to rebuttal: There is only a "challenge" if you accept the authors' a priori assumption that all of these tasks must have a common explanation and rely on a single neural mechanism. I do not accept that assumption, and I don't think the authors provide evidence to support the assumption. There is nothing "unclear" about how search, oddball, etc. have been thoroughly explained, separately, in the psychophysical literature that spans more than a century.

Authors rebuttal: • You are indeed correct in noting that both Experiment 1 & 2 involve oddball search, and so at the superficial level, it looks circular that the oddball search data of Experiment 1 is being used to explain the oddball search data of Experiment 2.

However a deeper scrutiny reveals more fundamental differences: Experiment 1 consisted of only oddball search with the target appearing on the left or right, whereas Experiment 2 consisted of oddball search with the target either present or completely absent. In fact, we were merely using the search dissimilarities from Experiment 1 to reconstruct the underlying object representation, because it is well-known that neural dissimilarities are predicted well by search dissimilarities (Sripati & Olson, 2009; Zhivago et al, 2014).

Reviewer response to rebuttal: Here again the authors cite differences between their multiple experiments as a virtue that supports their conclusions. Instead, the experiments should have been designed for maximum similarity if the authors intended to explain them with the same theory.

Authors rebuttal: To thoroughly refute any lingering concern about circularity, we reasoned that the model predictions for Experiment 2 could have been obtained by a distance-to-center computation on any brain like object representation. To this end, we used object representations from deep neural networks pretrained on object categorization, whose representations are known to match well with the brain, and asked if a distance-to-centre computation on these representations could predict the search data in Experiment 2. This was indeed the case, and these results are now included an additional section in Supplementary Material (Section S1).

Reviewer response to rebuttal: The authors' claims are about human performance and how it is based on the human brain. Their claims are not well supported by the human experiments that they performed. It serves no purpose to redo the same experiments in silico, which cannot provide stronger evidence that compensates for what was lacking in the human data.

Authors rebuttal: "Confirming the generality of visual homogeneity

We performed several additional analyses to confirm the generality of our results, and to reject alternate explanations.

First, it could be argued that our results are circular because they involve taking oddball search times from Experiment 1 and using them to explain search response times in Experiment 2. This is a superficial concern since we are using the search dissimilarities from Experiment 1 only as a proxy for the underlying neural representation, based on previous reports that neural dissimilarities closely match oddball search dissimilarities (Sripati and Olson, 2010; Zhivago and Arun, 2014). Nonetheless, to thoroughly refute this possibility, we reasoned that we would get similar predictions of the target present/absent responses in Experiment using any other brain-like object representation. To confirm this, we replaced the object representations derived from Experiment 1 with object representations derived from deep neural networks pretrained for object categorization, and asked if distance-to-center computations could predict the target present/absent responses in Experiment 2. This was indeed the case (Section S1).

Second, we wondered whether the nonlinear optimization process of finding the best-fitting center could be yielding disparate optimal centres each time. To investigate this, we repeated the optimization procedure with many randomly initialized starting points, and obtained the same best-fitting center each time (see Methods).

Third, to confirm that the above model fits are not due to overfitting, we performed a leave-one-out cross validation analysis. We left out all target-present and target-absent searches involving a particular image, and then predicted these searches by calculating visual homogeneity estimated from all other images. This too yielded similar positive and negative correlations (r = 0.63, p < 0.0001 for target-present, r = -0.63, p < 0.001 for target-absent).

Fourth, if heterogeneous displays indeed elicit similar neural responses due to mixing, then their average distance to other objects must be related to their visual homogeneity. We confirmed that this was indeed the case, suggesting that the average distance of an object from all other objects in visual search can predict visual homogeneity (Section S1).

Fifth, the above results are based on taking the neural response to oddball arrays to be the average of the target and distractor responses. To confirm that averaging was indeed the optimal choice, we repeated the above analysis by assuming a range of relative weights between the target and distractor. The best correlation was obtained for almost equal weights in the lateral occipital (LO) region, consistent with averaging and its role in the underlying perceptual representation (Section S1).

Finally, we performed several additional experiments on a larger set of natural objects as well as on silhouette shapes. In all cases, present/absent responses were explained using visual homogeneity (Section S2)."

Reviewer response to rebuttal: The authors can experiment on side questions for as long as they please, but none of the results described above answer the concern about how center-fitting undercuts the evidentiary value of their main results.

Authors rebuttal: • While it is true that the optimal center needs to be found by fitting to the data, there no particular mystery to the algorithm: we are simply performing a standard gradient-descent to maximize the fit to the data. We have described the algorithm clearly and are making our codes public. We find the algorithm to yield stable optimal centers despite many randomly initialized starting points. We find the optimal center to be able to predict responses to entirely novel images that were excluded during model training. We are making no assumption about the location of centre with respect to individual points. Therefore, we see no cause for concern regarding the center-finding algorithm.

Reviewer response to rebuttal: The point of the original comment was that center-fitting should not be done in the first place because it introduces unknowable effects.

•Authors rebuttal: Most visual tasks, such as finding an animal, are thought to involve building a decision boundary on some underlying neural representation. Even visual search has been portrayed as a signal-detection problem where a particular target is to be discriminated from a distractor. However none of these formulations work in the case of property-based visual tasks, where there is no unique feature to look for.

We are proposing that, when we view a search array, the neural response to the search array can be deduced from the neural responses to the individual elements using well-known rules, and that decisions about an oddball target being present or absent can be made by computing the distance of this neural response from some canonical mean firing rate of a population of neurons. This distance to center computation is what we denote as visual homogeneity. We have revised our manuscript throughout to make this clearer and we hope that this helps you understand the logic better.

• You are absolutely correct that the stimulus complexity should matter, but there are no good empirically derived measures for stimulus complexity, other than subjective ratings which are complex on their own and could be based on any number of other cognitive and semantic factors. But considering what factors are correlated with target-absent response times is entirely different from asking what decision variable or template is being used by participants to solve the task.

Reviewer response to rebuttal: If stimulus complexity is what matters, as the authors agree here, then it is incumbent on them to measure stimulus complexity. The difficulty of measuring stimulus complexity does not justify avoiding the problem with an analysis that ignores complexity.

Authors rebuttal: • We have provided empirical proof for our claims, by showing that target-present response times in a visual search task are correlated with "different" responses in the same-different task, and that target-absent response times in the visual search task are correlated with "same" responses in the same-different task (Section S4).

Reviewer response to rebuttal: Sorry, but there is still no reason to think that same-different judgments are based on a mythical boundary halfway between the two. If there is a boundary, it will be close to the same end of the continuum, where subjects might conceivably miss some tiny difference between two stimuli. The vast majority of "different" stimuli will be entirely different from the same stimulus, producing no confusability, and certainly not a decision boundary halfway between two extremes.

Authors rebuttal: • Again, the opposite correlations between target present/absent search times with VH are the crucial empirical validation of our claims that a distance-to-center calculation explain how we perform these property-based tasks. The VH predictions do not fully explain the data. We have explicitly acknowledged this shortcoming, so we are hardly dismissing it as a problem.

Reviewer response to rebuttal: The authors' acknowledgement of flaws in the ms does not argue in favor of publication, but rather just the opposite.

Authors rebuttal: • Finding an oddball, deciding if two items are same or different and symmetry tasks are disparate visual tasks that do not fit neatly into standard models of decision-making. The key conceptual advance of our study is that we propose a plausible neural representation and decision variable that allows all three property-based visual tasks to be reconciled with standard models of decision-making.

Reviewer response to rebuttal: The original comment stands as written. Same/different will have a boundary very close to the "same" end of the continuum. The boundary is only halfway between two choices if the stimulus design forces the boundary to be there, as in the motion and cat/dog experiments.

Authors rebuttal: "There is no inherent middle point boundary between target present and target absent. Instead, in both types of trial, maximum information is present when target and distractors are most dissimilar, and minimum information is present when target and distractors are most similar. The point of greatest similarity occurs at then limit of any metric for similarity. Correspondingly, there is no middle point dip in information that would produce greater difficulty and higher response times. Instead, task difficulty and response times increase monotonically with similarity between targets and distractors, for both target present and target absent decisions. Thus, in Figs. 2F and 2G, response times appear to be highest for animals, which share the largest numbers of closely similar distractors."

• Your alternative explanation rests on vague factors like "maximum information" which cannot be quantified. By contrast we are proposing a concrete, falsifiable model for three property-based tasks - same/different, oddball present/absent and object symmetry. Any argument based solely on item similarity to explain visual search or symmetry responses cannot explain systematic variations observed for target-absent arrays and for symmetric objects, for the reasons explained earlier.

Reviewer response to rebuttal: There is nothing vague about this comment. The authors use an analysis that assumes a decision boundary at the centerpoint of their arbitrarily defined stimulus space. This assumption is not supported, and it is unlikely, considering that subjects are likely to notice all but the smallest variations between same and different stimuli, putting the boundary nearly at the same end of the continuum, not the very middle.

Authors rebuttal: "(1) The area VH boundaries from different experiments are nearly completely non-overlapping.

In line with their theory that VH is a single continuum with a decision boundary somewhere in the middle, the authors use fMRI searchlight to find an area whose responses positively correlate with homogeneity, as calculated across all of their target present and target absent arrays. They report VH-correlated activity in regions anterior to LO. However, the VH defined by symmetry Experiments 3 and 4 (VHsymmetry) is substantially anterior to LO, while the VH defined by target detection Experiments 1 and 2 (VHdetection) is almost immediately adjacent to LO. Fig. S13 shows that VHsymmetry and VHdetection are nearly non-overlapping. This is a fundamental problem with the claim of discovering a new area that represents a new quantity that explains response times across multiple visual tasks. In addition, it is hard to understand why VHsymmetry does not show up in a straightforward subtraction between symmetric and asymmetric objects, which should show a clear difference in homogeneity."

• We respectfully disagree. The partial overlap between the VH regions identified in Experiments 1 & 2 can hardly be taken as evidence against the quantity VH itself, because there are several other obvious alternate explanations for this partial overlap, as summarized earlier as well. The VH region does show up in a straightforward subtraction between symmetric and asymmetric objects (Section S7), so we are not sure what the Reviewer is referring to here.

Reviewer response to rebuttal: In disagreeing with the comment quoted above, the authors are maintaining that a new functional area of cerebral cortex can be declared even if that area changes location on the cortical map from one experiment to another. That position is patently absurd.

Authors rebuttal: "(3) Definition of the boundaries and purpose of a new visual area in the brain requires circumspection, abundant and convergent evidence, and careful controls.

Even if the VH metric, as defined and calculated by the authors here, is a meaningful quantity, it is a bold claim that a large cortical area just anterior to LO is devoted to calculating this metric as its major task. Vision involves much more than target detection and symmetry detection. Cortex anterior to LO is bound to perform a much wider range of visual functionalities. If the reported correlations can be clarified and supported, it would be more circumspect to treat them as one byproduct of unknown visual processing in cortex anterior to LO, rather than treating them as the defining purpose for a large area of visual cortex."

• We totally agree with you that reporting a new brain region would require careful interpretation and abundant and converging evidence. However, this requires many studies worth of work, and historically category-selective regions like the FFA have achieved consensus only after they were replicated and confirmed across many studies. We believe our proposal for the computation of a quantity like visual homogeneity is conceptually novel, and our study represents a first step that provides some converging evidence (through replicable results across different experiments) for such a region. We have reworked our manuscript to make this point clearer (Discussion, p 32).

Reviewer response to rebuttal: Indeed, declaring a new brain area depends on much more work than is done here. Thus, the appropriate course here is to wait before claiming to have identified a new cortical area.

---

## [Referee Report · Reviewer #2 (Public review)]

Summary:

This study proposes visual homogeneity as a novel visual property that enables observers perform to several seemingly disparate visual tasks, such as finding an odd item, deciding if two items are same, or judging if an object is symmetric. In Exp 1, the reaction times on several objects were measured in human subjects. In Exp 2, visual homogeneity of each object was calculated based on the reaction time data. The visual homogeneity scores predicted reaction times. This value was also correlated with the BOLD signals in a specific region anterior to LO. Similar methods were used to analyze reaction time and fMRI data in a symmetry detection task. It is concluded that visual homogeneity is an important feature that enables observers to solve these two tasks.

Strengths:

(1) The writing is very clear. The presentation of the study is informative.

(2) This study includes several behavioral and fMRI experiments. I appreciate the scientific rigor of the authors.

Weaknesses:

Before addressing the manuscript itself, I would like to comment the review process first. Having read the lasted revised manuscript, I shared many of the concerns raised by the two reviewers in the last two rounds of review. It appears that the authors have disagreed with the majority of comments made by the two reviewers. If so, I strongly recommend that the authors proceed to make this revision as a Version of Record and conclude this review process. According to eLife's policy that the authors have the right to make a Version of Record at any time during the review process, and I fully respect that right. However, I also ask that the authors respect the reviewer's right to retain the comments regarding this paper.

Beside that, I still have several further questions about this study.

(1) My main concern with this paper is the way visual homogeneity is computed. On page 10, lines 188-192, it says: "we then asked if there is any point in this multidimensional representation such that distances from this point to the target-present and target-absent response vectors can accurately predict the target-present and target-absent response times with a positive and negative correlation respectively (see Methods)". This is also true for the symmetry detection task. If I understand correctly, the reference point in this perceptual space was found by deliberating satisfying the negative and positive correlations in response times. And then on page 10, lines 200-205, it shows that the positive and negative correlations actually exist. This logic is confusing. The positive and negative correlations emerge only because this method is optimized to do so. It seems more reasonable to identify the reference point of this perceptual space independently, without using the reaction time data. Otherwise, the inference process sounds circular. A simple way is to just use the mean point of all objects in Exp 1, without any optimization towards reaction time data.

I raised this question in my initial review. However, the authors did not address whether the positive and negative correlations still hold if the mean point is defined as the reference point without any optimization. The authors also argue that it is similar to a case of fitting a straight line. It is fine that the authors insist on the straight line (e.g., correlation). However, I would not call "straight line correlations" a "quantitative model" as a high-profile journals like eLife. Please remove all related arguments of a novel quantitative model.

(2) Visual homogeneity (at least given the current form) is an unnecessary term. It is similar to distractor heterogeneity/distractor variability/distractor saliency in literature. However, the authors attempt to claim it as a novel concept. Both R1 and me raised this question in the very first review. However, the authors refused to revise the manuscript. In the last review, I mentioned this and provided some example sentences claiming novelty. The authors only revised the last sentence of the abstract, and even did not bother to revise the last sentence of significance: "we show that these tasks can be solved using a simple property WE DEFINE as visual homogeneity". Also, lines 851 still shows "we have defined a NOVEL image property, visual homogeneity...". I am confused about whether the authors agree or disagree that "visual homogeneity is an unnecessary term". If the authors agree, they should completely remove the related phrase throughout the paper. If not, they should keep all these and state the reasons. I don't think this is a correct approach to revising a manuscript.

(3) If the authors agree that visual homogeneity is not new, I suggest a complete rewrite of the title, abstract, significance, and introduction. Let me ask a simple question, can we remove "visual homogeneity" and use some more well-established term like "image feature similarity"? If yes, visual homogeneity is unnecessary.

(4) If I understand it correctly, one of the key findings of this paper is "the response times for target-present searches were positively correlated with visual homogeneity. By contrast, the response times for target-absent searches were negatively correlated with visual homogeneity" (lines 204-207). I think the authors have already acknowledged that this positive correlation is not surprising at all because it reflects the classic target-distractor similarity effect. If this is the case, please completely remove the positive correlation as a novel prediction and finding.

(5) In my last review, I mentioned the seminal paper by Duncan and Humphreys (1989) has clearly stated that "difficulty increases with increased similarity of targets to nontargets and decreased similarity between nontargets" (the sentence in their abstract). Here, "similarity between nontargets" is the same as the visual homogeneity defined here. Similar effects have been shown in Duncan (1989) and Nagy, Neriani, and Young (2005). See also the inconsistent results in Nagy& Thomas, 2003, Vicent, Baddeley, Troscianko&Gilchrist, 2009. More recently, Wei Ji Ma has systematically investigated the effects of heterogeneous distractors in visual search. I think the introduction part of Wei Ji Ma's paper (2020) provides a nice summary of this line of research.

Thanks to the authors' revision, I now better understand the negative correlation. The between-distrator similarity mentioned above describes the heterogeneity of distractors WITHIN an image. However, if I understand it correctly, this study aims to address the negative correlation of reaction time and target-absent stimuli ACROSS images. In other words, why do humans show a shorter reaction time to an image of four pigeons than to an image of four dogs (as shown in Figure 2C), simply because the later image is closer to the reference point of the image space. In this sense, this negative correlation is indeed not the same as distractor heterogeneity. However, this is known as the saliency effect or oddball effects. For example, it seems quite natural to me that humans respond faster to a fish image if the image set contains many images of four-leg dogs that look very different from fish. If this is indeed a saliency effect, why should we define a new term "visual homogeneity"?

(6) The section "key predictions" is quite straightforward. I understand the logic of positive and negative correlations. However, what is the physical meaning of "decision boundary" (Fig. 1G) here? How does the "decision boundary" map on the image space?

(7) In my opinion, one of the advantages of this study is the fMRI dataset, which is valuable because previous studies did not collect fMRI data. The key contribution may be the novel brain region associated with display heterogeneity. If this is the case, I would suggest using a more parametric way to measure this region. For example, one can use Gabor stimuli and systematically manipulate the variations of multiple Gabor stimuli, the same logic also applies to motion direction. If this study uses static Gabor, random dot motion, object images that span from low-level to high-level visual stimuli, and consistently shows that the stimulus heterogeneity is encoded in one brain region, I would say this finding is valuable. But this sounds another experiment. In other words, it is insufficient to claim a new brain region given the current form of the manuscript.

References:

* Duncan, J., & Humphreys, G. W. (1989). Visual search and stimulus similarity. Psychological Review, 96(3), 433-458. doi: 10.1037/0033-295x.96.3.433

* Duncan, J. (1989). Boundary conditions on parallel processing in human vision. Perception, 18(4), 457-469. doi: 10.1068/p180457

* Nagy, A. L., Neriani, K. E., & Young, T. L. (2005). Effects of target and distractor heterogeneity on search for a color target. Vision Research, 45(14), 1885-1899. doi: 10.1016/j.visres.2005.01.007

* Nagy, A. L., & Thomas, G. (2003). Distractor heterogeneity, attention, and color in visual search. Vision Research, 43(14), 1541-1552. doi: 10.1016/s0042-6989(03)00234-7

* Vincent, B., Baddeley, R., Troscianko, T., & Gilchrist, I. (2009). Optimal feature integration in visual search. Journal of Vision, 9(5), 15-15. doi: 10.1167/9.5.15

* Singh, A., Mihali, A., Chou, W. C., & Ma, W. J. (2023). A Computational Approach to Search in Visual Working Memory.

* Mihali, A., & Ma, W. J. (2020). The psychophysics of visual search with heterogeneous distractors. BioRxiv, 2020-08.

* Calder-Travis, J., & Ma, W. J. (2020). Explaining the effects of distractor statistics in visual search. Journal of Vision, 20(13), 11-11.

---

## [Referee Report · Reviewer #3 (Public review)]

Summary of the review process from the Reviewing Editor:

The authors and the reviewers did not agree on several important points made in this paper. The reviewers were critical of the operationalisation of the concept of visual homogeneity (VH), and questioned its validity. For instance, they found it unsatisfying that VH was not calculated on the basis of images themselves, but on the basis of reaction times instead. The authors responded by providing further explanation and argumentation for the importance of this novel concept, but the reviewers were not persuaded. The reviewers also pointed out some data features that did not fit the theory (e.g., overlapping VH between present and absent stimuli), which the authors acknowledge as a point that needs further refining. Finally, the reviewers pointed out that the new so-called visual homogeneity brain region does not overlap very much in the two studies, to which the authors have responded that it is remarkable that there is even partial overlap, given the many confounding differences between the two studies. Altogether, the authors have greatly elaborated their case for VH as an important concept, but the reviewers were not persuaded, and we conclude that the current evidence does not yet meet the high bar for declaring that a novel image property, visual homogeneity, is computed in a localised brain region.

---

## [Author Response]

The following is the authors’ response to the previous reviews.

We are grateful to the editors and reviewers for their careful reading and constructive comments. We have now done our best to respond to them fully through additional analyses and text revisions. In the sections below, the original reviewer comments are in black, and our responses are in red.

To summarize, the major changes in this round of review are as follows:

(1) We have included a new introductory figure (Figure 1) to explain the distinction between feature-based tasks and property-based tasks.

(2) We have included a section on “key predictions” and a section on “overview of this study” in the Introduction to clearly delineate our key predictions and provide a overview of our study.

(3) We have included additional analyses to address the reviewers’ concerns about circularity in Experiments 1 & 2. We show that distance-to-center or visual homogeneity computations performed on object representations obtained from deep networks (instead of the perceptual dissimilarities from Experiment 1) also yields comparable predictions of target-present and target-absent responses in Experiment 2.

(4) We have extensively reworked the manuscript wherever possible to address the specific concerns raised by the reviewers.

We hope that the revised manuscript adequately addresses the concerns raised in this round of review, and we look forward to a positive assessment.

**eLife Assessment**
This study uses carefully designed experiments to generate a useful behavioural and neuroimaging dataset on visual cognition. The results provide solid evidence for the involvement of higher-order visual cortex in processing visual oddballs and asymmetry. However, the evidence provided for the very strong claims of homogeneity as a novel concept in vision science, separable from existing concepts such as target saliency, is inadequate.

Thank you for your positive assessment. We agree that visual homogeneity is similar to existing concepts such as target saliency, memorability etc. We have proposed it as a separate concept because visual homogeneity has an independent empirical measure (the reciprocal of target-absent search time in oddball search, or the reciprocal of same response time in a same-different task, etc) that may or may not be the same as other empirical measures such as saliency and memorability. Investigating these possibilities is beyond the scope of our study but would be interesting for future work. We have now clarified this in the revised manuscript (Discussion, p. 42).

However, we’d like to emphasize that the question of whether visual homogeneity is novel or related to existing concepts misses entirely the key contribution of our study.

Our key contribution is a quantitative, falsifiable model for how the brain could be solving property-based tasks like same-different, oddball or symmetry. Most theories of decision making consider feature-based tasks where there is a well-defined feature space and decision variable. Property-based tasks pose a significant challenge to standard theories since it is not clear how these tasks could be solved. In fact, oddball search, same-different and symmetry tasks have been considered so different that they are rarely even mentioned in the same study. Our study represents a unifying framework showing that all three tasks can be understood as solving the same underlying fundamental problem, and presents evidence in favor of this solution.

**Public Reviews:**

**Reviewer #1 (Public Review):**
Summary:The authors define a new metric for visual displays, derived from psychophysical response times, called visual homogeneity (VH). They attempt to show that VH is explanatory of response times across multiple visual tasks. They use fMRI to find visual cortex regions with VH-correlated activity. On this basis, they declare a new visual region in human brain, area VH, whose purpose is to represent VH for the purpose of visual search and symmetry tasks.

Thank you for your accurate and positive assessment.

Strengths:The authors present carefully designed experiments, combining multiple types of visual judgments and multiple types of visual stimuli with concurrent fMRI measurements. This is a rich dataset with many possibilities for analysis and interpretation.

Thank you for your accurate and positive assessment.

Weaknesses:The datasets presented here should provide a rich basis for analysis. However, in this version of the manuscript, I believe that there are major problems with the logic underlying the authors' new theory of visual homogeneity (VH), with the specific methods they used to calculate VH, and with their interpretation of psychophysical results using these methods. These problems with the coherency of VH as a theoretical construct and metric value make it hard to interpret the fMRI results based on searchlight analysis of neural activity correlated with VH.

We respectfully disagree with your concerns, and have done our best to respond to them fully below.

In addition, the large regions of VH correlations identified in Experiments 1 and 2 vs. Experiments 3 and 4 are barely overlapping. This undermines the claim that VH is a universal quantity, represented in a newly discovered area of visual cortex, that underlies a wide variety of visual tasks and functions.

We respectfully disagree with your assertion. First of all, there is partial overlap between the VH regions, for which there are several other obvious explanations that must be considered first before dismissing VH outright as a flawed construct. We acknowledge these alternatives in the Results (p. 27), and the relevant text is reproduced below.

“We note that it is not straightforward to interpret the overlap between the VH regions identified in Experiments 2 & 4. The lack of overlap could be due to stimulus differences (natural images in Experiment 2 vs silhouettes in Experiment 4), visual field differences (items in the periphery in Experiment 2 vs items at the fovea in Experiment 4) and even due to different participants in the two experiments. There is evidence supporting all these possibilities: stimulus differences (Yue et al., 2014), visual field differences (Kravitz et al., 2013) as well as individual differences can all change the locus of neural activations in object-selective cortex (Weiner and Grill-Spector, 2012a; Glezer and Riesenhuber, 2013). We speculate that testing the same participants on search and symmetry tasks using similar stimuli and display properties would reveal even larger overlap in the VH regions that drive behavior.”

Maybe I have missed something, or there is some flaw in my logic. But, absent that, I think the authors should radically reconsider their theory, analyses, and interpretations, in light of detailed comments below, in order to make the best use of their extensive and valuable datasets combining behavior and fMRI. I think doing so could lead to a much more coherent and convincing paper, albeit possibly supporting less novel conclusions.

We respectfully disagree with your assessment, and we hope that our detailed responses below will convince you of the merit of our claims.

THEORY AND ANALYSIS OF VH(1) VH is an unnecessary, complex proxy for response time and target-distractor similarity.VH is defined as a novel visual quality, calculable for both arrays of objects (as studied in Experiments 1-3) and individual objects (as studied in Experiment 4). It is derived from a center-to-distance calculation in a perceptual space. That space in turn is derived from multi-dimensional scaling of response times for target-distractor pairs in an oddball detection task (Experiments 1 and 2) or in a same different task (Experiments 3 and 4). Proximity of objects in the space is inversely proportional to response times for arrays in which they were paired. These response times are higher for more similar objects. Hence, proximity is proportional to similarity. This is visible in Fig. 2B as the close clustering of complex, confusable animal shapes.VH, i.e. distance-to-center, for target-present arrays is calculated as shown in Fig. 1C, based on a point on the line connecting target and distractors. The authors justify this idea with previous findings that responses to multiple stimuli are an average of responses to the constituent individual stimuli. The distance of the connecting line to the center is inversely proportional to the distance between the two stimuli in the pair, as shown in Fig. 2D. As a result, VH is inversely proportional to distance between the stimuli and thus to stimulus similarity and response times. But this just makes VH a highly derived, unnecessarily complex proxy for target-distractor similarity and response time. The original response times on which the perceptual space is based are far more simple and direct measures of similarity for predicting response times.

Thank you for carefully thinking through our logic. We agree that a distance-to-centre calculation is entirely unnecessary as an explanation for target-present visual search. The difficulty of target-present search is already known to be directly proportional to the similarity between target and distractor, so there is nothing new to explain here.

However, this is a narrow and selective interpretation of our findings because you are focusing only on our results on target-present searches, which are only half of all our data. The other half is the target-absent responses which previously have had no clear explanation. You are also missing the fact that we are explaining same-different and symmetry tasks as well using the same visual homogeneity computation.

We urge you to think more deeply about the problem of how to decide whether an oddball is present or not in the first place. How do we actually solve this task? There must be some underlying representation and decision process. Our study shows that a distance-to-centre computation can actually serve as a decision variable to solve disparate property-based visual tasks. These tasks pose a major challenge to standard models of decision making, because the underlying representation and decision variable have been unclear. Our study resolves this challenge by proposing a novel computation that can be used by the brain to solve all these disparate tasks, and bring these tasks into the ambit of standard theories of decision making.

Our results also explain several interesting puzzles in the literature. If oddball search was driven only by target-distractor similarity, the time taken to respond when a target is absent should not vary at all, and should actually take longer than all target-present searches. But in fact, systematic variations in target-absent times have been observed always in the literature, but have never been explained using any theoretical models. Our results explain why target-absent times vary systematically – it is due to visual homogeneity.

Similarly, in same-different tasks, participants are known to take longer to make a “different” response when the two items differ only slightly. By this logic, they should take the longest to make a “same” response, but in fact, paradoxically, participants are actually faster to make “same” responses. This fast-same effect has been noted several times, but never explained using any models. Our results provide an explanation of why “same” responses to an image vary systematically – it is due to visual homogeneity.

Finally, in symmetry tasks, symmetric objects evoke fast responses, and this has always been taken as evidence for special symmetry computations in the brain. But we show that the same distance-to-center computation can explain both responses to symmetric and asymmetric objects. Thus there is no need for a special symmetry computation in the brain.

(2) The use of VH derived from Experiment 1 to predict response times in Experiment 2 is circular and does not validate the VH theory.The use of VH, a response time proxy, to predict response times in other, similar tasks, using the same stimuli, is circular. In effect, response times are being used to predict response times across two similar experiments using the same stimuli. Experiment 1 and the target present condition of Experiment 2 involve the same essential task of oddball detection. The results of Experiment 1 are converted into VH values as described above, and these are used to predict response times in experiment 2 (Fig. 2F). Since VH is a derived proxy for response values in Experiment 1, this prediction is circular, and the observed correlation shows only consistency between two oddball detection tasks in two experiments using the same stimuli.

You are indeed correct in noting that both Experiment 1 & 2 involve oddball search, and so at the superficial level, it looks circular that the oddball search data of Experiment 1 is being used to explain the oddball search data of Experiment 2.

However a deeper scrutiny reveals more fundamental differences: Experiment 1 consisted of only oddball search with the target appearing on the left or right, whereas Experiment 2 consisted of oddball search with the target either present or completely absent. In fact, we were merely using the search dissimilarities from Experiment 1 to reconstruct the underlying object representation, because it is well known that neural dissimilarities are predicted well by search dissimilarities (Sripati & Olson, 2009; Zhivago et al, 2014).

To thoroughly refute any lingering concern about circularity, we reasoned that the model predictions for Experiment 2 could have been obtained by a distance-to-center computation on any brain like object representation. To this end, we used object representations from deep neural networks pretrained on object categorization, whose representations are known to match well with the brain, and asked if a distance-to-centre computation on these representations could predict the search data in Experiment 2. This was indeed the case, and these results are now included an additional section in Supplementary Material (Section S1).

(3) The negative correlation of target-absent response times with VH as it is defined for target-absent arrays, based on distance of a single stimulus from center, is uninterpretable without understanding the effects of center-fitting. Most likely, center-fitting and the different VH metric for target-absent trials produce an inverse correlation of VH with target-distractor similarity.

Unfortunately, as we have mentioned above, target-distractor similarity cannot explain how target-absent searches behave, since there is no distractor in such searches.

We do understand your broader concern about the center-fitting algorithm itself. We performed a number of additional analyses to confirm the generality of our results and reject alternate explanations – these are summarized in a new section titled “Confirming the generality of visual homogeneity” (p. 12), and the section is reproduced below for your convenience.

“Confirming the generality of visual homogeneity

We performed several additional analyses to confirm the generality of our results, and to reject alternate explanations.

First, it could be argued that our results are circular because they involve taking oddball search times from Experiment 1 and using them to explain search response times in Experiment 2. This is a superficial concern since we are using the search dissimilarities from Experiment 1 only as a proxy for the underlying neural representation, based on previous reports that neural dissimilarities closely match oddball search dissimilarities (Sripati and Olson, 2010; Zhivago and Arun, 2014). Nonetheless, to thoroughly refute this possibility, we reasoned that we would get similar predictions of the target present/absent responses in Experiment using any other brain-like object representation. To confirm this, we replaced the object representations derived from Experiment 1 with object representations derived from deep neural networks pretrained for object categorization, and asked if distance-to-center computations could predict the target present/absent responses in Experiment 2. This was indeed the case (Section S1).

Second, we wondered whether the nonlinear optimization process of finding the best-fitting center could be yielding disparate optimal centres each time. To investigate this, we repeated the optimization procedure with many randomly initialized starting points, and obtained the same best-fitting center each time (see Methods).

Third, to confirm that the above model fits are not due to overfitting, we performed a leave-one-out cross validation analysis. We left out all target-present and target-absent searches involving a particular image, and then predicted these searches by calculating visual homogeneity estimated from all other images. This too yielded similar positive and negative correlations (r = 0.63, p < 0.0001 for target-present, r = -0.63, p < 0.001 for target-absent).

Fourth, if heterogeneous displays indeed elicit similar neural responses due to mixing, then their average distance to other objects must be related to their visual homogeneity. We confirmed that this was indeed the case, suggesting that the average distance of an object from all other objects in visual search can predict visual homogeneity (Section S1).

Fifth, the above results are based on taking the neural response to oddball arrays to be the average of the target and distractor responses. To confirm that averaging was indeed the optimal choice, we repeated the above analysis by assuming a range of relative weights between the target and distractor. The best correlation was obtained for almost equal weights in the lateral occipital (LO) region, consistent with averaging and its role in the underlying perceptual representation (Section S1).

Finally, we performed several additional experiments on a larger set of natural objects as well as on silhouette shapes. In all cases, present/absent responses were explained using visual homogeneity (Section S2).”

The construction of the VH perceptual space also involves fitting a "center" point such that distances to center predict response times as closely as possible. The effect of this fitting process on distance-to-center values for individual objects or clusters of objects is unknowable from what is presented here. These effects would depend on the residual errors after fitting response times with the connecting line distances. The center point location and its effects on distance-to-center of single objects and object clusters are not discussed or reported here.

While it is true that the optimal center needs to be found by fitting to the data, there no particular mystery to the algorithm: we are simply performing a standard gradient-descent to maximize the fit to the data. We have described the algorithm clearly and are making our codes public. We find the algorithm to yield stable optimal centers despite many randomly initialized starting points. We find the optimal center to be able to predict responses to entirely novel images that were excluded during model training. We are making no assumption about the location of centre with respect to individual points. Therefore, we see no cause for concern regarding the center-finding algorithm.

Yet, this uninterpretable distance-to-center of single objects is chosen as the metric for VH of target-absent displays (VHabsent). This is justified by the idea that arrays of a single stimulus will produce an average response equal to one stimulus of the same kind. But it is not logically clear why response strength to a stimulus should be a metric for homogeneity of arrays constructed from that stimulus, or even what homogeneity could mean for a single stimulus from this set. And it is not clear how this VHabsent metric based on single stimuli can be equated to the connecting line VH metric for stimulus pairs, i.e. VHpresent, or how both could be plotted on a single continuum.

Most visual tasks, such as finding an animal, are thought to involve building a decision boundary on some underlying neural representation. Even visual search has been portrayed as a signal-detection problem where a particular target is to be discriminated from a distractor. However none of these formulations work in the case of property-based visual tasks, where there is no unique feature to look for.

We are proposing that, when we view a search array, the neural response to the search array can be deduced from the neural responses to the individual elements using well known rules, and that decisions about an oddball target being present or absent can be made by computing the distance of this neural response from some canonical mean firing rate of a population of neurons. This distance to center computation is what we denote as visual homogeneity. We have revised our manuscript throughout to make this clearer and we hope that this helps you understand the logic better.

It is clear, however, what *should* be correlated with difficulty and response time in the target-absent trials, and that is the complexity of the stimuli and the numerosity of similar distractors in the overall stimulus set. Complexity of the target, similarity with potential distractors, and number of such similar distractors all make ruling out distractor presence more difficult. The correlation seen in Fig. 2G must reflect these kinds of effects, with higher response times for complex animal shapes with lots of similar distractors and lower response times for simpler round shapes with fewer similar distractors.

You are absolutely correct that the stimulus complexity should matter, but there are no good empirically derived measures for stimulus complexity, other than subjective ratings which are complex on their own and could be based on any number of other cognitive and semantic factors. But considering what factors are correlated with target-absent response times is entirely different from asking what decision variable or template is being used by participants to solve the task.

The example points in Fig. 2G seem to bear this out, with higher response times for the deer stimulus (complex, many close distractors in the Fig. 2B perceptual space) and lower response times for the coffee cup (simple, few close distractors in the perceptual space). While the meaning of the VH scale in Fig. 2G, and its relationship to the scale in Fig. 2F, are unknown, it seems like the Fig. 2G scale has an inverse relationship to stimulus complexity, in contrast to the expected positive relationship for Fig. 2F. This is presumably what creates the observed negative correlation in Fig. 2G.Taken together, points 1-3 suggest that VHpresent and VHabsent are complex, unnecessary, and disconnected metrics for understanding target detection response times. The standard, simple explanation should stand. Task difficulty and response time in target detection tasks, in both present and absent trials, are positively correlated with target-distractor similarity.

We strongly disagree. Your assessment seems to be based on only considering target-present searches, which are of course driven by target-distractor similarity. Your argument is flawed because systematic variations in target-absent trials cannot be linked to any target-distractor similarity since there are no targets in the first place in such trials.

We have shown that target-absent response times are in fact, independent of experimental context, which means that they index an image property that is independent of any reference target (Results, p. 15; Section S4). This property is what we define as visual homogeneity.

I think my interpretations apply to Experiments 3 and 4 as well, although I find the analysis in Fig. 4 especially hard to understand. The VH space in this case is based on Experiment 3 oddball detection in a stimulus set that included both symmetric and asymmetric objects. But the response times for a very different task in Experiment 4, a symmetric/asymmetric judgment, are plotted against the axes derived from Experiment 3 (Fig. 4F and 4G). It is not clear to me why a measure based on oddball detection that requires no use of symmetry information should be predictive of within-stimulus symmetry detection response times. If it is, that requires a theoretical explanation not provided here.

We were simply using an oddball detection task to construct the underlying object representation, on the basis of observations that search dissimilarities are strongly correlated with neural dissimilarities. In Section S1, we show that similar results could have been obtained using other object representations such as deep networks, as long as the representation is brain-like.

(4) Contrary to the VH theory, same/different tasks are unlikely to depend on a decision boundary in the middle of a similarity or homogeneity continuum.

We have provided empirical proof for our claims, by showing that target-present response times in a visual search task are correlated with “different” responses in the same-different task, and that target-absent response times in the visual search task are correlated with “same” responses in the same-different task (Section S4).

The authors interpret the inverse relationship of response times with VHpresent and VHabsent, described above, as evidence for their theory. They hypothesize, in Fig. 1G, that VHpresent and VHabsent occupy a single scale, with maximum VHpresent falling at the same point as minimum VHabsent. This is not borne out by their analysis, since the VHpresent and VHabsent value scales are mainly overlapping, not only in Experiments 1 and 2 but also in Experiments 3 and 4. The authors dismiss this problem by saying that their analyses are a first pass that will require future refinement. Instead, the failure to conform to this basic part of the theory should be a red flag calling for revision of the theory.

Again, the opposite correlations between target present/absent search times with VH are the crucial empirical validation of our claims that a distance-to-center calculation explain how we perform these property-based tasks. The VH predictions do not fully explain the data. We have explicitly acknowledged this shortcoming, so we are hardly dismissing it as a problem.

The reason for this single scale is that the authors think of target detection as a boundary decision task, along a single scale, with a decision boundary somewhere in the middle, separating present and absent. This model makes sense for decision dimensions or spaces where there are two categories (right/left motion; cats vs. dogs), separated by an inherent boundary (equal left/right motion; training-defined cat/dog boundary). In these cases, there is less information near the boundary, leading to reduced speed/accuracy and producing a pattern like that shown in Fig. 1G.

Finding an oddball, deciding if two items are same or different and symmetry tasks are disparate visual tasks that do not fit neatly into standard models of decision making. The key conceptual advance of our study is that we propose a plausible neural representation and decision variable that allow all three property-based visual tasks to be reconciled with standard models of decision making.

This logic does not hold for target detection tasks. There is no inherent middle point boundary between target present and target absent. Instead, in both types of trial, maximum information is present when target and distractors are most dissimilar, and minimum information is present when target and distractors are most similar. The point of greatest similarity occurs at then limit of any metric for similarity. Correspondingly, there is no middle point dip in information that would produce greater difficulty and higher response times. Instead, task difficulty and response times increase monotonically with similarity between targets and distractors, for both target present and target absent decisions. Thus, in Figs. 2F and 2G, response times appear to be highest for animals, which share the largest numbers of closely similar distractors.

Your alternative explanation rests on vague factors like “maximum information” which cannot be quantified. By contrast we are proposing a concrete, falsifiable model for three property-based tasks – same/different, oddball present/absent and object symmetry. Any argument based solely on item similarity to explain visual search or symmetry responses cannot explain systematic variations observed for target-absent arrays and for symmetric objects, for the reasons explained earlier.

DEFINITION OF AREA VH USING fMRI(1) The area VH boundaries from different experiments are nearly completely non-overlapping.In line with their theory that VH is a single continuum with a decision boundary somewhere in the middle, the authors use fMRI searchlight to find an area whose responses positively correlate with homogeneity, as calculated across all of their target present and target absent arrays. They report VH-correlated activity in regions anterior to LO. However, the VH defined by symmetry Experiments 3 and 4 (VHsymmetry) is substantially anterior to LO, while the VH defined by target detection Experiments 1 and 2 (VHdetection) is almost immediately adjacent to LO. Fig. S13 shows that VHsymmetry and VHdetection are nearly non-overlapping. This is a fundamental problem with the claim of discovering a new area that represents a new quantity that explains response times across multiple visual tasks. In addition, it is hard to understand why VHsymmetry does not show up in a straightforward subtraction between symmetric and asymmetric objects, which should show a clear difference in homogeneity.

We respectfully disagree. The partial overlap between the VH regions identified in Experiments 1 & 2 can hardly be taken as evidence against the quantity VH itself, because there are several other obvious alternate explanations for this partial overlap, as summarized earlier as well. The VH region does show up in a straightforward subtraction between symmetric and asymmetric objects (Section S7), so we are not sure what the Reviewer is referring to here.

(2) It is hard to understand how neural responses can be correlated with both VHpresent and VHabsent.The main paper results for VHdetection are based on both target-present and target-absent trials, considered together. It is hard to interpret the observed correlations, since the VHpresent and VHabsent metrics are calculated in such different ways and have opposite correlations with target similarity, task difficulty, and response times (see above). It may be that one or the other dominates the observed correlations. It would be clarifying to analyze correlations for target-present and target-absent trials separately, to see if they are both positive and correlated with each other.

Thanks for raising this point. We have now confirmed that the positive correlation between VH and neural response holds even when we do the analysis separately for target-present and -absent searches correlation between neural response in VH region and visual homogeneity (n = 32, r = 0.66, p < 0.0005 for target-present searches & n = 32, r = 0.56, p < 0.005 for target-absent searches).

(3) Definition of the boundaries and purpose of a new visual area in the brain requires circumspection, abundant and convergent evidence, and careful controls.Even if the VH metric, as defined and calculated by the authors here, is a meaningful quantity, it is a bold claim that a large cortical area just anterior to LO is devoted to calculating this metric as its major task. Vision involves much more than target detection and symmetry detection. Cortex anterior to LO is bound to perform a much wider range of visual functionalities. If the reported correlations can be clarified and supported, it would be more circumspect to treat them as one byproduct of unknown visual processing in cortex anterior to LO, rather than treating them as the defining purpose for a large area of visual cortex.

We totally agree with you that reporting a new brain region would require careful interpretation and abundant and converging evidence. However, this requires many studies worth of work, and historically category-selective regions like the FFA have achieved consensus only after they were replicated and confirmed across many studies. We believe our proposal for the computation of a quantity like visual homogeneity is conceptually novel, and our study represents a first step that provides some converging evidence (through replicable results across different experiments) for such a region. We have reworked our manuscript to make this point clearer (Discussion, p 32).

**Reviewer #3 (Public Review):**
Summary:This study proposes visual homogeneity as a novel visual property that enables observers perform to several seemingly disparate visual tasks, such as finding an odd item, deciding if two items are same, or judging if an object is symmetric. In Exp 1, the reaction times on several objects were measured in human subjects. In Exp 2, visual homogeneity of each object was calculated based on the reaction time data. The visual homogeneity scores predicted reaction times. This value was also correlated with the BOLD signals in a specific region anterior to LO. Similar methods were used to analyze reaction time and fMRI data in a symmetry detection task. It is concluded that visual homogeneity is an important feature that enables observers to solve these two tasks.

Thank you for your accurate and positive assessment.

Strengths:(1) The writing is very clear. The presentation of the study is informative.(2) This study includes several behavioral and fMRI experiments. I appreciate the scientific rigor of the authors.

We are grateful to you for your balanced assessment and constructive comments.

Weaknesses:(1) My main concern with this paper is the way visual homogeneity is computed. On page 10, lines 188-192, it says: "we then asked if there is any point in this multidimensional representation such that distances from this point to the target-present and target-absent response vectors can accurately predict the target-present and target-absent response times with a positive and negative correlation respectively (see Methods)". This is also true for the symmetry detection task. If I understand correctly, the reference point in this perceptual space was found by deliberating satisfying the negative and positive correlations in response times. And then on page 10, lines 200-205, it shows that the positive and negative correlations actually exist. This logic is confusing. The positive and negative correlations emerge only because this method is optimized to do so. It seems more reasonable to identify the reference point of this perceptual space independently, without using the reaction time data. Otherwise, the inference process sounds circular. A simple way is to just use the mean point of all objects in Exp 1, without any optimization towards reaction time data.

We disagree with you since the same logic applies to any curve-fitting procedure. When we fit data to a straight line, we are finding the slope and intercept that minimizes the error between the data and the straight line, but we would hardly consider the process circular when a good fit is achieved – in fact we take it as a confirmation that the data can be fit linearly. In the same vein, we would not have observed a good fit to the data, if there did not exist any good reference point relative to which the distances of the target-present and target-absent search arrays predicted these response times.

In Section S2, we show that the visual homogeneity estimates for each object is strongly correlated with the average distance of each object to all other objects (r = 0.84, p<0.0005, Figure S1).

We have performed several additional analyses to confirm the generality of our results and to reject alternate explanations (see Results, p. 12, Section titled “Confirming the generality of visual homogeneity”). In particular, to confirm that the results we obtained are not due to overfitting, we performed a cross-validation analysis, where we removed all searches involving a particular image and predicted these response times using visual homogeneity. This too revealed a significant model correlation confirming that our results are not due to overfitting.

(2) Visual homogeneity (at least given the current from) is an unnecessary term. It is similar to distractor heterogeneity/distractor variability/distractor statics in literature. However, the authors attempt to claim it as a novel concept. The title is "visual homogeneity computations in the brain enable solving generic visual tasks". The last sentence of the abstract is "a NOVEL IMAGE PROPERTY, visual homogeneity, is encoded in a localized brain region, to solve generic visual tasks". In the significance, it is mentioned that "we show that these tasks can be solved using a simple property WE DEFINE as visual homogeneity". If the authors agree that visual homogeneity is not new, I suggest a complete rewrite of the title, abstract, significance, and introduction.

We respectfully disagree that visual homogeneity is an unnecessary term. Please see our comments to Reviewer 1 above. Just like saliency and memorability can be measured empirically, we propose that visual homogeneity can be empirically measured as the reciprocal of the target-absent search time in a search task, or as the reciprocal of the “same” response time in a same-different task. Understanding how these three quantities interact will require measuring them empirically for an identical set of images, which is beyond the scope of this study but an interesting possibility for future work.

(3) Also, "solving generic tasks" is another overstatement. The oddball search tasks, same-different tasks, and symmetric tasks are only a small subset of many visual tasks. Can this "quantitative model" solve motion direction judgment tasks, visual working memory tasks? Perhaps so, but at least this manuscript provides no such evidence. On line 291, it says "we have proposed that visual homogeneity can be used to solve any task that requires discriminating between homogeneous and heterogeneous displays". I think this is a good statement. A title that says "XXXX enable solving discrimination tasks with multi-component displays" is more acceptable. The phrase "generic tasks" is certainly an exaggeration.

Thank you for your suggestion. We have now replaced the term “generic tasks” with the term property-based tasks, which we feel is more appropriate and reflect the fact that oddball search, same-different and symmetry tasks all involve looking for a specific image property.

(4) If I understand it correctly, one of the key findings of this paper is "the response times for target-present searches were positively correlated with visual homogeneity. By contrast, the response times for target-absent searches were negatively correlated with visual homogeneity" (lines 204-207). I think the authors have already acknowledged that the positive correlation is not surprising at all because it reflects the classic target-distractor similarity effect. But the authors claim that the negative correlations in target-absent searches is the true novel finding.(5) I would like to make it clear that this negative correlation is not new either. The seminal paper by Duncan and Humphreys (1989) has clearly stated that "difficulty increases with increased similarity of targets to nontargets and decreased similarity between nontargets" (the sentence in their abstract). Here, "similarity between nontargets" is the same as the visual homogeneity defined here. Similar effects have been shown in Duncan (1989) and Nagy, Neriani, and Young (2005). See also the inconsistent results in Nagy & Thomas, 2003, Vicent, Baddeley, Troscianko & Gilchrist, 2009. More recently, Wei Ji Ma has systematically investigated the effects of heterogeneous distractors in visual search. I think the introduction part of Wei Ji Ma's paper (2020) provides a nice summary of this line of research. I am surprised that these references are not mentioned at all in this manuscript (except Duncan and Humphreys, 1989).

You are right in noting that Duncan and Humphreys (1989) propose that searches are more difficult when nontargets are dissimilar. However, since our searches have identical distractors, the similarity between nontargets is always constant across target-absent searches, and therefore this cannot predict any systematic variation in target-absent search that is observed in our data. By contrast, our results explain both target-absent searches and target-present searches.

Thank you for pointing us to previous work. These studies show that it is not just the average distractor similarity but the statistics of the distractor similarity that drive visual search. However these studies do not explain why target-absent searches should vary systematically.

(6) If the key contribution is the quantitative model, the study should be organized in a different way. Although the findings of positive and negative correlations are not novel, it is still good to propose new models to explain classic phenomena. I would like to mention the three studies by Wei Ji Ma (see below). In these studies, Bayesian observer models were established to account for trial-by-trial behavioral responses. These computational models can also account for the set-size effect, behavior in both localization and detection tasks. I see much more scientific rigor in their studies. Going back to the quantitative model in this paper, I am wondering whether the model can provide any qualitative prediction beyond the positive and negative correlations? Can the model make qualitative predictions that differ from those of Wei Ji's model? If not, can the authors show that the model can quantitatively better account for the data than existing Bayesian models? We should evaluate a model either qualitatively or quantitatively.

Thank you for pointing us to prior work by Wei Ji Ma. These studies systematically examined visual search for a target among heterogeneous distractors using simple parametric stimuli and a Bayesian modeling framework. By contrast, our experiments involve searching for single oddball targets among multiple identical distractors, so it is not clear to us that the Wei Ji Ma models can be easily used to generate predictions about these searches used in our study.

We are not sure what you mean by offering quantitative predictions beyond positive and negative correlations. We have tried to explain systematic variation in target-present and target-absent response times using a model of how these decisions are being made. Our model explains a lot of systematic variation in the data for both types of decisions.

(7) In my opinion, one of the advantages of this study is the fMRI dataset, which is valuable because previous studies did not collect fMRI data. The key contribution may be the novel brain region associated with display heterogeneity. If this is the case, I would suggest using a more parametric way to measure this region. For example, one can use Gabor stimuli and systematically manipulate the variations of multiple Gabor stimuli, the same logic also applies to motion direction. If this study uses static Gabor, random dot motion, object images that span from low-level to high-level visual stimuli, and consistently shows that the stimulus heterogeneity is encoded in one brain region, I would say this finding is valuable. But this sounds like another experiment. In other words, it is insufficient to claim a new brain region given the current form of the manuscript.

We agree that parametric stimulus manipulations are important for studying early visual areas where stimulus dimensions are known (e.g. orientation, spatial frequency). Using parametric stimulus manipulations for more complex stimuli is fraught with issues because the underlying representation may not be encoding the dimensions being manipulated. This is the reason why we attempted to recover the underlying neural representation using dissimilarities measured using visual search, and then asked whether a decision making process operating on this underlying representation can explain how decisions are made. Therefore we disagree that parametric stimulus manipulations are the only way to obtain insight into such tasks.

We have proposed a quantitative model that explains how decisions about target present and absent can be made through distance-to-center computations on an underlying object representation. We feel that the behavioural and the brain imaging results strongly point to a novel computation that is being performed in a localized region in the brain. These results represent an important first step in understanding how complex, property-based tasks are performed by the brain. We have revised our manuscript to make this point clearer.

REFERENCES

- Duncan, J., & Humphreys, G. W. (1989). Visual search and stimulus similarity. Psychological Review, 96(3), 433-458. doi: 10.1037/0033-295x.96.3.433

- Duncan, J. (1989). Boundary conditions on parallel processing in human vision. Perception, 18(4), 457-469. doi: 10.1068/p180457

- Nagy, A. L., Neriani, K. E., & Young, T. L. (2005). Effects of target and distractor heterogeneity on search for a color target. Vision Research, 45(14), 1885-1899. doi: 10.1016/j.visres.2005.01.007

- Nagy, A. L., & Thomas, G. (2003). Distractor heterogeneity, attention, and color in visual search. Vision Research, 43(14), 1541-1552. doi: 10.1016/s0042-6989(03)00234-7

- Vincent, B., Baddeley, R., Troscianko, T., & Gilchrist, I. (2009). Optimal feature integration in visual search. Journal of Vision, 9(5), 15-15. doi: 10.1167/9.5.15

- Singh, A., Mihali, A., Chou, W. C., & Ma, W. J. (2023). A Computational Approach to Search in Visual Working Memory.

- Mihali, A., & Ma, W. J. (2020). The psychophysics of visual search with heterogeneous distractors. BioRxiv, 2020-08.

- Calder-Travis, J., & Ma, W. J. (2020). Explaining the effects of distractor statistics in visual search. Journal of Vision, 20(13), 11-11.

**Recommendations for the authors:**

**Reviewer #1 (Recommendations For The Authors):**
The authors have not made substantive changes to address my major concerns. Instead, they have responded with arguments about why their original manuscript was good as written. I did not find these arguments persuasive. Given that, I've left my public review the same, since it still represents my opinions about the paper. Readers can judge which viewpoints are more persuasive.

We respectfully disagree: we have tried our best to address your concerns with additional analysis wherever feasible, and by acknowledging any limitations.

**Reviewer #3 (Recommendations For The Authors):**
(1) As I mentioned above, please consider rewriting title, abstract, introduction, and significance. Please remove the word "visual homogeneity" and instead use distractor heterogeneity/distractor variability/distractor statistics as often used in literature.

To clarify, visual homogeneity is NOT the same as distractor homogeneity. Visual homogeneity refers to a distance-to-center computation and represents an image-computable property that can vary systematically even when all distractors are identical. By contrast distractor heterogeneity varies only when distractors are different from each other.

(2) Better to remove the phrase "generic tasks".

Thanks for your suggestions. We now refer to these tasks as property-based tasks.

(3) Better to explicitly specify the predictions made by the quantitative model beyond positive and negative correlations.

The predictions of the quantitative model are to explain systematic variation in the response times. We are not sure what else is there to predict in the response times.

(4) If the quantitative model is the key contribution, better to highlight the details and algorithmic contribution of the model, and show the advantage of this model either qualitatively and quantitatively.

Please see our responses above. Our quantitative model explains behavior and brain imaging data on three disparate tasks – the same/different, oddball visual search and symmetry tasks.

(5) If the new brain region is the key contribution, better to downplay the quantitative model.

Please see our responses above. Our quantitative model explains behavior and brain imaging data on three disparate tasks – the same/different, oddball visual search and symmetry tasks.